# Cutting out the middleman: Calibrating and validating a dynamic vegetation model (ED2-PROSPECT5) using remotely sensed surface reflectance

Alexey N. Shiklomanov[1], Michael C. Dietze[2], Istem Fer[3], Toni Viskari[3], and Shawn P. Serbin[4]

[1]NASA Goddard Space Flight Center, Greenbelt, MD, USA
[2]Department of Earth and Environment, Boston University, Boston, MA, USA
[3]Finnish Meteorological Institute, Helsinki, Finland
[4]Environmental and Climate Sciences Department, Brookhaven National Laboratory, Upton, NY, USA

**Correspondence:** Dr. Alexey N. Shiklomanov (alexey.shiklomanov@nasa.gov)

**Abstract.** Canopy radiative transfer is the primary mechanism by which models relate vegetation composition and state to the surface energy balance, which is important to light- and temperature-sensitive plant processes as well as understanding land-atmosphere feedbacks. In addition, certain parameters (e.g., SLA) that have an outsized influence on vegetation model behavior can be constrained by observations of shortwave reflectance, thus reducing model predictive uncertainty. Importantly, calibrating against radiative transfer outputs allows models to directly use remote sensing reflectance products without relying on highly derived products (such as MODIS leaf area index) whose assumptions may be incompatible with the target vegetation model and whose uncertainties are usually not well quantified. Here, we coupled the two-stream representation of canopy radiative transfer in the Ecosystem Demography model (ED2) with a leaf radiative transfer model (PROSPECT 5) and a simple soil reflectance model to predict full-range, high spectral resolution surface reflectance that is dependent on the underlying ED2 model state. We then calibrated this model against estimates of hemispherical reflectance (corrected for directional effects) from the NASA Airborne VIsible/InfraRed Imaging Spectrometer (AVIRIS) and survey data from 54 temperate forest plots in the northeastern United States. The calibration significantly reduced uncertainty in model parameters related to leaf biochemistry and morphology and canopy structure for five plant functional types. Using a single common set of parameters across all sites, the calibrated model was able to accurately reproduce surface reflectance for sites with highly varied forest composition and structure. However, the calibrated model's predictions of leaf area index (LAI) were less robust, capturing only 46% of the variability in the observations. Comparing the ED2 radiative transfer model with a similar model commonly used in remote sensing studies (PRO4SAIL) illustrated structural errors in the ED2 representation of direct radiation backscatter that resulted in systematic underestimation of reflectance. In addition, we also highlight that, to directly compare with a two-stream radiative transfer model like EDR, we had to perform an additional processing step to convert the directional reflectance estimates of AVIRIS to hemispherical reflectance (a.k.a., "albedo"). In future work, we recommend that vegetation models add the capability to predict directional reflectance, to allow them to more directly assimilate a wide range of airborne and satellite reflectance products. We ultimately conclude that despite these challenges, using dynamic vegetation models to predict surface reflectance is a promising avenue for model calibration and validation using remote sensing data.

## 25  1  Introduction

Dynamic vegetation models play a vital role in modern terrestrial ecology, and Earth science more generally. The terrestrial carbon cycle is a major biogeochemical feedback in the global climate system (Heinze et al., 2019), and accurate predictions of terrestrial carbon cycling rely on accurate representations of vegetation dynamics (Pacala and Deutschman, 1995). Vegetation also plays an important role in the water cycle and surface energy balance, with major climate implications (Bonan, 2008).

In addition, the distribution of tree species, the structure of plant canopies, and many other variables simulated by dynamic vegetation models are also important predictors of biodiversity, making vegetation models an important tool for conservation management (McMahon et al., 2011). Robust calibration and validation of model projections is therefore of broad concern.

Past efforts to calibrate or constrain dynamic vegetation model parameters and states used a variety of data streams. Among these data streams, remote sensing is particularly promising due to its consistent measurement methodology and largely un-

interrupted global coverage in recent decades. Data products derived from remote sensing observations have been used to inform, among others, vegetation phenology (Knorr et al., 2010; Viskari et al., 2015) and absorbed photosynthetically-active radiation (Peylin et al., 2016; Schürmann et al., 2016; Zobitz et al., 2014). However, there are issues with using derived remote sensing products to calibrate vegetation models. The relationships between remotely sensed surface reflectance and vegetation structure and function are complex and multifaceted. Simple polynomial relationships between spectral indices (e.g., Normal-

ized Difference Vegetation Index, NDVI; Enhanced Vegetation Index, EVI) and vegetation properties (e.g., leaf area index, LAI) are often confounded by other ecosystem characteristics, including soil (Myneni and Williams, 1994) and snow (Zhang et al., 2020), or sensor configuration (Fensholt et al., 2004). More sophisticated approaches for estimating vegetation properties based on physically-based radiative transfer models face issues of equifinality, whereby many different combinations of vegetation and soil properties can ultimately produce the same modeled surface reflectance (Combal et al., 2003; Lewis

and Disney, 2007). Meanwhile, estimating quantities with more indirect relationships to surface reflectance, such as rates of primary productivity, requires a number of assumptions about resource use efficiency and other factors (Running et al., 2004) that can introduce considerable uncertainty and bias into the estimates. Collectively, these issues help explain the large differences in estimates of surface characteristics across different remote sensing instruments (Liu et al., 2018). Robust, pixel-level uncertainty estimates for remote sensing data products would help alleviate some of these concerns, but such estimates are not

widely available for most data products.

One way to overcome these limitations of derived remote sensing data products while still leveraging the capabilities of remote sensing is to work with lower-level surface reflectance products. Although generating these reflectance products still requires multiple processing steps, such as atmospheric correction, orthorectification, and correction for sun-sensor geometry effects, all of these processing steps involve significantly fewer assumptions about the relationship between the remotely

sensed signal and the surface property or phenomenon of interest than derived products. This can be accomplished by coupling dynamic vegetation models with leaf and canopy radiative transfer models that simulate surface reflectance as a function

of known surface characteristics (Knorr and Lakshmi, 2001; Nouvellon et al., 2001; Quaife et al., 2008). Such an approach draws on decades of research on simulation of vegetation optical properties given their structural and biochemical characteristics (Dickinson, 1983; Sellers, 1985; Verhoef, 1984; Lewis and Disney, 2007; Jacquemoud et al., 2009; Pinty et al., 2004;

Widlowski et al., 2007, 2015; Hogan et al., 2018) while avoiding the computational and conceptual challenges of inverse parameter estimation in radiative transfer modeling (Combal et al., 2003; Lewis and Disney, 2007).

Instead of coupling a dynamic vegetation model to an external canopy radiative transfer model, we propose a more direct approach of using a vegetation model's own internal representation of canopy radiative transfer. Radiative transfer models have long been an important component of land surface models (Dickinson, 1983; Sellers, 1985). Canopy radiative transfer is the

primary mechanism by which models relate vegetation composition and state to the surface energy balance. This is important to both the plants themselves—as many plant processes (including evaporation and enzyme kinetics) are sensitive to temperature (Serbin et al., 2012)—and to the impact of plants on local, regional, and global climate (Bonan, 2008). Canopy radiative transfer also controls how much light is available to plants for photosynthesis and is therefore a first-order driver of plant function (Hikosaka and Terashima, 1995; Robakowski et al., 2004; Niinemets, 2016; Keenan and Niinemets, 2016). Canopy

radiative transfer is particularly important to the current generation of demographically-enabled dynamic vegetation models, where differences in canopy radiative transfer representations and parameters have major impacts on predicted community composition and biogeochemistry (Loew et al., 2014; Fisher et al., 2018; Viskari et al., 2019). Finally, parameters to which vegetation models are known to be highly sensitive—namely, those related to leaf biochemistry and canopy structure (Dietze et al., 2014; Raczka et al., 2018; Shiklomanov et al., 2020a)—play an important role in canopy radiative transfer. Therefore,

calibration and validation against radiative transfer outputs can be an important source of constraint on a variety of model processes.

Our previous work demonstrated that predictions of carbon cycling and community composition by the Ecosystem Demography model, version 2 (ED2; Medvigy et al., 2009) are highly sensitive to changes in parameters related to canopy structure and radiative transfer (Viskari et al., 2019). In this study, we build on this work by calibrating and validating the ED2 model us-

ing remotely sensed surface reflectance. First, we couple the internal ED2 canopy radiative transfer model to the PROSPECT 5 leaf radiative transfer model (Feret et al., 2008) and the Hapke soil reflectance model (Verhoef and Bach, 2007) to allow ED2 to predict surface reflectance spectra at 1 nm resolution across the complete visible-shortwave infrared (VSWIR) spectral region (400 to 2500 nm). Second, we jointly calibrate this model at 54 sites in the US Midwest and Northeast where coincident vegetation survey data and NASA Airborne Visible/Infrared Imaging Spectrometer-Classic (AVIRIS-Classic) surface reflectance

observations are available. We hypothesize that, with known stand composition and informative priors on foliar biochemistry, calibration against airborne imaging spectroscopy will significantly constrain model parameters related to canopy structure. Although the scope of our study is limited to the ED2 model, both the underlying size-and-age structure approximation of ED2 as well as many aspects of its canopy radiative transfer (e.g., two-stream approximation, treatment of leaf angles) are common to other land surface models (e.g., FATES; Koven et al., 2020), meaning the insights from this work more broadly applicable

in model vegetation modeling.

## 2 Methods

### 2.1 ED2 model description

The Ecosystem Demography version 2.2 (ED2) model simulates plot-level vegetation dynamics and biogeochemistry (Moorcroft et al., 2001; Medvigy et al., 2009; Longo et al., 2019). ED2 has a fundamentally hierarchical structure: The fundamental unit of analysis is a plant *cohort*—a group of individual plants of similar size, age, and species composition (grouped into plant functional types, PFTs). A group of cohorts with a common disturbance history (i.e., time since last disturbance) constitutes a *patch*, and a group of co-located patches experiencing the same meteorological conditions constitute a *site*. At the spatial scale of this work (60 x 60m plots; see Section 2.3), we assume one patch per site.

Relevant to this work, ED2 includes a multi-layer canopy radiative transfer model that is a generalization of the two-layer two-stream radiative transfer scheme in CLM 4.5 (Oleson et al., 2013), which in turn is derived from Sellers (1985). At every time step, this model simulates the bi-hemispherical reflectance (BHR;a.k.a., "intrinsic surface albedo" or "blue-sky albedo"; see Schaepman-Strub et al. 2006) as a function of that time step's vegetation composition and canopy structure. A complete description of the model derivation is provided in the supplementary information of Longo et al. (2019), but for completeness, we provide an abbreviated description below.

### 2.1.1 Radiative transfer parameters

Two-stream radiative transfer theory (Meador and Weaver, 1980) defines the change in radiative flux through a medium in terms of hemispherically-integrated upward ($F_i^\uparrow$) and downward ($F_i^\downarrow$) radiative fluxes via the following system of differential equations (adapting the notation of Yuan et al. 2017):

$$-\frac{dF_i^\uparrow}{dx} = -\underbrace{(a_i + \gamma_i)F_i^\uparrow}_{\text{Interception}} + \underbrace{\gamma_i F_i^\downarrow}_{\text{Diffuse scatter}} + \underbrace{s_i F_i^\odot}_{\text{Direct backscatter}} \tag{1}$$

$$\frac{dF_i^\downarrow}{dx} = -\underbrace{(a_i + \gamma_i)F_i^\downarrow}_{\text{Interception}} + \underbrace{\gamma_i F_i^\uparrow}_{\text{Diffuse scatter}} + \underbrace{s_i' F_i^\odot}_{\text{Direct scatter}} \tag{2}$$

where $dx$ represents the vertical change in the total plant area index (combined area of leaves and woody elements), $a$ describes absorption of diffuse radiation, $\gamma$ describes scattering of diffuse radiation, $s$ and $s'$ describe the upward and downward scattering of direct ("beam") radiation, and $F_i^\odot$ is the incident direct (or "beam") radiative flux at canopy layer $i$.

Following Sellers (1985), the coefficients above are defined as follows:

$$a_i + \gamma_i = [1 - (1 - \beta_i)] \frac{1}{\bar{\mu}_i} \tag{3}$$

$$\gamma_i = \beta_i \omega_i \frac{1}{\bar{\mu}_i} \tag{4}$$

$$s_i = \frac{1}{\mu_i^{\odot}} \omega_i \beta_{0,i} \tag{5}$$

$$s_i' = \frac{1}{\mu_i^{\odot}} \omega_i (1 - \beta_{0,i}) \tag{6}$$

where $\bar{\mu}_i$ is the optical depth per unit plant area index for diffuse radiation, $\beta_i$ is the backscattering coefficient for diffuse

radiation, $\omega_i$ is the scattering coefficient for diffuse radiation, $\mu_i^{\odot}$ is the optical depth per unit plant area index for direct radiation, and $\beta_0$ is the backscattering coefficient for direct radiation.

For a given incident radiation (i.e., solar zenith) angle $\theta$ and leaf orientation angle $\varphi$, the optical depth is defined as:

$$\mu(\theta, \varphi) = \frac{\cos(\theta)}{G(\theta, \varphi)} \tag{7}$$

where $G(\theta, \varphi)$ is a function describing the projected leaf area. Following Goudriaan (1977), this function can be approxi-

mated as:

$$G(\theta, \varphi) \approx G^*(\theta, \chi_i) = \phi_{1,i} + \phi_{2,i} \cos(\theta) \tag{8}$$

$$\phi_{1,i} = 0.5 - 0.633\chi_i - 0.33\chi_i^2 \tag{9}$$

$$\phi_{2,i} = 0.877 - (1 - 2\phi_{1,i}) \tag{10}$$

where $\chi$ is the *leaf orientation factor*—a PFT-specific parameter whose values theoretically range from -1 (vertical leaves)

through 0 (randomly-distributed leaf angles) to 1 (horizontal leaves), but in practice are restricted to $-0.4 \leq \chi \leq 0.6$.

For a given solar zenith angle ($\theta_s$), the optical depth for direct radiation, $\mu_i^{\odot}$, is defined as:

$$\mu_i^{\odot} = \mu(\theta_s, \chi_i) = \frac{\cos(\theta_s)}{G^*(\theta_s, \chi_i)} \tag{11}$$

The optical depth for diffuse radiation, $\bar{\mu}_i$, is defined as the integral of equation 7 over all zenith angles:

$$\bar{\mu}_i = \int_0^{\frac{\pi}{2}} \frac{\cos(\theta)}{G^*(\theta, \chi_i)} d\theta = \frac{1}{\phi_{2,i}} \left[ 1 + \frac{\phi_{1,i}}{\phi_{2,i}} \ln \frac{\phi_{1,i}}{\phi_{1,i} + \phi_{2,i}} \right] \tag{12}$$

Following Sellers (1985), diffuse scattering ($\omega$) and backscattering ($\beta$) coefficients of canopy elements (leaves or stems) are defined as a function of those elements' reflectance ($R$) and transmittance ($T$; wood transmittance is assumed to be zero). (We use index $p$ to refer to PFT and $p(i)$ to refer to the PFT of cohort $i$).

$$\omega_{i,\text{leaf}} = R_{p(i),\text{leaf}} + T_{p(i),\text{leaf}} \tag{13}$$

$$\omega_{i,\text{wood}} = R_{p(i),\text{wood}} \tag{14}$$

$$\beta_{i,\text{leaf}} = \frac{1}{2\omega_i} \left[ R_{i,\text{leaf}} + T_{i,\text{leaf}} + (R_{i,\text{leaf}} - T_{i,\text{leaf}}) J(\chi_i) \right] \tag{15}$$

$$\beta_{i,\text{wood}} = \frac{1}{2\omega_i} \left[ R_{i,\text{wood}} + R_{i,\text{wood}} J(\chi_i) \right] \tag{16}$$

where $J(\chi_i)$ captures the effect of leaf and branch inclination and is approximated as (similarly to Oleson et al. 2013):

$$J(\chi_i) = \frac{1 + \chi_i}{2} \tag{17}$$

Both $\omega$ and $\beta$ are calculated independently for leaves and wood and then averaged based on the relative effective area of leaves ($L_i$) and wood ($W_i$) within a canopy layer.

$$\omega_i = \omega_{i,\text{leaf}} \frac{L_i}{L_i + W_i} + \omega_{i,\text{wood}} (1 - \frac{L_i}{L_i + W_i}) \tag{18}$$

$$\beta_i = \beta_{i,\text{leaf}} \frac{L_i}{L_i + W_i} + \beta_{\text{wood}} (1 - \frac{L_i}{L_i + W_i}) \tag{19}$$

To account for non-uniform distribution of leaves within a canopy, ED2 has a PFT-specific *clumping factor* ($q$) parameter that serves as a scaling factor on leaf area index. Therefore the effective leaf area index ($L$) is related to the true leaf area index (LAI) by:

$$L_i = \text{LAI}_i \times q_{p(i)} \tag{20}$$

The leaf area of a cohort ($\text{LAI}_i$) is calculated as a function of leaf biomass ($B_{\text{leaf},i}$, kgC plant$^{-1}$), specific leaf area (SLA$_p$, m$^2$ kgC$^{-1}$), and stem density ($n_{\text{plant}}$, plants m$^{-2}$):

$$\text{LAI}_i = n_{\text{plant},i} B_{\text{leaf},i} \text{SLA}_{p(i)} \tag{21}$$

In turn, $B_{\text{leaf},i}$ is calculated from cohort diameter at breast height (DBH$_i$, cm) according to the following allometric equations:

$$B_{\text{leaf},i} = \text{b1Bl}_{p(i)} \text{DBH}_i^{\text{b2Bl}_{p(i)}} \tag{22}$$

where $\text{b1Bl}_{p(i)}$ and $\text{b2Bl}_{p(i)}$ are PFT-specific parameters. The wood area of a cohort ($\text{WAI}_i$) is calculated directly from DBH according to a similar allometric equation:

$$\text{WAI}_i = n_{\text{plant},i}\text{b1Bw}_{p(i)}\text{DBH}_i^{\text{b2Bw}_{p(i)}} \tag{23}$$

where $\text{b1Bw}_{p(i)}$ and $\text{b2Bw}_{p(i)}$ are PFT-specific parameters.

Backscattering of direct radiation ($\beta_i^\odot$) is defined as a function of single scattering albedo, ($\alpha_s(\theta_s)$):

$$\beta_i^\odot = \frac{\bar{\mu}_i + \mu_i^\odot}{\bar{\mu}_i}\alpha_s(\theta_s) \tag{24}$$

Single scattering albedo ($\alpha_s(\theta_s)$) is in turn defined as an integral over all illumination angles ($\vartheta$), following Sellers (1985) :

$$\alpha_s(\theta_s) = \omega \int_0^{\frac{\pi}{2}} \frac{\Gamma(\theta_s,\vartheta)\cos(\vartheta)}{G(\theta_s,\varphi)\cos(\theta_s) + G(\vartheta,\varphi)cos(\vartheta)}\sin(\vartheta)d\vartheta \tag{25}$$

$$\Gamma(\theta,\vartheta) = G(\theta,\varphi)G(\vartheta,\varphi)P(\theta,\vartheta) \tag{26}$$

$$\int_{-\frac{\pi}{2}}^{\frac{\pi}{2}} P(\theta,\vartheta)G(\vartheta,\varphi)\sin(\vartheta)d\vartheta = 1 \tag{27}$$

where $P(\theta,\vartheta)$ is the scattering phase function that defines the relative fraction of scattered flux in any direction relative to the projected leaf area in that direction (Dickinson, 1983). Substituting $G = G^*$ (equation 8), and assuming uniform scattering (i.e., $P(\theta,\vartheta) = \frac{1}{4\pi}$, and therefore, $\Gamma(\theta,\vartheta) = \frac{G(\theta,\varphi)}{2}$; see detailed discussion of these assumptions in Yuan et al. 2017) gives the following analytical solution to this integral:

$$\alpha_{s,i}(\theta) = \frac{\omega_i}{2\left(1 + \phi_{2,i}\mu_i^\odot\right)}\left[1 - \frac{\phi_{1,i}\mu_i^\odot}{1 + \phi_{2,i}\mu_i^\odot}\ln\left(\frac{1 + (\phi_{1,i} + \phi_{2,i})\mu_i^\odot}{\phi_{1,i}\mu_i^\odot}\right)\right] \tag{28}$$

### 2.1.2 Solution for the multi-layer canopy

In ED2, the direct radiation profile, $F_i^\odot$, is governed by exponential decay, following Beer's Law:

$$F_i^\odot = F_{i+1}^\odot \exp\left(-\frac{\text{TAI}_i}{\mu_i^\odot}\right) \tag{29}$$

$$F_{n+1}^\odot = F_{\text{sky}}^\odot \tag{30}$$

where $F_{\text{sky}}^\odot$ is the incident direct ("beam") radiation from the atmosphere, a prescribed input; and $\text{TAI}_i$ is the total plant area index, defined as the sum of effective leaf area index ($L_i$; equation 20) and wood area index (WAI)

For $n$ cohorts, the full diffuse canopy radiation profile in ED2 is defined by a vector $\boldsymbol{A}$ of size $2n+2$ that contains the upward ($F_i^{\uparrow}$) and downward ($F_i^{\downarrow}$) radiative fluxes for every "interface" *immediately below cohort $i$*; therefore, $F_{(n+1)}^{\uparrow}$ refers to the upward diffuse radiative flux from the top of the canopy towards the atmosphere (the quantity used to calculate the albedo), $F_{(n+1)}^{\downarrow}$ refers to the downward diffuse radiative flux from the atmosphere into the top of the canopy, $F_1^{\uparrow}$ refers to the upward diffuse radiative flux from the ground into the canopy layer of the shortest cohort, and $F_1^{\downarrow}$ refers to the downard diffuse radiative flux from the canopy layer of the shortest cohort towards the ground. To derive each of these $F$ terms, ED2 uses the following analytical solution for equations 2 and 1 (see Longo et al. 2019, Section S12, for a full derivation):

$$F_i^{\downarrow} = x_{(2i-1)}\gamma_i^{+}\exp\left(-\lambda_i\mathrm{TAI}\right) + x_{(2i)}\gamma_i^{-}\exp\left(+\lambda_i\mathrm{TAI}\right) + \delta^{+}\exp\left(-\frac{\mathrm{TAI}_i}{\mu_i^{\odot}}\right) \tag{31}$$

$$F_i^{\uparrow} = x_{(2i-1)}\gamma_i^{-}\exp\left(-\lambda_i\mathrm{TAI}\right) + x_{(2i)}\gamma_i^{+}\exp\left(+\lambda_i\mathrm{TAI}\right) + \delta^{-}\exp\left(-\frac{\mathrm{TAI}_i}{\mu_i^{\odot}}\right) \tag{32}$$

where $x_i$ is a vector of wavelength- and cohort-specific unknowns, and the remaining terms are:

$$\gamma_i^{\pm} = \frac{1}{2}\left(1\pm\sqrt{\frac{1-\omega_i}{1-(1-2\beta_i)\omega_i}}\right) \tag{33}$$

$$\delta_i^{\pm} = \frac{\left(\kappa^{+}\pm\kappa^{-}\right)\mu_i^{\odot 2}}{2\left(1-\lambda_i^2\mu_i^{\odot 2}\right)} \tag{34}$$

$$\lambda_i^2 = \frac{\left[1-(1-2\beta_i)\omega_i\right]\left(1-\omega_i\right)}{\bar{\mu}_i^2} \tag{35}$$

$$\kappa_i^{+} = -\left[\frac{1-(1-2\beta_i)\omega_i}{\bar{\mu}_i} + \frac{1-2\beta_i}{\mu_i^{\odot}}\right]\frac{\omega_i F_{(i+1)}^{\odot}}{\mu_k^{\odot}} \tag{36}$$

$$\kappa_i^{-} = -\left[\frac{(1-2\beta_i)\left(1-\omega_i\right)}{\bar{\mu}_i} + \frac{1}{\mu_i^{\odot}}\right]\frac{\omega_i F_{(i+1)}^{\odot}}{\mu_k^{\odot}} \tag{37}$$

The problem of solving for $x_i$ in equations 31 and 32 can be written as a matrix equation:

$$\mathbf{S}\boldsymbol{x} = \boldsymbol{Y} \tag{38}$$

where $\boldsymbol{x} = (x_1, x_2, \ldots, x_{2n+1}, x_{2n+2})$, $\boldsymbol{y} = \left(y_1, y_2, \ldots, y_{(2n+1)}, y_{(2n+2)}\right)$, and $\mathbf{S}$ is a $(2n+2)\times(2n+2)$ tridiagonal matrix. To solve this matrix equation, ED2 defines the following boundary conditions: At the top of the canopy ($i = n+1$), $F_{(n+1)}^{\downarrow} \equiv F_{\mathrm{sky}}^{\downarrow}$, the incident diffuse flux from the atmosphere (a prescribed input); $\mathrm{TAI}_{(n+1)} = 0$; $\bar{\mu}_{(n+1)} = 1$; $\omega_{(n+1)} = 1$ (no absorption); and $\beta_{(n+1)} = \beta_{(n+1)}^{\odot} = 0$ (no scattering; all radiance is transmitted). At the bottom of the canopy ($i = 1$), we re-define the ground scattering, $\omega_g$, based on a soil radiative transfer model (Section 2.2). With these boundary conditions, the elements of $\mathbf{S}$ are given by:

$$S_{1,1} = \left(\gamma_1^- - \omega_g \gamma^+\right)\exp(-\lambda_1 \mathrm{TAI}_1)$$

$$S_{1,2} = \left(\gamma_1^+ - \omega_g \gamma^-\right)\exp(+\lambda_1 \mathrm{TAI}_1)$$

$$S_{(2i,2i-1)} = \gamma_i^+$$

$$S_{(2i,2i)} = \gamma_i^-$$

$$S_{(2i,2i+1)} = -\gamma_{(i+1)}^+ \exp(-\lambda_{(i+1)} \mathrm{TAI}_{(i+1)})$$

$$S_{(2i,2i+2)} = -\gamma_{(i+1)}^- \exp(+\lambda_{(i+1)} \mathrm{TAI}_{(i+1)})$$

$$S_{(2i+1,2i-1)} = \gamma_i^-$$

$$S_{(2i+1,2i)} = \gamma_i^+$$

$$S_{(2n,2n+1)} = -\gamma_{(n+1)}^+ \exp(-\lambda_{(n+1)} \mathrm{TAI}_{(n+1)})$$

$$S_{(2n,2n+2)} = -\gamma_{(n+1)}^- \exp(+\lambda_{(n+1)} \mathrm{TAI}_{(n+1)})$$

$$(39)$$

and the elements of $\boldsymbol{y}$ are given by:

$$y_1 = \omega_0 F_1^\odot - \left(\delta_1^- - \omega_g \delta_1^+\right)\exp\left(-\frac{\mathrm{TAI}}{\mu_1^\odot}\right)$$

$$y_{(2i)} = \delta_{(i+1)}^+ \exp\left(-\frac{\mathrm{TAI}_{(i+1)}}{\mu_{(i+1)}^\odot}\right) - \delta_i^+$$

$$y_{(2i+1)} = \delta_{(i+1)}^- \exp\left(-\frac{\mathrm{TAI}_{(i+1)}}{\mu_{(i+1)}^\odot}\right) - \delta_i^-$$

$$y_{(2n+2)} = F_{\mathrm{sky}}^\downarrow - \delta_{(n+1)}^+$$

$$(40)$$

Finally, the surface albedo ($\rho$) is defined as the fraction of the total radiative flux incident on the canopy ($F_{\mathrm{sky}}^\odot + F_{\mathrm{sky}}^\downarrow$) that is reflected:

$$\rho = \frac{F_{(n+1)}^\uparrow}{F_{\mathrm{sky}}^\odot + F_{\mathrm{sky}}^\downarrow} \tag{41}$$

## 2.2 ED2-PROSPECT coupling

By default, ED2 performs the canopy shortwave radiative transfer calculations described in two broad spectral bands: visible (400–700 nm) and near-infrared (700–2500 nm). For each of these regions, ED2 has user-defined prescribed, PFT-specific

leaf and wood reflectance and transmittance values, and calculates soil reflectance as the average of constant wet and dry soil reflectance values weighted by the relative soil moisture (0 = fully dry, 1 = fully wet). In this study, we modified ED2 to perform
the same canopy radiative transfer calculations but in 1 nm increments across the range 400–2500 nm. We then simulated leaf reflectance and transmittance using the PROSPECT 5 leaf RTM, which has the following five parameters: Effective number of leaf mesophyll layers ($N$, unitless, >= 1), total chlorophyll content ($Cab$, $\mu g\, cm^{-2}$), total carotenoid content ($Car$, $\mu g\, cm^{-2}$), water content ($Cw$, $g\, cm^{-2}$), and dry matter content ($Cm$, $g\, cm^{-2}$) (Feret et al., 2008). For wood reflectance, we used a single representative spectrum—the mean of all wood spectra from Asner (1998), resampled to 1 nm resolution—for all PFTs.
For soil scattering ($\omega_g$), we used the simple Hapke soil submodel used in the Soil-Leaf-Canopy RTM (Verhoef and Bach, 2007), whereby soil reflectance is the average of prescribed wet and dry soil reflectance spectra weighted by a relative soil moisture parameter ($\varrho_{soil}$, unitless, 0–1). The final coupled PROSPECT-ED2 canopy radiative transfer model (hereafter known as "EDR") has 12 parameters for each PFT— 5 parameters for PROSPECT, specific leaf area, two parameters each for the leaf and wood allometries, and clumping ($q$) and leaf orientation ($\chi$) factors—and one site-specific parameter—the relative soil
moisture (Table 1).

EDR shares many assumptions and internal coefficients with SAIL (Verhoef, 1984; Verhoef and Bach, 2007), a canopy radiative transfer model that is popular in the optical remote sensing community due to its ability to simulate both hemispherical and directional reflectance. Unlike EDR's vertically heterogeneous canopy, SAIL takes only a single homogenous canopy layer as an input, which precludes a valid comparison of the two models' simulations for real heterogeneous sites. Nevertheless, to
230 help identify possible structural issues with EDR, and to explore differences between hemispherical and directional reflectance streams, we compared the sensitivities of EDR and SAIL to LAI and solar zenith angles for a single-layer homogeneous canopy.

## 2.3 Site and data description

For model calibration, we selected 54 sites from the NASA Forest Functional Types (FFT) field campaign that contained
plot-level inventory data coincident with observations of the NASA Airborne Visible/Infrared Imaging Spectrometer-Classic (AVIRIS-Classic). A full description of this dataset is provided in Singh et al. (2015). Briefly, each site consisted of a 60 × 60 m transect within which forest inventory data (stem density, species identity, and diameter at breast height, DBH) were collected. These sites are located in the United States Upper Midwest, northern New York, and western Maryland (Figure 1), and include stands dominated by either evergreen or deciduous trees and spanning a wide range of structures, from dense
groups of saplings to sparse groups of large trees (Figure 2).

For this study, because our goal was only to calibrate the ED2 canopy radiative transfer parameters and not to evaluate ED2 predictions of vegetation dynamics, we prescribed the vegetation composition at each site based on the inventory data described above. We grouped the tree species in these sites into five different PFTs as defined by ED2: Early successional hardwood, northern mid-successional hardwood, late successional hardwood, northern pine, and late successional conifer. The mappings
of tree species onto these PFTs are provided as a CSV-formatted table in the file inst/pfts-species.csv in the source code repository for this project (see Code and Data Availability section).

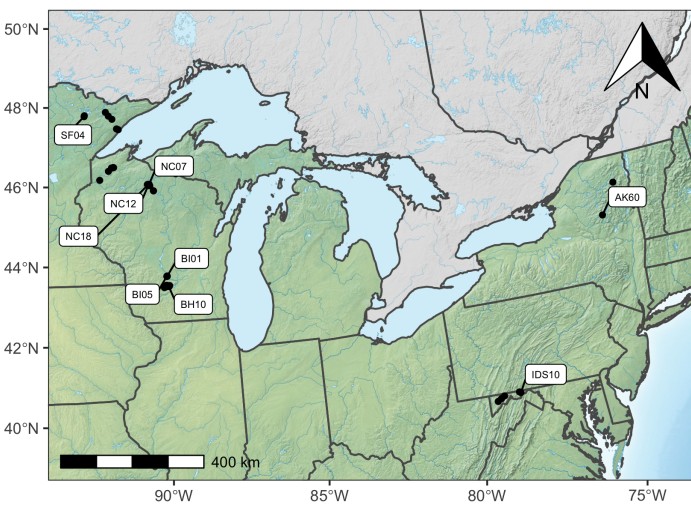

**Figure 1.** Map of sites used in this analysis. Sites shown in Figure 6 are labeled.

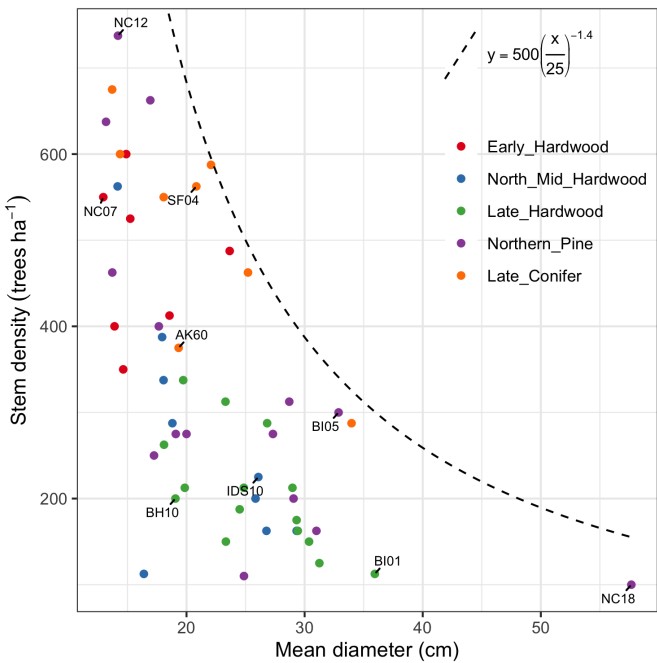

**Figure 2.** Stand structure and composition characteristics of sites selected for analysis. The dashed line is a forest self-thinning curve (c.f., Zeide, 2010) parameterized based on an analysis of US Forest Service Forest Inventory and Analysis (FIA) data (T. Andrews, *unpublished*). Colors indicate dominant PFT, calculated as the PFT contributing most to site total stem density-weighted diameter at breast height, DBH. Sites shown in in Figure 6 are labeled.

AVIRIS-Classic measures the directional radiance of a surface from 365 to 2500 nm at approximately 10 nm increments. Atmospheric correction routines use this level 1 radiance product to estimate the surface reflectance (technically, hemispherical-directional reflectance factor, HDRF, sensu Schaepman-Strub et al. 2006)—a quantity that can be more directly related to intrinsic physical properties of the surface. For this study, in addition to the standard atmospheric correction and orthorectification conducted by NASA Jet Propulsion Laboratory (JPL), the AVIRIS data were also cross-track illumination corrected and bidirectional reflectance distribution function (BRDF) corrected, following the procedure of Lucht et al. (2000). Briefly, this BRDF correction estimates "intrinsic surface albedo"—the quantity that is simulated by EDR—from directional reflectance data by fitting a polynomial approximation to the Ross-Li semiempirical BRDF model and then integrating this model over all angles. The full AVIRIS processing pipeline for the AVIRIS data (including the BRDF approximation) we used is described in Singh et al. (2015).

Because of unrealistic values in the shortwave infrared spectral region (>1300 nm) in the AVIRIS observations (likely caused by faulty atmospheric correction), we only used observations from 400 to 1300 nm for model calibration and validation. Following Shiklomanov et al. (2016), we used the relative spectral response functions of AVIRIS-Classic to relate the 1 nm EDR predictions to the 10 nm AVIRIS-Classic measurements.

For each AVIRIS-Classic observation, we retrieved the solar zenith angle directly from the flightline metadata where available, and calculated it based on the local time and position if not. In addition, we retrieved the relative fraction of diffuse vs. direct incident radiation from the hourly MERRA-2 meteorological reanalysis (GMAO 2015) for each observation's location and time (rounded to the nearest hour).

## 2.4   Model calibration

To estimate EDR parameters from AVIRIS observations, we used a Bayesian approach that builds on our previous work at the leaf scale (Shiklomanov et al., 2016). For a parameter vector $\boldsymbol{\Theta}$ and matrix of observations $\mathbf{X}$, the typical form of Bayes' rule is given by:

$$\underbrace{P(\boldsymbol{\Theta}|\mathbf{X})}_{\text{Posterior}} \sim \underbrace{P(\mathbf{X}|\boldsymbol{\Theta})}_{\text{Likelihood}} \underbrace{P(\boldsymbol{\Theta})}_{\text{Prior}} \tag{42}$$

Rather than performing a separate calibration at each site, we performed a single joint calibration across all sites. Therefore, our overall likelihood ($P(\mathbf{X}|\boldsymbol{\Theta})$) was the product of the likelihood at each site ($P(\mathbf{X}_s|\boldsymbol{\Theta})$, for site $s$):

$$P(\mathbf{X}|\boldsymbol{\Theta}) = \prod_s P(\mathbf{X}_s|\boldsymbol{\Theta}) \tag{43}$$

The likelihood at each site $s$ is based on how well EDR predicted albedo ($R_{\text{pred},s}$) matches that site's observed AVIRIS albedo ($\mathbf{X}_s$) given the known forest composition at that site ($\text{comp}_s$) and the current estimate of the overall parameter vector.

Similar to Shiklomanov et al. (2016), we assumed the residual error between predicted and observed reflectance followed a multivariate normal distribution (MvNormal):

$$P(\mathbf{X}_s|\mathbf{\Theta}) = \mathrm{MvNormal}(\mathbf{X}_s|R_{\mathrm{pred},s}, \mathbf{\Sigma_s}) \tag{44}$$

where $\Sigma$ is the residual variance-covariance matrix. Shiklomanov et al. (2016) assumed $\Sigma$ was a diagonal matrix with the same residual variance for all elements. For this study, we made two important changes to this methodology: First, to account
for the large differences in the range of feasible reflectance values in different wavelength regions (for vegetation, reflectance in the 400–700 nm range is typically much lower than in the 700–1400 nm range), we used a heteroskedastic error model where the residual standard deviation ($\boldsymbol{\sigma}_s$) was a linear function of the predicted reflectance $R_{(\mathrm{pred},s)}$ with slope $m$ and intercept $b$. Second, to account for autocorrelation in hyperspectral bands, we replaced the diagonal residual covariance matrix with an order-1 autoregressive (AR-1) covariance matrix. Collectively, these two changes produce the following calculation for $\Sigma$:

$$\boldsymbol{\sigma_s} = mR_{\mathrm{pred},s} + b \tag{45}$$
$$\mathbf{\Sigma_s} = \boldsymbol{\sigma_s}\varrho^{\mathbf{H}}\boldsymbol{\sigma_s} \tag{46}$$

where $\mathbf{H}$ is a matrix describing the distance between bands (0 on the diagonal, increasing regularly toward the corners) and $\varrho$ is the AR-1 autocorrelation parameter. To simplify the inversion procedure, we first performed the inversion using a diagonal covariance matrix (i.e., $\varrho = 0$), then calculated the mean $\varrho$ from the residuals of this fit, and then used this average value (0.700)
for our final inversion.

In addition, to mitigate sampling issues related to EDR's saturating response to increasing total LAI (Figure A1), we added an additional term to our likelihood that assigns a uniform probability distribution over the range 0 to 10 to the EDR predicted LAI for a given site ($\mathrm{LAI}_{\mathrm{pred},s}$). In practice, this term causes any parameters resulting in total LAI greater than 10 to be immediately rejected, but has no effect on parameters with LAI values less than 10. The maximum value of 10 was selected
as a reasonable upper bound on temperate deciduous and evergreen forests in our study region. By comparison, the *global* maximum of MODIS LAI estimates is between 6 and 7, depending on collection (with most values less than 5; Fang et al. 2012; Yan et al. 2016). A global database of field LAI measurements (Iio et al., 2014) contains values as high as 23.5 for evergreen conifer trees and 12.1 for deciduous broadleaf trees, but these are extreme values, and our maximum of 10 is at least 3 standard deviations away from the mean value for evergreen conifer and deciduous broadleaf trees.
The final expression for the site-specific likelihood is therefore:

$$R_{\mathrm{pred},s}, \mathrm{LAI}_{\mathrm{pred},s} = \mathrm{EDR}(\mathbf{\Theta}|\mathrm{comp}_s) \tag{47}$$
$$P(\mathbf{X}_s|\mathbf{\Theta}) = \mathrm{MvNormal}(\mathbf{X}_s|R_{\mathrm{pred},s}, \mathbf{\Sigma})\ \mathrm{Uniform}(\mathrm{LAI}_{\mathrm{pred},s}|0, 10) \tag{48}$$

Therefore, our parameter vector $\mathbf{\Theta}$ consists of the following (summarized in Table 1): 10 EDR parameters per PFT—5 parameters for the PROSPECT 5 model ($N$, *Cab*, *Car*, *Cw*, *Cm*) and 5 EDR parameters related to canopy structure ($q$, $\chi$, SLA,

**Table 1.** EDR parameters and prior distributions

| Type | Name | Description | Unit | Prior |
|------|------|-------------|------|-------|
| Leaf RTM parameters | $N$ | Effective number of leaf mesophyll layers | unitless | $\text{MvNormal}(\boldsymbol{\mu}, \boldsymbol{\Sigma})$[1] |
| (1 per PFT) | $Cab$ | Total leaf chlorophyll content | $\mu\text{g cm}^{-2}$ | $\text{MvNormal}(\boldsymbol{\mu}, \boldsymbol{\Sigma})$[1] |
| | $Car$ | Total leaf carotenoid content | $\mu\text{g cm}^{-2}$ | $\text{MvNormal}(\boldsymbol{\mu}, \boldsymbol{\Sigma})$[1] |
| | $Cw$ | Leaf water content | $\text{g cm}^{-2}$ | $\text{MvNormal}(\boldsymbol{\mu}, \boldsymbol{\Sigma})$[1] |
| | $Cm$ | Leaf dry matter content | $\text{g cm}^{-2}$ | $\text{MvNormal}(\boldsymbol{\mu}, \boldsymbol{\Sigma})$[1] |
| Canopy RTM parameters | SLA | Specific leaf area | $\text{kg m}^{-2}$ | $\text{MvNormal}(\boldsymbol{\mu}, \boldsymbol{\Sigma})$[1] |
| (1 per PFT) | $q$ | Canopy clumping factor | unitless | $\text{Uniform}(0,1)$ |
| | $\chi$ | Leaf orientation factor | unitless | $2 \times \text{Beta}(18,12) - 1$ |
| | b1Bl | Leaf biomass allometry base | unitless | $\text{LogNormal}(m_l, s_l)$[2] |
| | b1Bw | Wood biomass allometry base | unitless | $\text{LogNormal}(m_w, s_w)$[2] |
| Other parameters | $\psi_s$ | Relative soil moisture content at site $s$ | unitless | $\text{Uniform}(0,1)$ |
| | $m$ | Residual slope | unitless | $\text{Exponential}(1)$ |
| | $b$ | Residual intercept | unitless | $\text{Exponential}(10)$ |

[1] PFT-specific multivariate normal distribution fit to PROSPECT parameters and SLA from Shiklomanov (2018), chapter 3.

[2] PFT-specific results from Bayesian fits of allometric equations to allometry data from Jenkins et al. (2003, 2004) using the `PEcAn.allometry` package.

b1Bl, b1Bw— 1 parameter per site (relative soil moisture, $\psi_s$), and the residual slope ($m$) and intercept ($b$). With 5 PFTs and 54 sites, this means that $\boldsymbol{\Theta}$ has length $(10 \times 5) + 54 + 2 = 106$.

For priors on the PROSPECT 5 parameters and SLA, we performed a hierarchical multivariate analysis (Shiklomanov et al., 2020b) on PROSPECT 5 parameters and direct SLA measurements from (Shiklomanov, 2018, Chapter 3). For priors on the leaf biomass allometry parameters, we fit a multivariate normal distribution to allometry coefficients from Jenkins et al. (2003,

2004) using the `PEcAn.allometry` package (https://github.com/pecanproject/pecan/tree/develop/modules/allometry). For the clumping factor, we used a uniform prior across its full range (0 to 1), and for the leaf orientation factor, we used a weakly informative beta distribution re-scaled to the range $(-1, 1)$ and centered on 0.5 (Table 1).

To alleviate issues with strong collinearity between the allometry parameters and the specific leaf area, we fixed the allometry exponent parameters (b2Bl and b2Bw) to their prior means for each PFT. Doing so dramatically improved the stability of the

inversion algorithm and the accuracy of the results.

We fit this model using the Differential Evolution with Snooker Update ("DEzs") Markov-Chain Monte Carlo (MCMC) sampling algorithm (ter Braak and Vrugt, 2008) as implemented in the R package `BayesianTools` (Hartig et al., 2019). We ran the algorithm using 3 independent chains for as many iterations as required to achieve convergence, assessed according to a Gelman-Rubin Potential Scale Reduction Factor (PSRF) diagnostic value of less than 1.1 for all parameters (Gelman and

Rubin, 1992).

## 2.5 Analysis

To assess the extent to which AVIRIS-Classic observations were able to constrain parameter estimates, we compared the prior and posterior distributions for all parameters. To evaluate the performance of the calibrated model, we compared the posterior credible and predicted 95% intervals of EDR-predicted spectra against the AVIRIS observations at each site. We examined the residuals between EDR predicted and AVIRIS observed reflectance across all sites pooled together, and evaluated whether residuals varied systematically with site composition or structure by separating sites based on the dominant PFT (calculated as the PFT with the largest $\sum_i \mathrm{DBH}_i n_{\mathrm{plant},i}$ at each site), mean DBH, or mean stem density. We also compared the EDR-predicted LAI against field observations at each site, both across all sites together and within the above site groups based on composition and structure. To evaluate goodness-of-fit and additive and multiplicative biases, we used an ordinary least squares regression of mean observed vs. posterior mean predicted LAI.

## 3 Results

Model calibration improved the precision of most PFT-specific parameter estimates, including estimates of leaf parameters whose prior distributions were already independently constrained by an earlier analysis (Figure 3). Across all PFT-specific parameters, the posterior 95% credible interval (CI) was, on average, 10% the size of the prior credible interval. The most constrained parameters on average were EDR canopy structure parameters—namely the wood biomass allometry (<1% of prior CI), leaf biomass allometry (1%), leaf orientation factor (8%), and clumping factor (9%)— while the least constrained parameters were those related to leaf morphology and biochemistry—namely, effective number of leaf layers (19%), total chlorphyll content (16%), total carotenoid content (15%), specific leaf area (13%), dry matter content (11%), and leaf water content (11%). By PFT, the largest average relative constraint was for early hardwood (7%) and the smallest relative constraint was for late hardwood (14% of prior CI).

For leaf traits, PFT rankings of the posterior estimates largely followed the relative positions of the priors, though there were a few exceptions. In both the prior and posterior, the estimated effective number of leaf mesophyll layers (a.k.a., PROSPECT *N* parameter) was higher for needleleaved than broadleaved PFTs, with the highest value for northern pine and the lowest value for mid hardwood. Similarly, specific leaf area (SLA) was lower in conifer than broadleaf PFTs, with the lowest value for late conifer, a higher value for northern pine (despite a similar prior), and higher values still in mid hardwood and late hardwood and the highest value for early hardwood. Estimated total chlorophyll contents (*Cab*) were similarly high for all hardwood PFTs in both the prior and posterior, but posterior estimates for late conifer and northern pine were lower. Posterior estimates of total carotenoid contents (*Car*) were lower in early and mid hardwood and northern pine and higher in mid hardwood and late conifer. Posterior estimates of leaf water content (*Cw*) were low for early hardwood and northern pine and high for mid- and late hardwood and late conifer; these differences were despite strongly overlapping priors across all PFTs. Posterior estimates of leaf dry matter content (*Cm*) were lowest for mid- and late hardwood, higher for early hardwood and northern pine, and highest for late conifer, again despite a strongly overlapping prior across all PFTs.

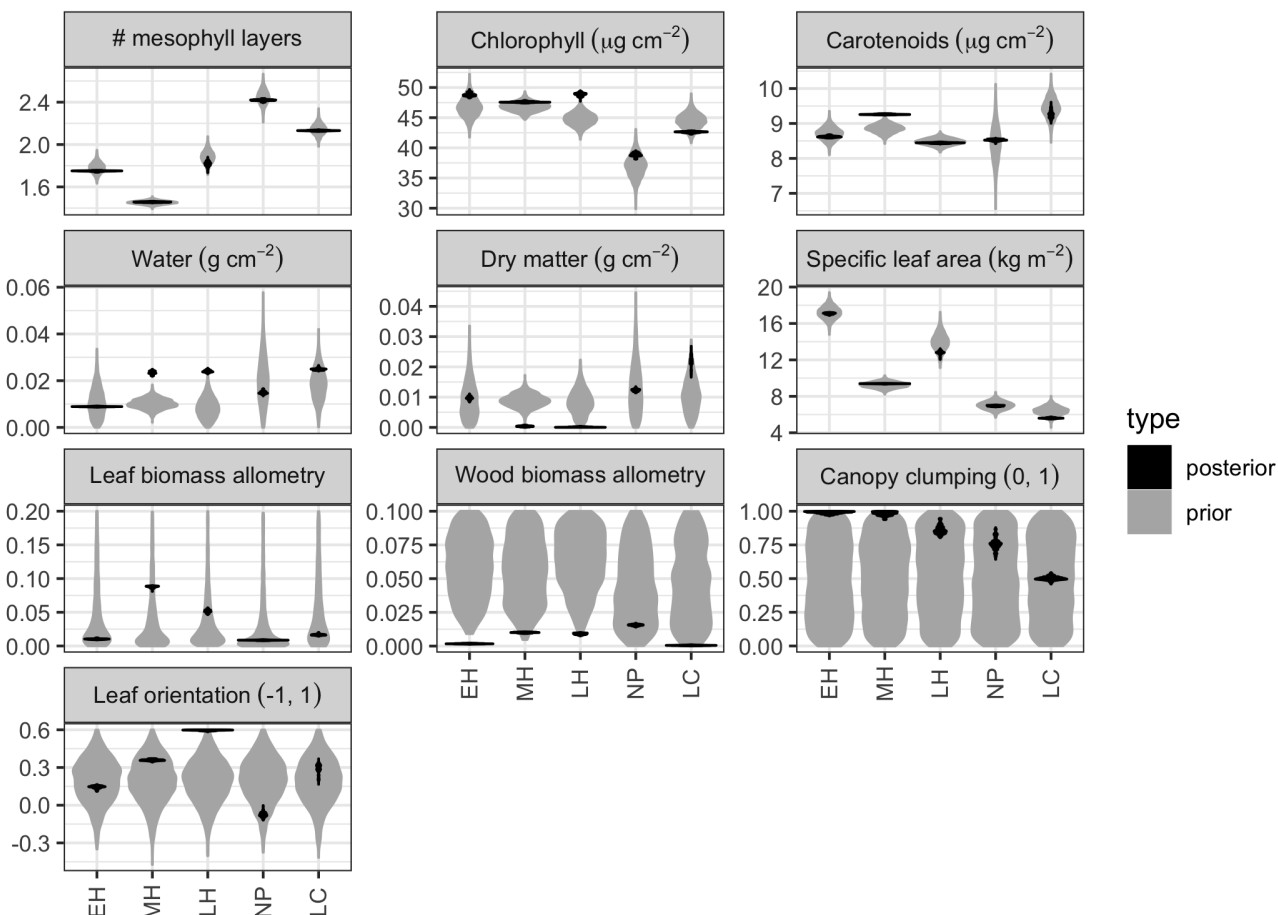

**Figure 3.** Marginal prior (pre-calibration; grey) and posterior (post-calibration; black) distributions of PFT-specific parameters related to leaf biochemistry and canopy structure. Distributions are shown as violin plots (rotated and mirrored kernel density plots). PFTs are abbreviated as follows: EH:Early Hardwood; MH:North Mid Hardwood; LH:Late Hardwood; NP:Northern Pine; LC:Late conifer. Leaf and wood biomass allometry panels are clipped at 0.2 to facilitate differentiation of posterior distributions.

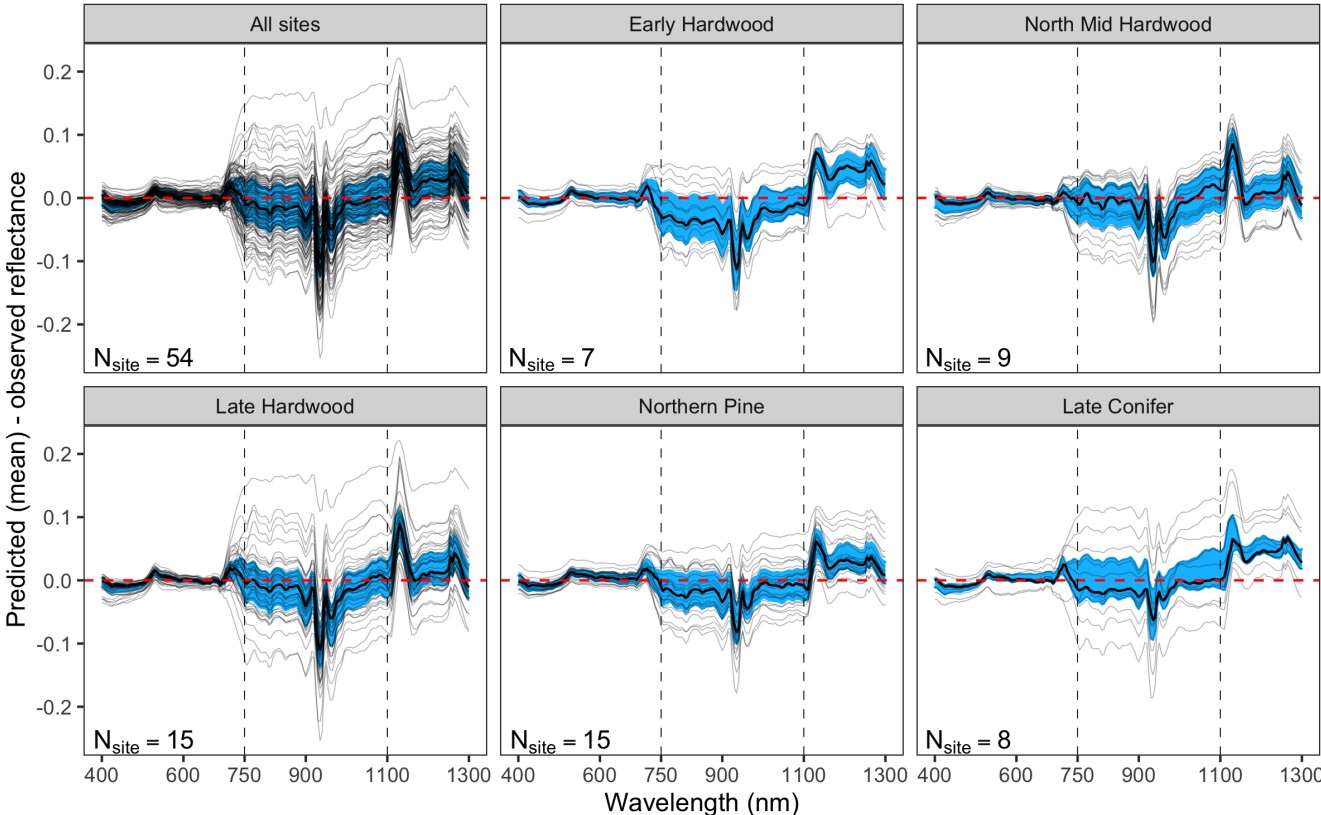

**Figure 4.** Differences between AVIRIS observed and EDR posterior predictive mean surface reflectance by site. Each thin gray line is a site-specific AVIRIS observation. Top left panel shows results across all sites, and remaining panels group sites according to dominant PFT. Within each panel, blue shading shows the 25–75% quantile range and the thick black line is the median by wavelength for that specific site.

Compared to leaf traits, canopy structural traits had less informative (and PFT-agnostic) priors, and the posterior distributions exhibited some differences across PFTs. Posterior leaf biomass allometry ($b1Bl$) estimates were lowest in early hardwood and
northern pine, higher for late conifer, and highest for late- and mid hardwood. Posterior wood biomass allometry ($b1Bw$) estimates were lowest for early hardwood and late conifer, slightly higher in mid- and late hardwood, and highest for northern pine. Posterior canopy clumping factor ($q$) estimates were clustered at or near its upper limit of 1 (i.e., exhibited "edge-hitting behavior") for early and mid hardwood, were slightly lower in late hardwood and northern pine, and lowest in late conifer. Posterior leaf orientation factor ($\chi$) estimates were lowest (near zero, indicating randomly distributed leaf angles) for northern
pine, higher (more horizontal leaves) for early and mid hardwoods and late conifer, and highest (at the upper limit of 0.6) for late hardwood. Finally, the calibration was able to constrain site-specific soil optical properties across all sites (Figure A4).

The accuracy and precision of EDR simulated spectra relative to AVIRIS observations varied across sites (Figures 4, 5, 6, and A5). On average, EDR tended to accurately (within 0.01) reproduce reflectance in the 400–750 nm range, underpredict

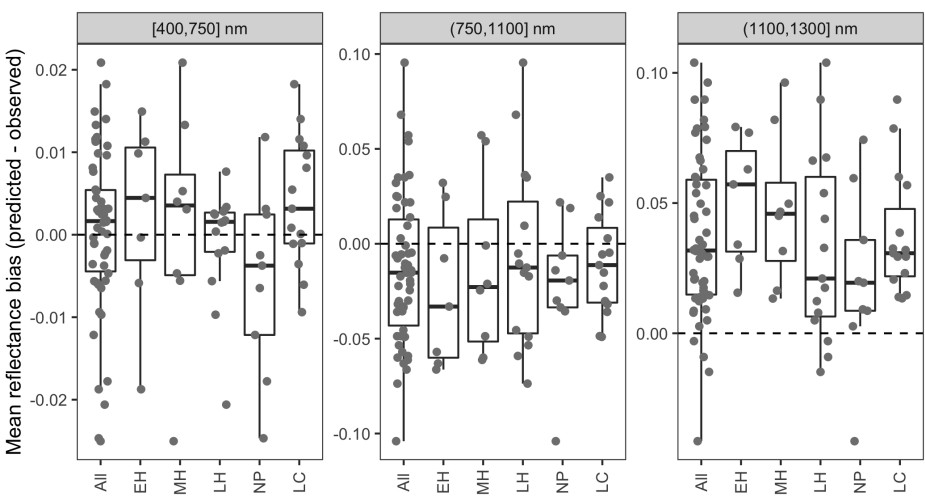

**Figure 5.** Differences between AVIRIS observed and EDR posterior predictive mean surface reflectance by site, averaged across wavelength regions. Sites are grouped by dominant PFT, as in Figure 4. Note the differences in the $y$ axis scale across panels.

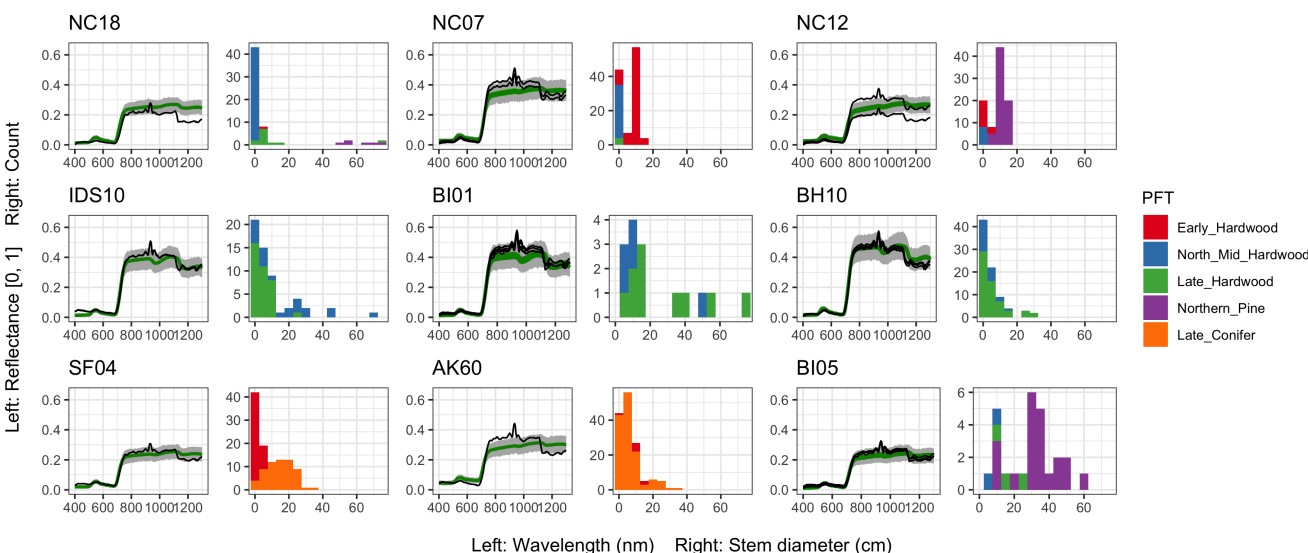

**Figure 6.** (*Left*) Comparison between AVIRIS observed (black) and EDR predicted (mean prediction in green, 95% posterior predictive interval in gray) surface reflectance for a geographically (Figure 1), compositionally, and structurally (Figure 2) representative sample of sites used in the calibration. (*Right*) Histogram of stem diameter at breast height (DBH) by plant functional type (PFT) at the corresponding site.

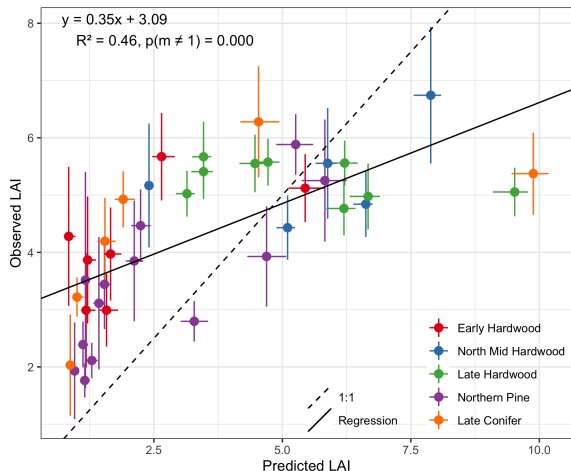

**Figure 7.** EDR predictions of site-specific true leaf area index (LAI) compared to observed values. Horizontal error bars are posterior 95% predictive intervals. Vertical error bars are mean $\pm$ 1 standard deviation of the observed values. Dashed line shows the 1:1 relationship, and solid line is a least-squares predicted vs. observed regression with the equation marked in the upper left corner. Points are colored according to dominant PFT, calculated as in Figure 4.

AVIRIS reflectance by $\tilde{0}.02$ in the 750–1100 nm range, and overpredict AVIRIS reflectance by $\tilde{0}.03$ in the 1100–1300 nm

range. However, only the latter behavior was consistent across sites; below 1100 nm, sites dominated by any PFT could have low, accurate, or high estimates relative to the AVIRIS observations. The only consistent bias (expressed as interquartile range in bias not overlapping zero) we observed with respect to PFT was an underestimate of reflectance in the 750–1100 nm range for northern pine; otherwise, we did not observe any consistent patterns in mismatch between AVIRIS observed and EDR predicted reflectance with respect to tree size, stem density, or composition (Figures 5 and A6–A15). The EDR posterior

predictive interval overlapped AVIRIS observations in all but a few individual cases (Figure A5), suggesting that our estimates of model uncertainty are realistic.

Leaf area index predicted from calibrated EDR parameters captured 46% of the variability in the observations (Figure 7). The observed vs. predicted line had a slope of 0.35 and an intercept of 3.09, indicating that EDR calibration underpredicted true LAI on average but exaggerated LAI variability across sites. In general, EDR tended to underpredict LAI at high-density

sites with low mean DBH and overpredict LAI at low-density sites with high mean DBH (Figures 8 and 9). The trend with mean DBH was generally true across all PFTs but was most pronounced for early hardwood- and late conifer-dominated sites (Figure 8), while the trend with stand density was most pronounced for late conifer-dominated sites (Figure 9).

For identical canopies, EDR consistently predicted lower hemispherical reflectance than PRO4SAIL (Figure 10). This difference was most pronounced when the sun was directly overhead ($\theta_s = 0°$; $\cos(\theta_s) = 1$) and declined with increasing solar

zenith angle. For solar zenith angles typical of our study, ($\theta_s \approx 30°$; $\cos(\theta_s) = 0.85$) EDR hemispherical reflectance predictions were very close to PRO4SAIL directional reflectance predictions over a wide range of LAI values (Figure A16).

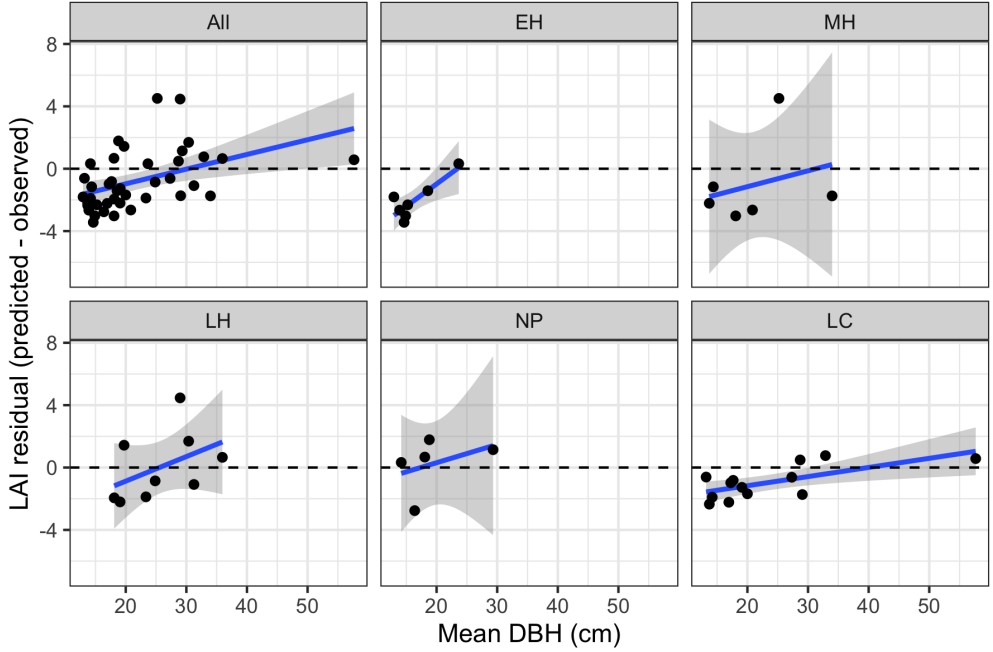

**Figure 8.** Bias in leaf area index (LAI) predictions from calibrated EDR relative to observations, as a function of site mean diameter at breast height (DBH). Sites are grouped according to dominant PFT, same as in Figure 4.

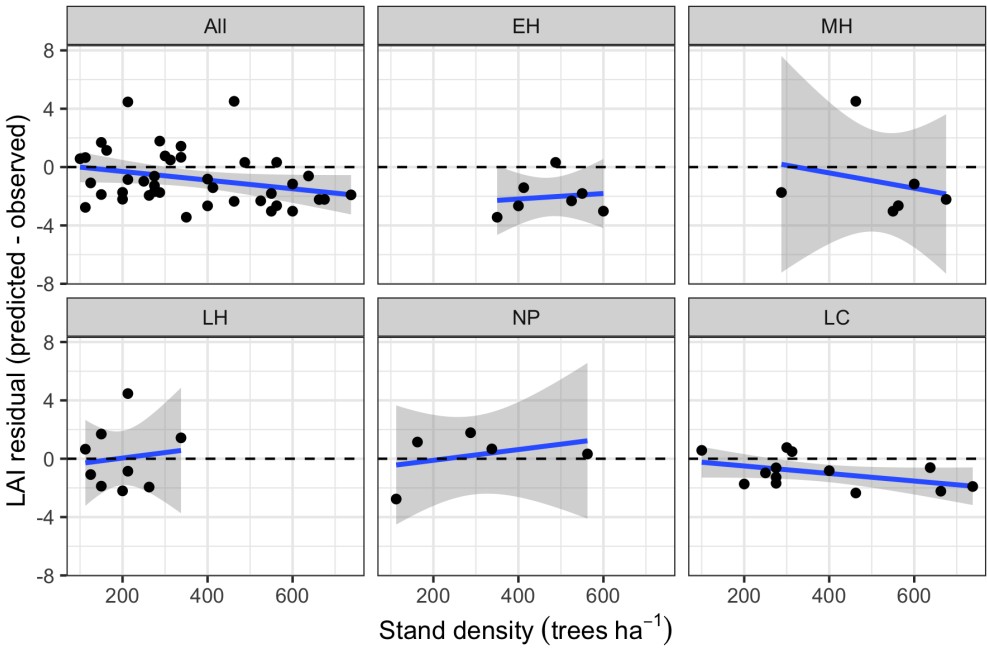

**Figure 9.** Same as above, but as a function of mean site stem density.

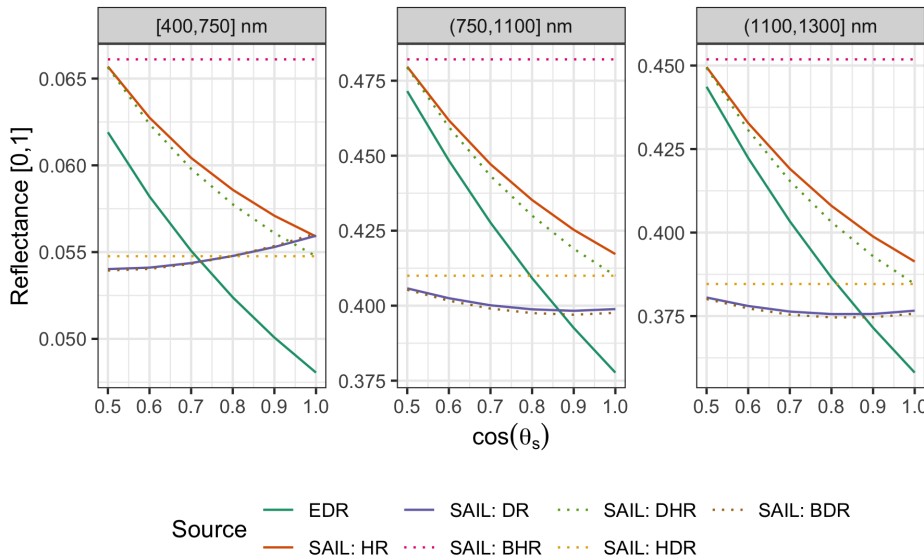

**Figure 10.** Comparison of EDR and PRO4SAIL (labelled as "SAIL" for conciseness) predictions of reflectance for identical, single-cohort canopies as a function of solar zenith angle ($\theta_s$). These simulations use identical PROSPECT and canopy structure parameters, a nadir-viewing sensor, and LAI of 3. For EDR, we use a single cohort with LAI of 3 prescribed directly. "SAIL:HR" is the "hemispherical reflectance" ("blue-sky albedo"), calculated (as in EDR) as the average of directional-hemispherical (DHR) and bi-hemispherical (BHR) reflectance streams weighted by the direct sky fraction (0.9; same value used for EDR). Similarly, "SAIL:DR" is the "directional reflectance", calculated analogously from the bi-directional (BDR) and hemispherical-directional (HDR) reflectance streams. Individual SAIL fluxes are shown with dotted lines.

## 4 Discussion

Calibrating and validating vegetation models using optical remote sensing data has typically involved derived data products (e.g., MODIS GPP) that rely on their own models; in other words, "bringing the observations closer to the models". In this study, we presented an alternative approach whereby we "bring the models closer to the observations" by training a vegetation model to simulate full-range hyperspectral surface reflectance that is closer to the measurements made by optical remote sensing instruments. We argue this is a more generalized approach, as many dynamic vegetation models already contain their own internal representations of canopy radiative transfer and thus can be modified to provide outputs that can mimic optical remote sensing (i.e., can be used as "satellite simulators" to connect underlying model state to emergent reflectance). We then demonstrated how this approach could be used to calibrate the model against airborne imaging spectroscopy data from AVIRIS-Classic. We found that such calibration reduced uncertainties in parameters related to leaf biochemistry and canopy structure, even for parameters with well-informed priors (Figure 3). Moreover, we found that that the calibrated model was able to reproduce observed surface albedo (Figures 4, 5, 6, and A5) reasonably well across large number of geographically

(Figure 1), structurally, and compositionally (Figure 2) diverse sites. However, the calibrated model underpredicted LAI at sites with mostly small trees and overpredicted LAI at sites with mostly large trees (Figures 7 and 8).

In this study, the vegetation composition at each site (including the PFT distribution and size-age structure) was prescribed in detail based on inventory data. This allowed us to focus the calibration on model parameters related canopy radiative transfer model parameters. However, ED2 is a dynamic vegetation model whose core purpose is to predict how vegetation composition and structure evolve through time. An important future direction of this work is to evaluate such dynamic ED2 simulations where vegetation composition and structure are predicted with some uncertainty. In ED2, shortwave canopy radiative transfer is already linked (through shared parameters and state variables) to other important model processes, including thermal radiative transfer, micrometeorology, photosynthesis, respiration, and competition (Longo et al., 2019), and therefore, changes in canopy radiative transfer parameters have profound consequences for ED2 predictions of ecosystem fluxes and composition (Viskari et al., 2019). Future work could further strengthen this link by embedding the PROSPECT coupling demonstrated in this study into ED2 itself, replacing ED2's currently prescribed leaf optical properties with simulated optical properties that change with leaf morphology and biochemistry. For example, the PROSPECT leaf water content parameter ($Cw$) provides a physical link between leaf optical properties and hydraulics, so such a configuration could allow surface reflectance information to constrain ED2's recently developed dynamic hydraulics module (Xu et al., 2021).

With 54 sites in our calibration, any single site represents $< 2\%$ of the data, and for a joint calibration without site random effects, we have every reason to believe that the calibration is not overfitting to any individual site. Trying to fit any one site well would cause others to do worse (especially given the large observed variability in forest structure; Figure 2) unless the EDR model structure was reasonable and the parameters chosen were genuinely good choices. We therefore did not perform an in-sample cross validation, as we believed the benefit of doing so would be low relative to the high computational cost. That said, we recognize that out-of-sample validation is a useful test of model performance, and we recommend performing out-of-sample validation in similar studies in the future.

The canopy radiative transfer model in ED2 is derived from the two-stream model of Sellers (1985) and adapted to a multi-level canopy. Similar versions of this two-stream formulation are present in other land surface models, including CLM (Oleson et al., 2013), SiB (Baker et al., 2008), Noah (Niu et al., 2011), tRIBS-VEGGIE (Ivanov et al., 2008), IMOGEN (Huntingford et al., 2008), and JULES (Best et al., 2011). Although the exact parameterization and implementation differs across these models, the similarity of the underlying conceptual framework and radiative transfer coefficients means that our approach should be directly transferable to all of these models.

Nevertheless, our analysis echoed some known challenges in canopy radiative transfer modeling. One challenge is equifinality in the contributions of leaf biochemistry, leaf morphology, and different aspects of canopy structure to canopy albedo, which means that multiple variable and parameter combinations can produce very similar canopy albedo responses (Lewis and Disney 2007; Figures A1–A3). We mitigated the equifinality between leaf traits and canopy structure by using informative priors on leaf traits from an independent data source (Shiklomanov, 2018). However, there is additional equifinality in the effects of the EDR canopy structure parameters. For example, because the effective LAI used in EDR's actual radiative transfer calculations is defined as the product of "true" LAI and clumping factor (equation 20), and because LAI is, in turn, derived

from multiple parameters (leaf biomass allometry, specific leaf area; equation 21), these parameters collectively cannot be in-
dependently determined from reflectance data alone. At the same time, increasing the leaf orientation factor (more horizontal, or "planophile", leaf orientation) has a similar (although not identical) effect to increasing LAI and clumping factor—namely, increasing canopy reflectance, especially in the near-infrared (Figure A3). Collectively, these issues may help explain some of the edge-hitting behavior (parameter distributions clustered at the ends of the distribution) observed in our posterior estimates (Figure 3), and some of the bias in our LAI estimates (Figure 7). In future work, we suggest combining our approach with ad-
ditional kinds of remote sensing measurements capable of directly constraining these structural parameters such as waveform LiDAR (which can provide a robust constraint on the canopy structural profile; Ferraz et al. 2020) to reduce equifinality.

That being said, one major advantage of the Bayesian calibration approach is that its output is a joint posterior distribution that includes not only fully quantified uncertainties for each parameter but also the variance-covariance matrix across the full set of parameters. Equifinality in parameters would manifest as strong pairwise correlation between parameters in the posterior distribution. Examining this correlation matrix shows that there are some parameter pairs with strong correlations, such as the hypothesized negative correlations between leaf allometry and clumping factor across some PFTs (Figure A17). However, these correlations do not occur in all parameters that exhibited edge-hitting behavior. For instance, clumping factor exhibited edge-hitting behavior only for early- and mid-successional hardwood PFTs (Figure 3), but the corresponding correlation coefficients were *positive* and near zero, respectively, while strong negative correlations for the other PFTs did not result in edge-hitting posteriors (Figure A17). Similarly, the edge-hitting leaf orientation factor posterior for late hardwood (Figure 3) had near-zero (or, in one case, positive) correlations with all other parameters (Figure A17). Strong correlations also occurred among some of the PROSPECT parameters, and between PROSPECT and structural parameters, but contributed little to equifinality because the strong constraints on PROSPECT led to overall small covariance terms (results not shown). Finally, because our calibration captured all of these covariances, the presence of moderate equifinality did not preclude ecologically meaningful parameter constraints or accurate predictions because these covariances are being propagated into predictions. This is directly analogous to how a linear regression can have a tight confidence interval, despite high correlations between the slope and intercept, with that equifinality driving the characteristic hourglass shape of a regression confidence interval.

EDR tended to underpredict LAI at high-density sites with low mean DBH and overpredict LAI at low-density sites with high mean DBH (Figures 7, 8, and 9). The relationship between DBH and LAI is controlled primarily by the leaf biomass allometry, which in EDR is fixed at the PFT level (equation 22). This fixed relationship neglects the known inter- and intra-specific variability in tree allocation strategies (Forrester et al., 2017; Dolezal et al., 2020). For example, Forrester et al. (2017) show that the relationship between DBH and foliar biomass is modulated by tree age, stand density, and climate variables, none of which are accounted for in the ED2 allometry routines. This variability can be incorporated directly into ED2 by making allometry parameters dynamic functions of some of the aforementioned covariates, or indirectly via hierarchical calibration whereby model parameters vary across sites (Dietze et al., 2008). Overall, this analysis reiterates the importance of evaluating models against multiple distinct variables—after all, none of these biases would have been apparent from looking at the reflectance simulations alone.

Our analysis also revealed some structural issues with EDR itself. EDR consistently predicted lower hemispherical reflectance than SAIL (Figures 10 and A16). This difference can be attributed primarily to differences in how each model defines the direct radiation backscatter coefficient in the radiative transfer equation. A detailed description of the discrepancy is provided in Yuan et al. (2017). Briefly, EDR (and the Sellers 1985 model from which EDR is derived) defines direct radiation backscatter as a function of the single-scattering albedo (equation 24), which in turn is a challenging integral involving the leaf scattering phase function and leaf projected area function (equation 25). The analytical solution to this integral in EDR (equation 28) assumes a uniform scattering phase function, which is appropriate for point scatterers but less so for horizontal surfaces like leaves. The practical consequence of this assumption is a lower value of the direct radiation backscatter and therefore a lower albedo, which is consistent with the results of our sensitivity analysis. This underestimation of albedo described above may also help explain the edge-hitting behavior in our posterior distributions (Figure 3) as well as the relatively low accuracy of our LAI estimates (Figure 7). Specifically, our EDR calibration may be trying to compensate for underestimated albedos via a tendency to prefer higher effective LAI values (which results in higher values of the leaf biomass allometry and clumping factor for some PFTs; e.g., early and mid hardwood and northern pine in Figure 3) and more horizontal leaf distributions (i.e., higher leaf orientation factor; e.g., late hardwood in Figure 3), both of which increase albedo (Figures A1–A3).

Meanwhile, the SAIL definition of direct backscatter is a more simple function of leaf scattering, leaf angle distribution, and canopy optical depth that also produces a more accurate albedo estimate (Yuan et al., 2017). Given that underestimating albedo can have significant consequences for the biological, ecological, and physical predictions of ED2 (Viskari et al., 2019), incorporating this fix into the ED2 canopy RTM is an important future direction of our work. However, doing so is outside the scope of this work because it would require propagating the different coefficients through the complex, multiple-canopy-layer solution of EDR (Section 2.1.2)—a non-trivial task.

A significant caveat to the broader application of our approach is that there is a subtle but significant difference between the physical quantity EDR predicts and the quantities typically measured by optical remote sensing. Specifically, EDR predicts the hemispherical reflectance—the ratio of total radiation leaving the surface to the total radiation incident upon the surface, integrated over all viewing angles (a.k.a, "blue-sky albedo"). On the other hand, optical remote sensing platforms typically measure the directional reflectance factor—the ratio of actual radiation reflected by a surface to the radiation reflected from an ideal diffuse scattering source (e.g., a Spectralon calibration panel) subject to the same illumination, in a specific viewing direction (Schaepman-Strub et al., 2006). These two quantities are numerically equivalent only for ideal Lambertian surfaces or, for non-Lambertian surfaces, under specific sun-sensor geometries. However, vegetation canopies—the focus of this study—are known to exhibit reflectance with very strong angular dependence, so a comparison of canopy hemispherical reflectance with a remotely sensed directional reflectance factor is invalid.

Our specific analysis is valid because—as described in the Methods (Section 2.3)—we used AVIRIS data that were also BRDF-corrected, whereby the directional reflectance estimates from the atmospheric correction process were further converted to estimates of hemispherical reflectance via a polynomial approximation of the Ross-Li semiempirical BRDF model (Lucht et al., 2000). Another dataset that would have been valid for our analysis (albeit, one with much lower spatial and spectral resolution) is the MODIS albedo product (MOD43), which takes advantage of the angular sampling of the MODIS instrument

to quantify the surface BRDF characteristics and therefore more precisely estimate the albedo (Wang et al., 2004; Schaaf and Wang, 2015). However, our approach as described here would *not* be valid for the surface reflectance products produced by nadir-viewing instruments such as Landsat, Sentinel-2, or most airborne platforms, at least without additional processing steps on the data, or, preferably, modification of the underlying radiative transfer models to allow for the prediction of directional reflectance. Fortunately, the assumptions and parameters that comprise two-stream radiative transfer models like EDR and its parent model (Sellers, 1985) are readily adaptable to prediction of directional reflectance. For example, the SAIL model (Verhoef, 1984; Verhoef and Bach, 2007)—which predicts both hemispherical and directional reflectance, and which has a long history of successful application to remote sensing—makes the same general assumptions as EDR and even shares many of the underlying coefficients (Yuan et al., 2017). Alternatively, land surface models can take advantage of recent advances in radiative transfer theory to improve their accuracy without significant computational penalty (e.g., Hogan et al., 2018).

A related issue is the missing or simplistic treatment of two- and three-dimensional heterogeneity in canopy structure in EDR. For one, the treatment of leaves as infinitely small elements randomly distributed through the canopy space—a common feature of all two-stream approximations—neglects complex realities of the canopy light environment such as gaps and self-shading. In EDR, self-shading is handled via the clumping factor parameter, which functions as a scalar correction on the leaf area index (Equation 20). A key feature of EDR design is its representation of multiple co-existing plant cohorts competing for light within a single patch; however, horizontal heterogeneity and competition between these cohorts is ignored. Improved representation of lateral energy transfer can improve the accuracy of simulations of the canopy light environment, and recent theoretical advances show that this can be accomplished without a significant loss in computational performance (Hogan et al., 2018). Treatment of horizontal competition also plays an important role in the outcomes of competition for light between different plants (Fisher et al., 2018). A useful avenue for development and parameterization of these models is comparison to more sophisticated and realistic three-dimensional representations of radiative transfer (e.g. Widlowski et al., 2007), which are themselves too computationally demanding to be coupled to ecosystem models, but from which empirical distributions and response functions could be derived and against which the behavior of simpler models could be evaluated.

## 5  Conclusions

Remote sensing observations are unrivaled in their spatial completeness and extent, notably extending to regions like the tropics and high latitudes that are relatively undersampled but have a disproportionate impact on the global climate system (Schimel et al., 2015) and/or global biodiversity (Jetz et al., 2016). At the same time, satellite time series provide multidecadal records with relatively high temporal frequency, which have tremendous utility for calibrating model projections of past ecological dynamics (Kennedy et al., 2014; Pasquarella et al., 2016). Used in combination with other emerging data sources, including global trait databases and eddy covariance measurements, remote sensing can be a transformative force in ecosystem ecology.

In this paper, we showed that using a vegetation model to directly simulate surface reflectance is a promising approach for calibrating and validating models against remotely sensed observations. To do this, we modified the ED2 dynamic vegetation model to predict full-range hyperspectral hemispherical surface reflectance and then calibrated this modified model against

airborne, BRDF-corrected imaging spectroscopy data. The calibration successfully reduced uncertainties in model parameters related to canopy structure and leaf biogeochemistry for five plant functional types for five plant functional types characteristic of temperate forests of the northeastern United States. The calibrated model was able to accurately reproduce observed surface reflectance for sites with highly varied forest composition and structure using a single common set of parameters (i.e., not site-specific parameters). However, the calibrated model predicted leaf area index values that did not agree well with observations and had parameter estimates that exhibited edge-hitting behavior, both of which suggest structural issues in the model. Comparison against a canopy radiative transfer model commonly used in the remote sensing community (PRO4SAIL, Verhoef and Bach, 2007) suggested that our model may be systematically underpredicting surface albedo. Given the direct role albedo plays in the canopy light and thermal environment in ED2, this bias could have significant downstream consequences for ED2 predictions of physiological and ecological processes. We therefore recommend structural changes to the ED2 canopy radiative transfer model to resolve this bias, and recommend calibrating the updated model against remotely sensed surface reflectance, as we demonstrated here. We note that the basic structure and assumptions of the ED2 canopy radiative transfer scheme are shared by many other vegetation models, so we expect that both this issue and our recommendations for resolving it are highly transferable within the vegetation modeling community. More generally, we recommend the development of additional "observation operators" similar to ours for other classes of remote sensing data, such as thermal, microwave, and LiDAR, in ED2 and other dynamic vegetation models to allow these models to take full advantage of remote sensing observations.

*Code and data availability.* All of the code and data required to reproduce this study are publicly available via an Open Science Framework (OSF) project, located at https://osf.io/b6umf/.

**Appendix A: Supplementary figures**

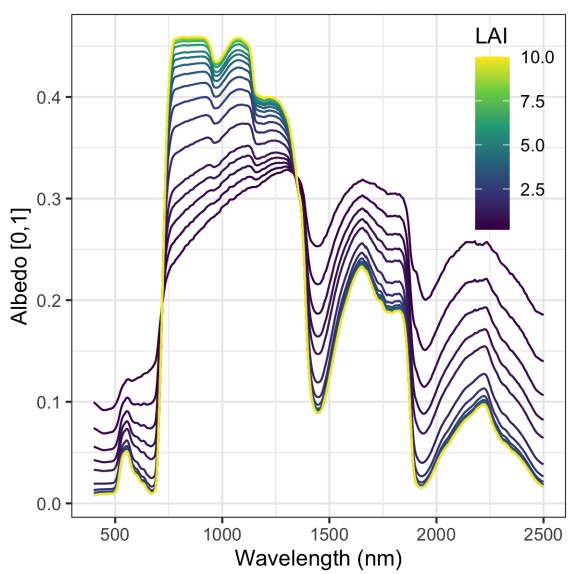

**Figure A1.** Sensitivity of EDR predicted hemispherical reflectance to true leaf area index (LAI). These simulations assume a single-cohort canopy with effective number of mesophyll layers $N = 1.4$, total chlorophyll content $Cab = 40$, total carotenoid content $Car = 10$, leaf water content $Cw = 0.01$, leaf dry matter content $Cm = 0.01$, clumping factor $q = 1$, leaf orientation factor $\chi = 0$, $\cos(\theta_s) = 0.85$, and soil moisture fraction $\psi = 0.5$.

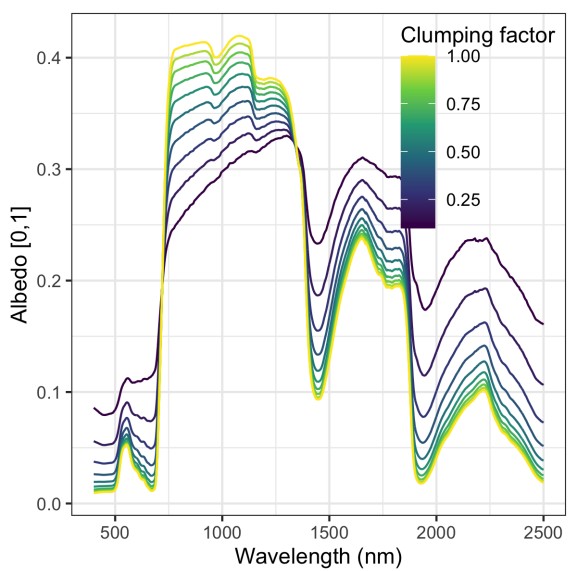

**Figure A2.** Same as above but with LAI fixed to 3 and varying clumping factor ($q$).

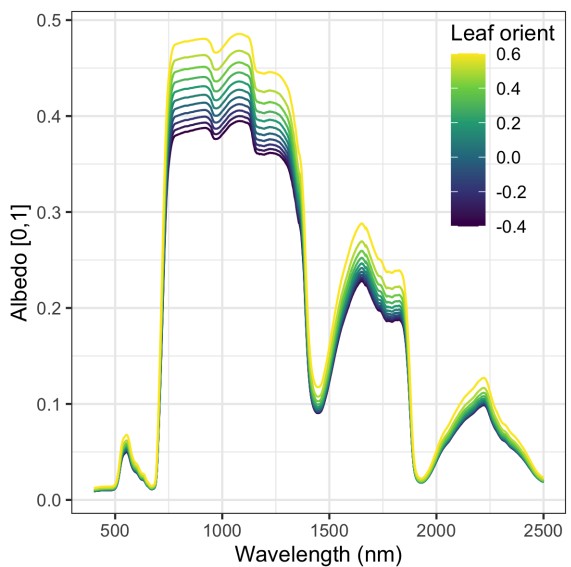

**Figure A3.** Same as above, but instead varying leaf orientation factor ($\chi$).

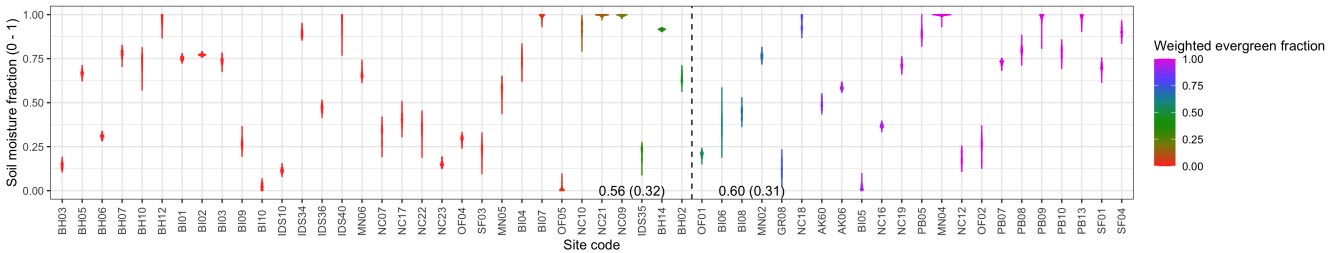

**Figure A4.** Site-specific relative soil moisture (0 = dry, 1 = wet) posterior estimates. Sites are sorted in order of increasing weighted evergreen fraction.

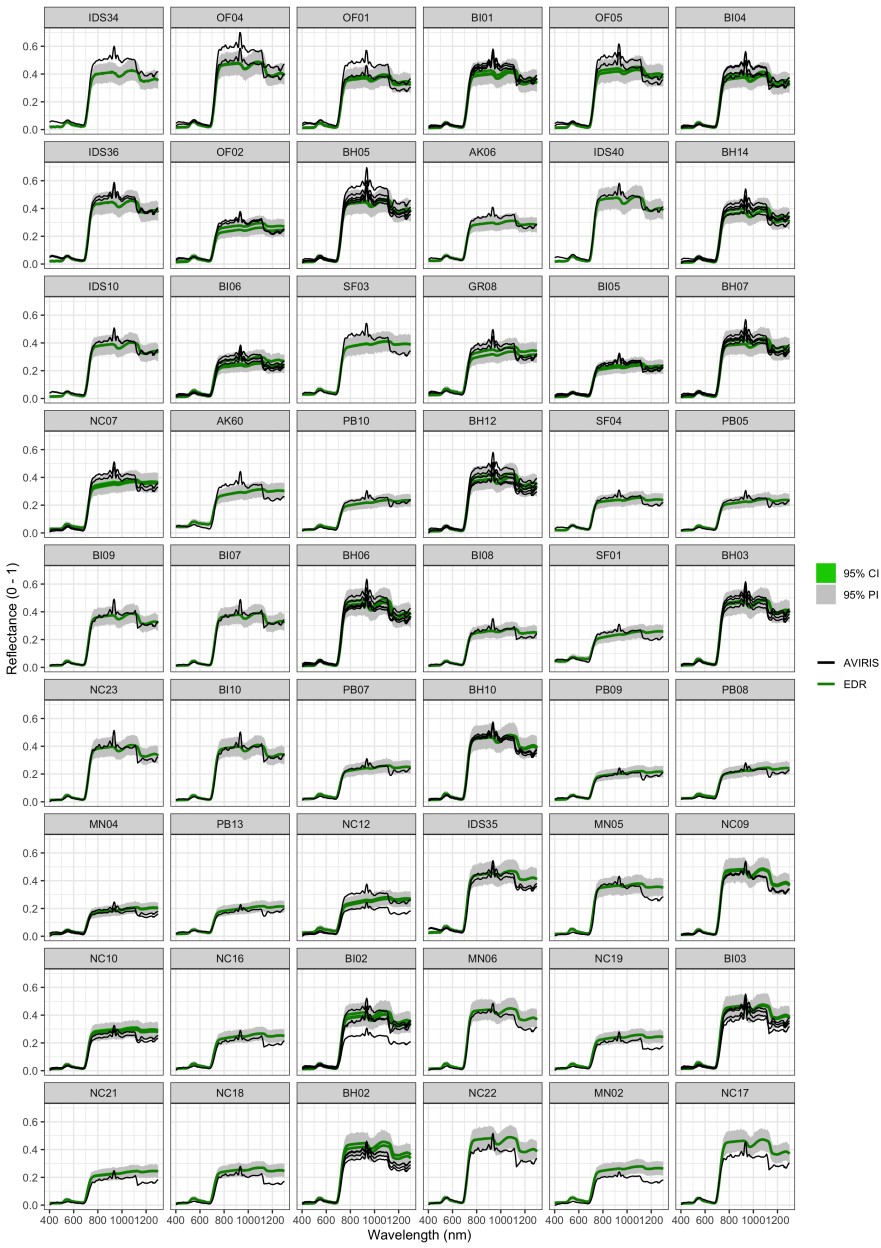

**Figure A5.** Comparison between AVIRIS observed (black) and surface reflectance for each site used in the calibration. Sites are sorted in order of decreasing mean difference between observed and EDR predicted reflectance (largest underestimates first, largest overestimates last).

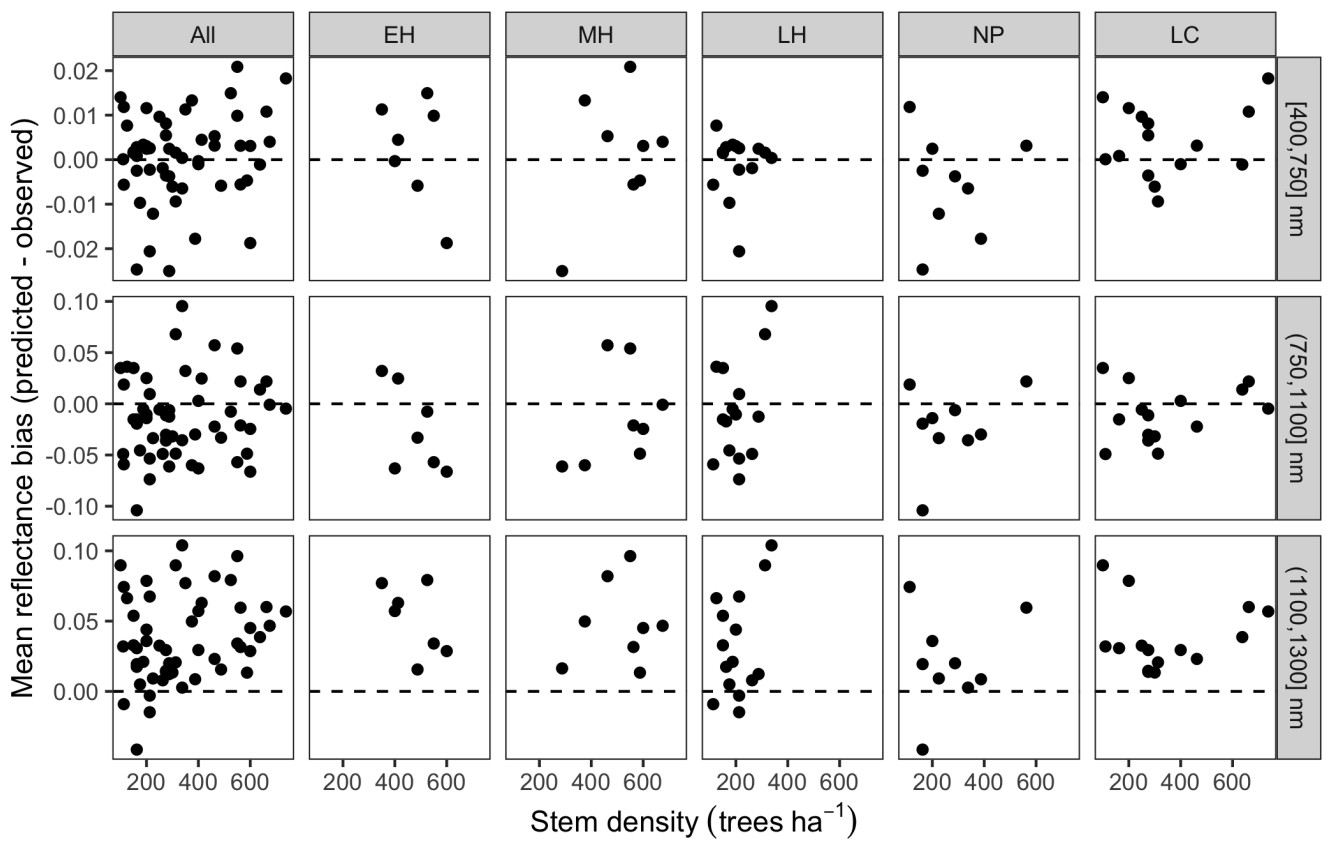

**Figure A6.** Mean reflectance bias (EDR predicted − observed) for each by spectral region and dominant plant functional type (PFT) as a function of site stem density. PFTs are abbreviated as follows: EH:Early Hardwood; MH:North Mid Hardwood; LH:Late Hardwood; NP:Northern Pine; LC:Late conifer

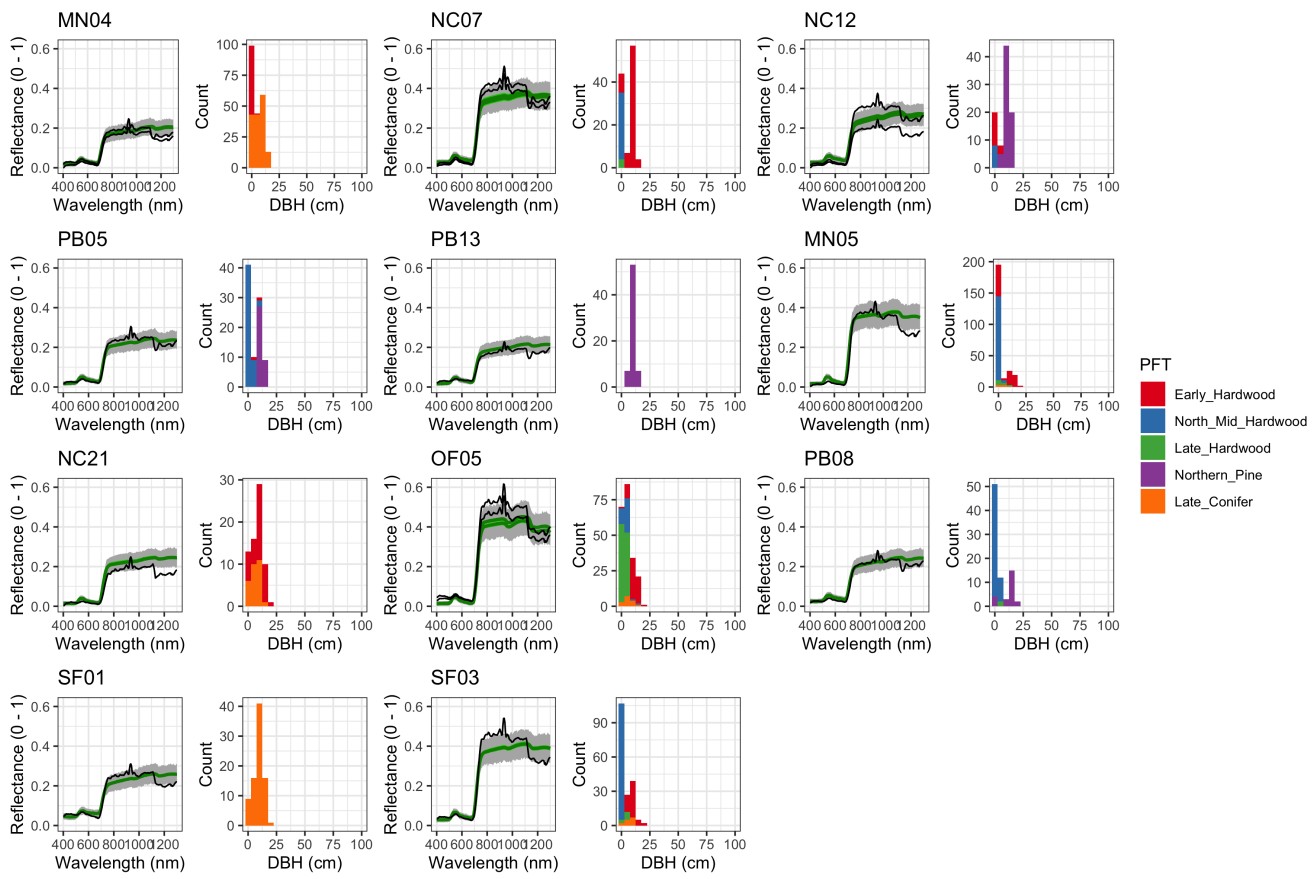

**Figure A7.** EDR predicted vs. observed spectra and species composition for the first quartile of sites by DBH.

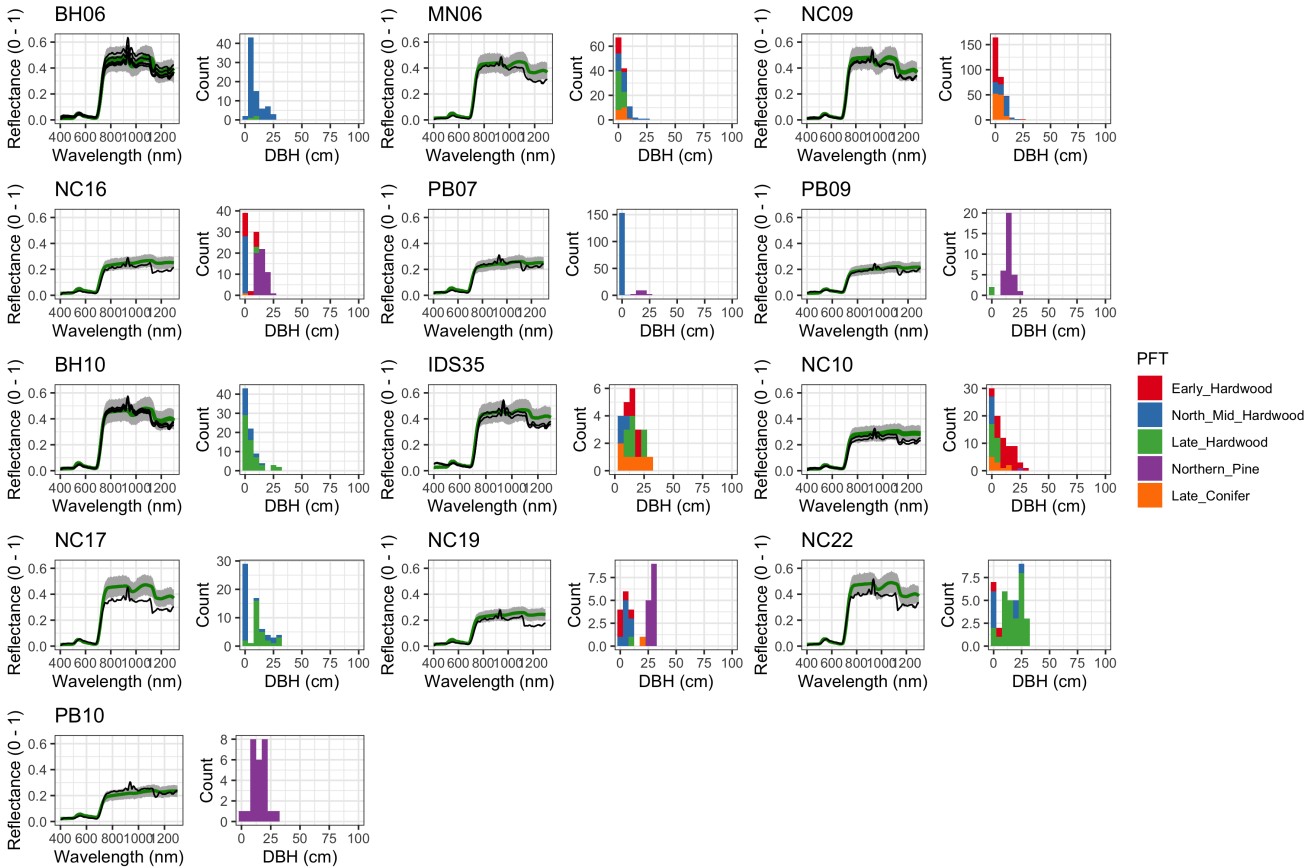

**Figure A8.** As above, but for the second quartile of sites by DBH.

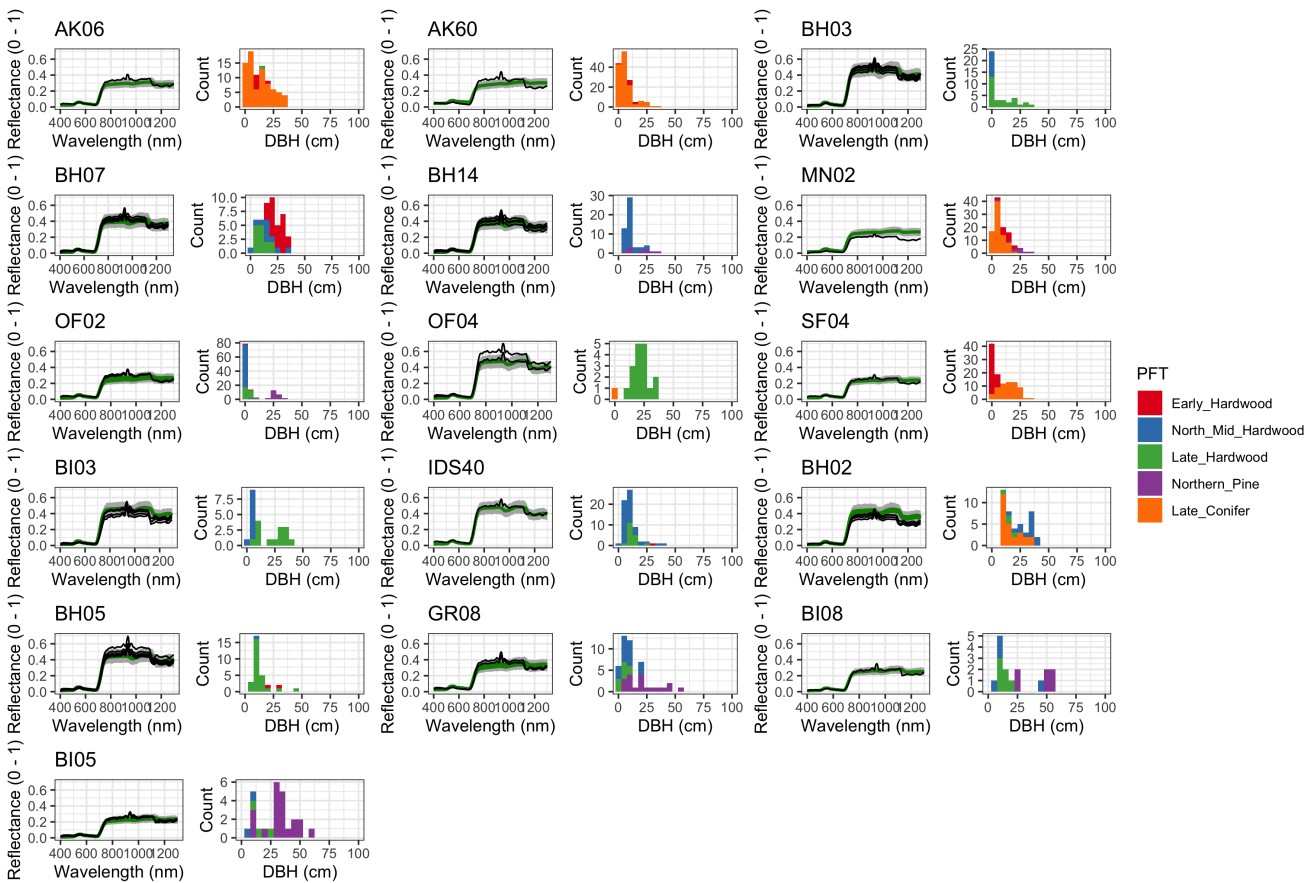

**Figure A9.** As above, but for the third quartile of sites by DBH.

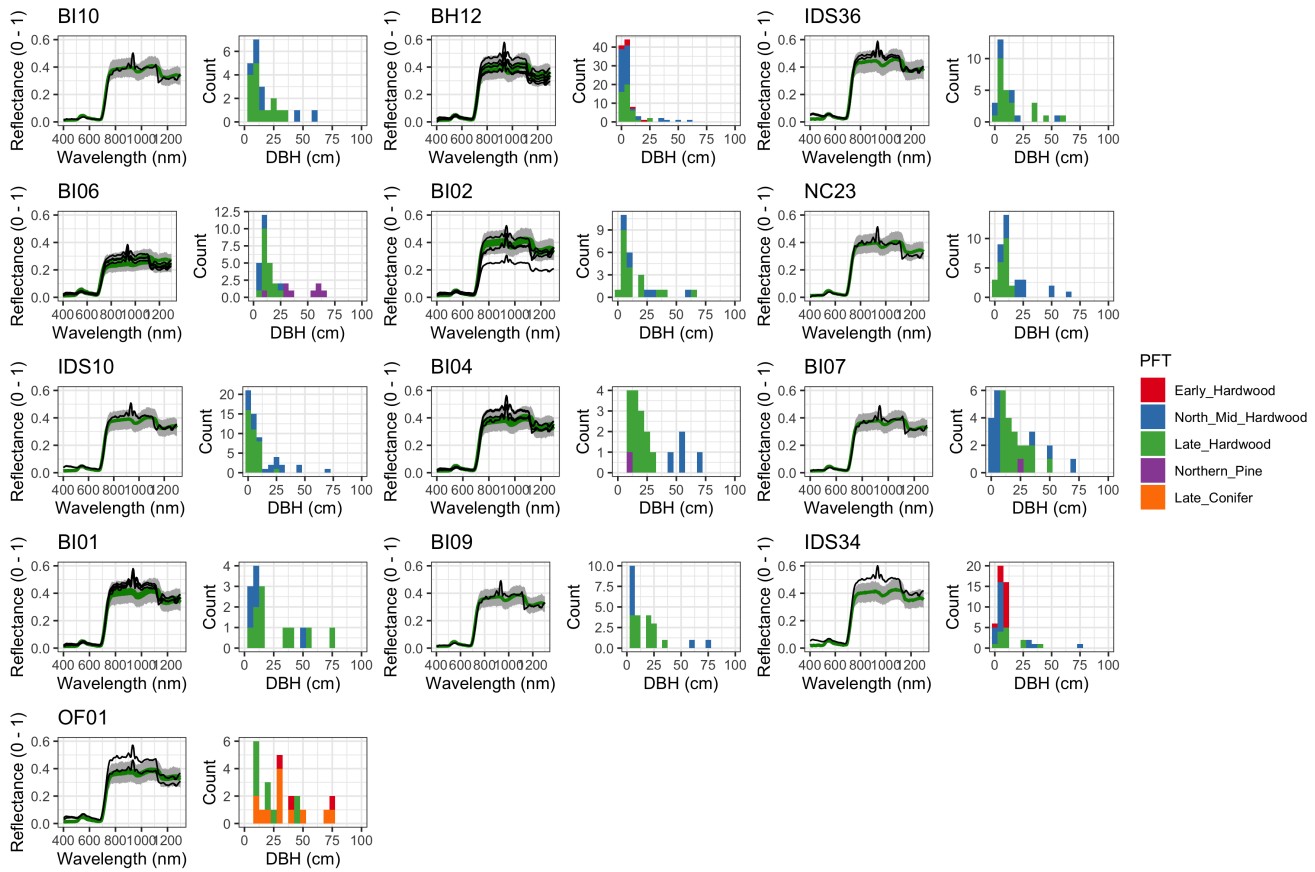

**Figure A10.** As above, but for the fourth quartile of sites by DBH.

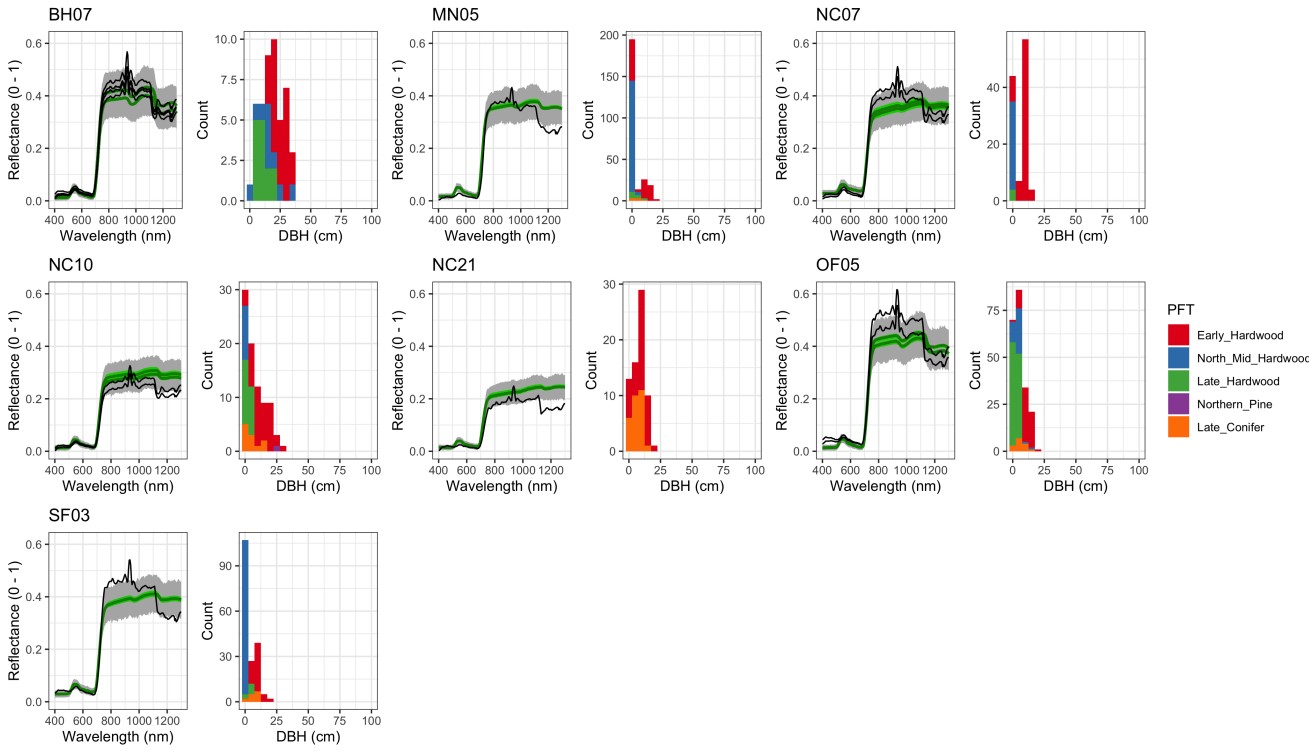

**Figure A11.** As above, but for sites where Early Hardwood trees had the largest mean DBH.

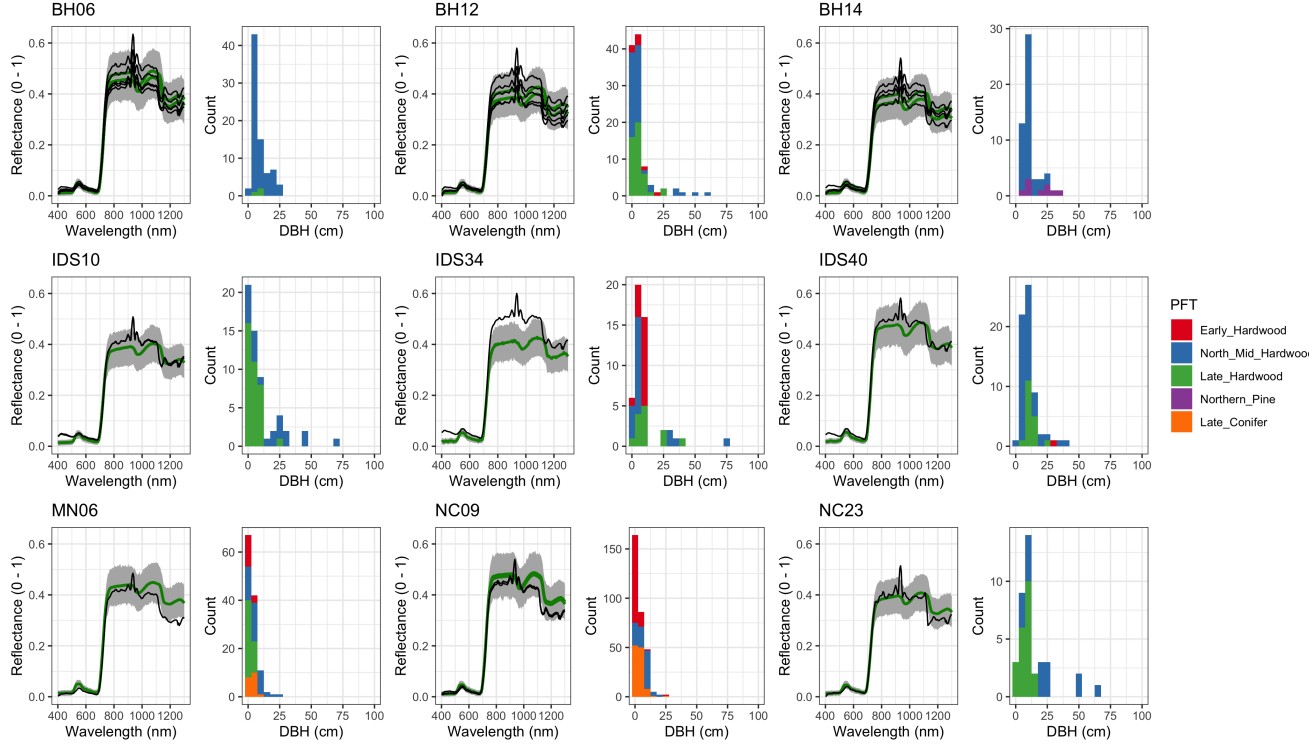

**Figure A12.** As above, but for sites where Mid Hardwood trees had the largest mean DBH.

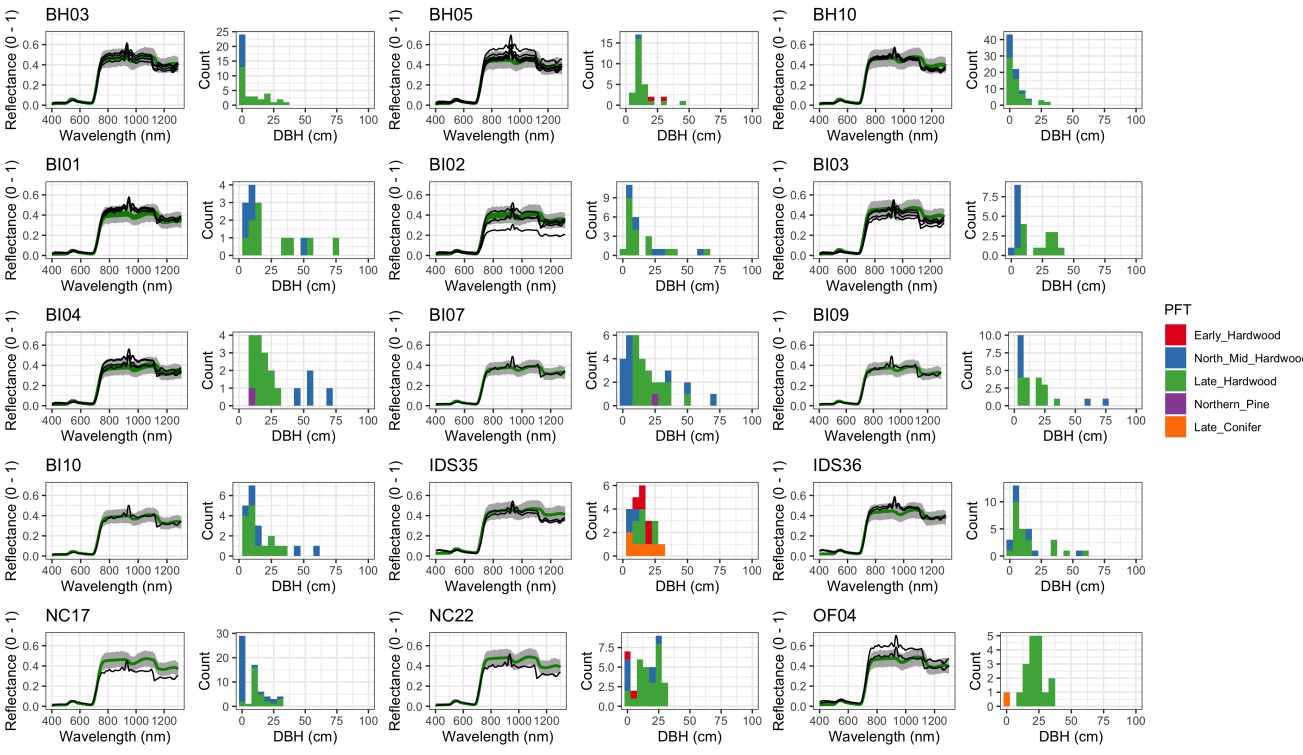

**Figure A13.** As above, but for sites where Late Hardwood trees had the largest mean DBH.

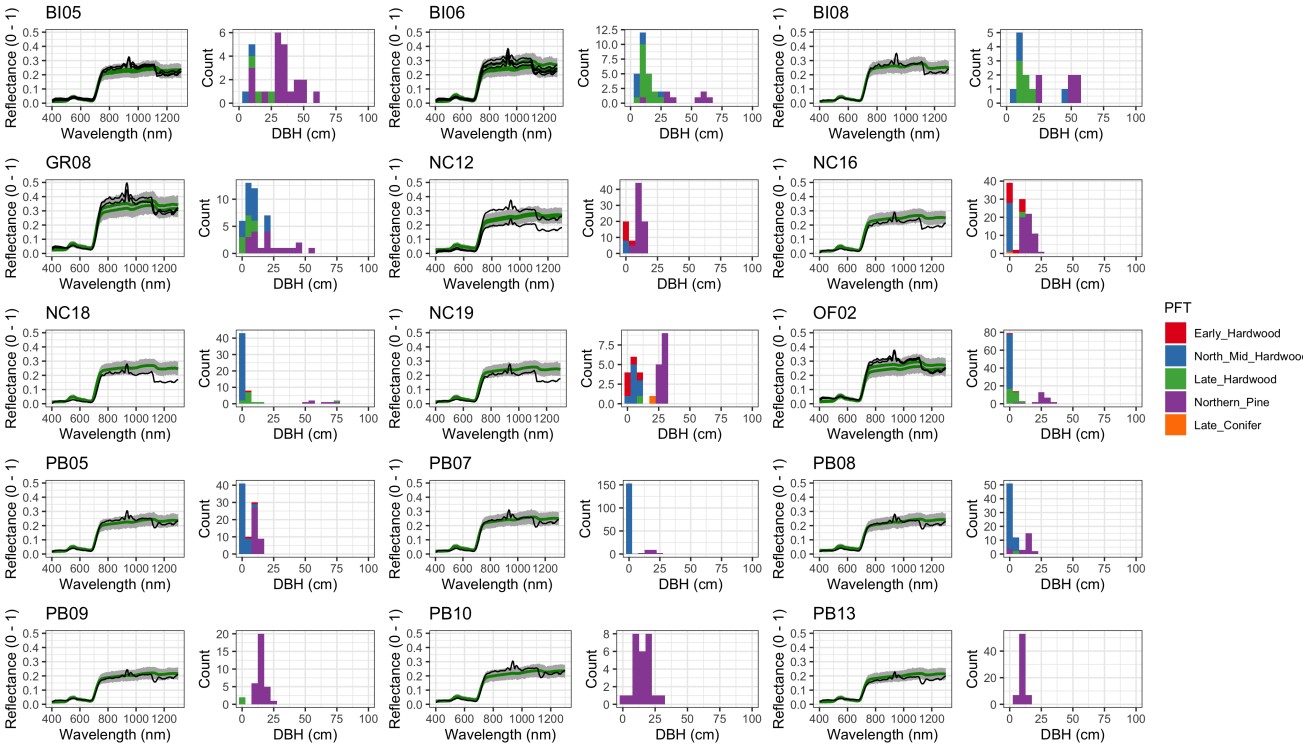

**Figure A14.** As above, but for sites where Pine trees had the largest mean DBH.

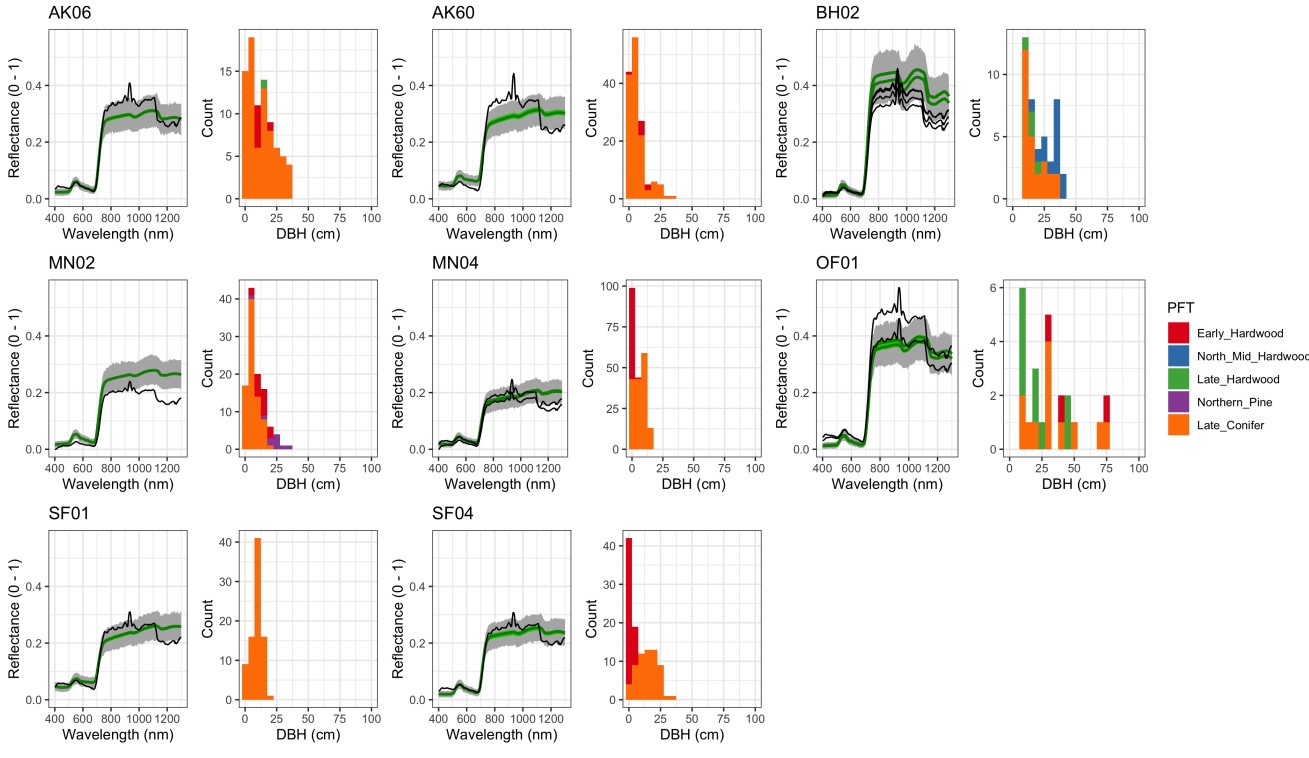

**Figure A15.** As above, but for sites where Late Conifer trees had the largest mean DBH.

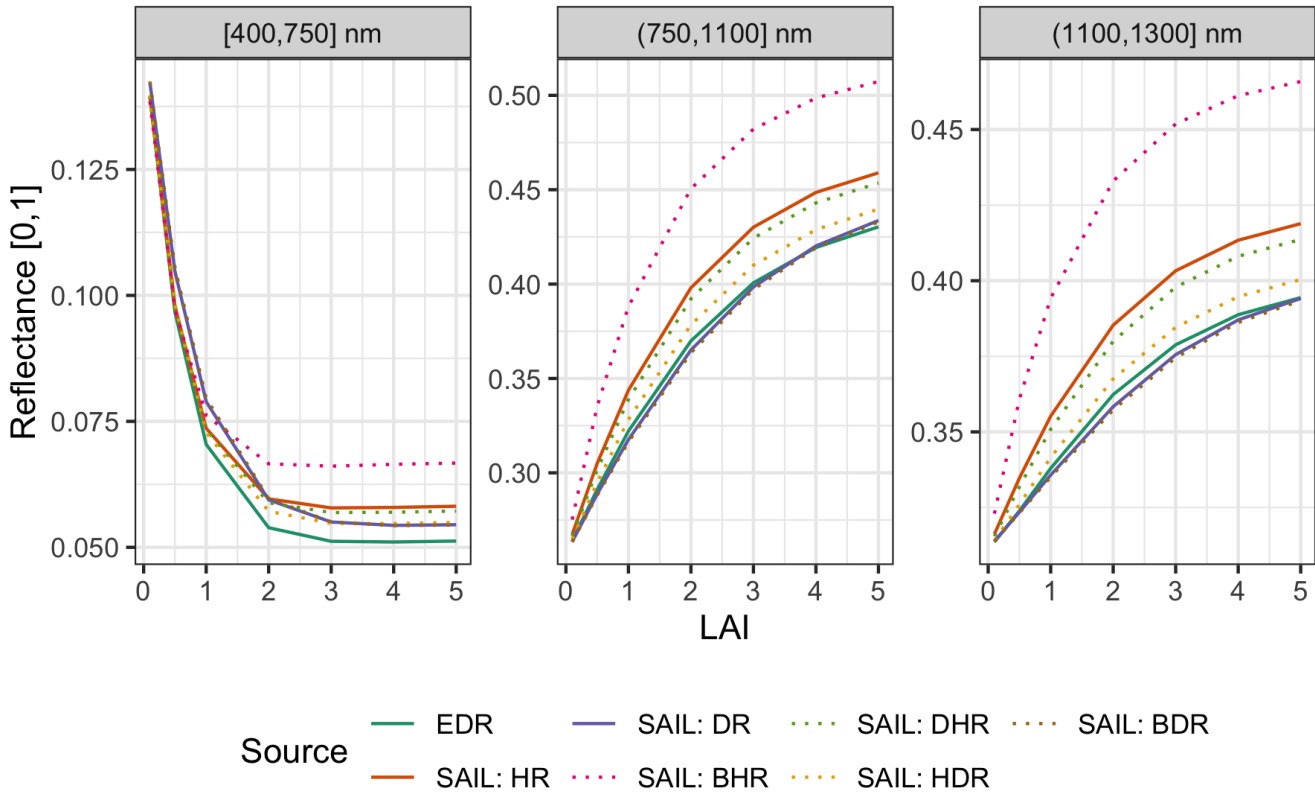

**Figure A16.** Same as Figure 10, but varying leaf area index (LAI) and fixing $\cos(\theta_s) = 0.85$, a typical value for our study.

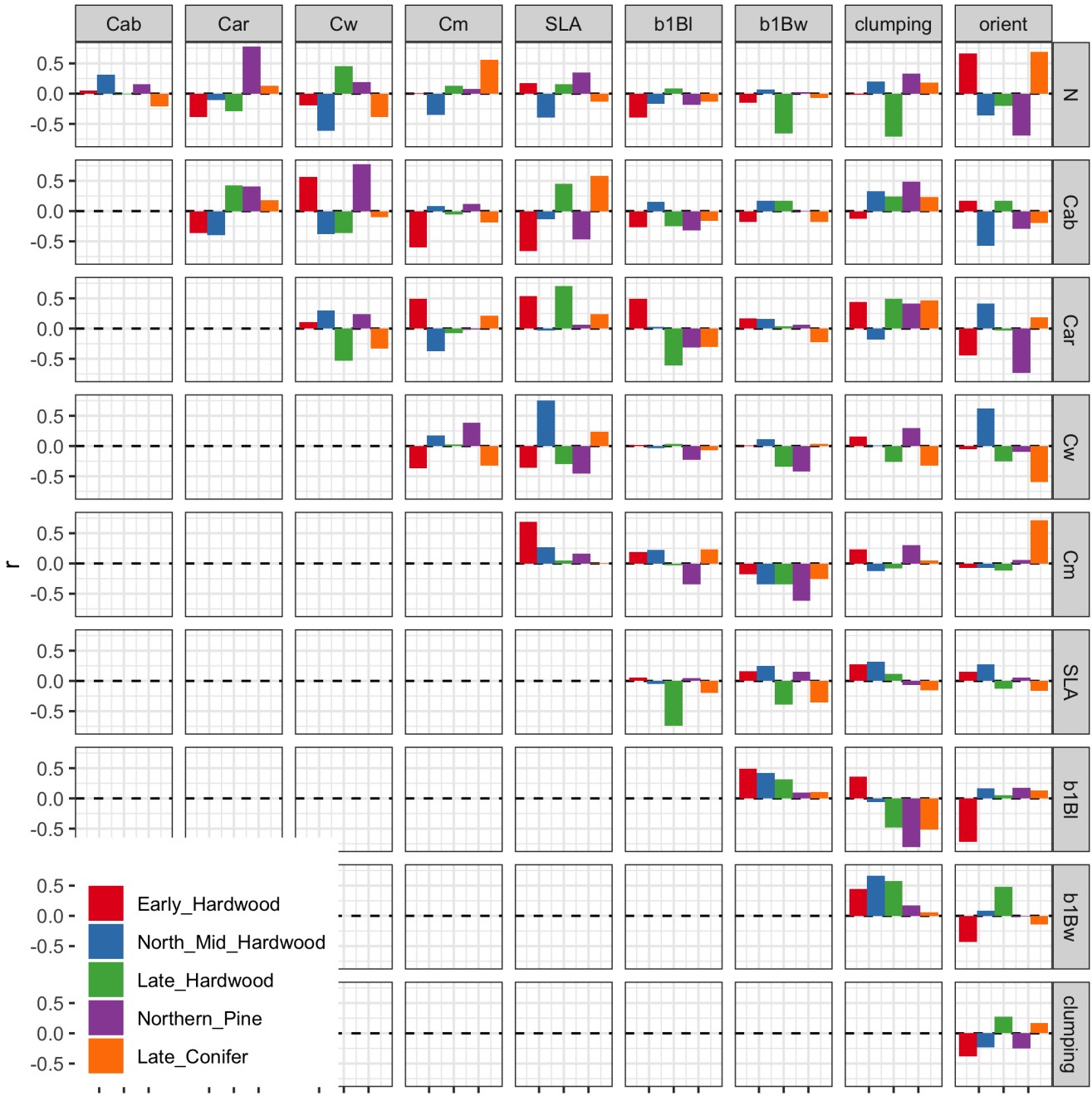

**Figure A17.** Posterior correlation matrix for PFT-specific parameters. Note that only correlations among parameters within the same PFT are shown—the full $106 \times 106$ dimensional correlation matrix is far too large to display in this format.

*Author contributions.* ANS led the analysis and manuscript preparation. MCD and SPS conceived of the original idea, participated in regular discussions about the study with ANS, and provided funding and infrastructure support. IF provided formatted input data on site structure and composition. TV developed the original version of the EDR code. All authors reviewed the manuscript draft and contributed revisions and suggestions.

*Competing interests.* The authors declare no competing interests.

*Acknowledgements.* This work was supported financially by NASA awards NNX14AH65G and NNX16AO13H, by NSF Awards 1655095 and 1457890, and by NASA's Surface Biology and Geology (SBG) mission study. Cyberinfrastructure for this work was provided by the Boston University Department of Earth and Environment, Brookhaven National Laboratory, Pacific Northwest National Laboratory, and NASA High-End Computing (HEC). We would also like to thank Dr. Tristan Quaife and one anonymous reviewer for their helpful feedback on the first version of this manuscript.

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
