# Peer review of "Cutting out the middleman: Calibrating and validating a dynamic vegetation model (ED2-PROSPECT5) using remotely sensed surface reflectance"

_Geoscientific Model Development, 2020_

## Referee Comment (RC1) · Anonymous Referee #1 · 6 Nov 2020

General Comments (overall quality): This manuscript uses remotely-sensed surface reflectance observations to calibrate/constrain a series of ecosystem parameters from the ED2-PROSPECT (EDR) model characterizing the land surface at 54 forested sites related to leaf biochemistry, canopy radiative transfer, and soil characteristics. An important innovation is the introduction of the radiative transfer model PROSPECT to the biosphere model ED2 to provide an improved spectrally resolved simulation of surface reflectance. This is done, in part, to bring the observed surface reflectance closer to what the model actually predicts, helping to reduce the impact of observational uncertainty as well as more effectively constraining multiple components of the model. The authors find that through the assimilation of surface reflectance EDR can provide better simulations of the surface reflectance spectra and leaf area. The findings suggest that this approach could be used to better constrain surface energy balance as well as overall ecosystem functioning. Given many other ecosystem models include a two-stream radiative approach they contend their results should be widely applicable.

Scientific Questions/Issues: The authors bring up the issue of equifinality in the introduction, which presents a challenge for this application in that the surface reflectance can be a function of leaf biochemical properties, leaf structure, and canopy and soil radiative transfer characteristics. To some degree, equifinality was reduced in that the prior parameter distributions for biochemical leaf properties were tightly restricted, and limited to the extent the surface reflectance observations could influence them. In contrast the canopy radiative transfer parameters and LAI (through SLA) were simultaneously being adjusted by the optimization. Was hoping the authors could comment more about how equifinality of the surface reflectance influenced their results.

As a follow up question to the equifinality question above – ED2 is a dynamic vegetation model with the ability to simulate competition amongst cohorts thus providing a simulation of co-existing dominant PFTs. It wasn't clear how well the simulation of cohort competition influenced the final distribution of PFTs and to what extent this matched the site level observed vegetation state. Given that the parameter optimization was PFT specific, the precise vegetation PFT distribution could have a large impact. Was the PFT distribution prescribed?

Was there any attempt to withhold some site reflectance data and apply the calibrated model parameters at those sites? It seems the optimized surface reflectance simulations were performed at sites that were already calibrated. The fact that the authors performed an across-site joint assimilation may in part account for this, but was interested how the calibrated parameters would perform at sites outside the calibration sites.

It is known that radiative transfer models are challenged in simulating evergreen species in part because of the irregular and open space canopy structure. Many of the figures in the supplement demonstrate stronger biases in simulated surface reflectance exist for evergreen sites as compared to deciduous. Was hoping the authors could comment on this, and recommendations for getting around this.

Whereas leaf level biochemistry related parameters were well constrained by the priors, many of the radiative transfer posterior parameters seemed to be edge hitting parameters. To what extent do the authors believe this behavior was caused from structural error in canopy radiation transfer and/or mismatch caused in part from radiation directionality differences between the simulated and observed canopy reflectance? The authors devote a considerable amount of general discussion regarding this topic, but don't directly address how this might have effected their own results.

The manuscript begins by justifying the inclusion of the PROSPECT model in to ED2 to bring the model closer to the observations, to, in part help reduce the uncertainty of the observation that are assimilated into EDR. More explanation on how the surface reflectance observational uncertainty was quantified here, and what it represents, and to what extent overconfidence in uncertainty may have led to the posterior edge hitting parameters.

Detailed Comments:

Abstract and manuscript in general: Need more discussion on what we hope to gain by this. We don't really care about surface reflectance (although energy balance is important), but we do care about how LAI, chlorophyll, pigments and water status influence ecosystem functioning through carbon and water exchange. I think this needs to be emphasized more, and provide evidence that this sort of setup can accomplish this.

Line 1: Remove 'derived'. The fact that they are 'data products' and not 'observations' gets across the point.

[Figure]

Line 5: 'compared against airborne and satellite data' Technically, this is still data and not observations in that even reflectance data requires RTM models, I believe. But it is more direct relationship

Line 23: add 'to' calibrate or constrain

Line 25: I know exactly what you mean by 'constrain', but could you use 'calibrate' or 'inform' in this context?

Line 32: "More sophisticated approaches for estimating vegetation properties based on physically-based radiative transfer models face issues of equifinality, whereby many different combinations of vegetation and soil properties can ultimately produce the same modeled surface reflectance (Combal et al., 2003; Lewis and 35 Disney, 2007)." This is important I think – and raises a key point for this analysis – is there not equifinality when trying to constrain leaf structure vs. leaf status? I would think equifinality could be a problem here, and I think you need to acknowledge this and how you might address this – Strong priors? Demonstration that surface reflectance can tease apart these two things. . . ...

Line 52: Awkward topic sentence. Simplify "Some land surface models already include there own . . .. . .. that allowd for a more direct comparison to remotely sensed surface reflectance."

Line 57:"Canopy radiative transfer plays a particularly important role in the current generation of demographically-enabled dynamic vegetation models, where differences in canopy radiative transfer representations and parametrizations have major impacts on predicted community composition and biogeochemistry (Loew et al., 2014; Fisher et al., 2018; Viskari et al., 2019)."

Seems weird to word it this way. Isn't it the other way around, community composition and biogeochemistry impact the RTM?

Line 72: ". . . . . .will significantly constrain model parameters related to canopy structure." So the goal all along was to constrain canopy structure with surface reflectance, not necessarily foliar biochemistry? Maybe talk a bit more about the differences in sensitivity of surface reflectance to canopy structure vs foliar biochemistry.

Section 2.3: Can you provide a sense of scale? For example for the 54 sites, what spatial range was the inventory data taken, and what spatial resolution did AVIRIS cover? Trying to get a feel for spatial mismatch, etc. Were sites chosen because they were rather homogeneous for certain PFTs?

Figure 1: Was a little surprised to see many sites so close to Lake Superior. No issues with interference from nearby water reflectance ? Line 200: Could you provide a bit more explanation of what including the EDR predicted LAI term within your probability function accomplishes? EDR response becomes saturated to LAI so is this an artificial way to account for increased reflectance ?

Line 227: So, to evaluate the model you compared the EDR-spectra against the AVIRIS observations at sites that were used to calibrate the model? Was there any attempt to withhold some site data and apply the calibrated model at those sites?

Line 232: Can you quantify what 'informative' means

Line 233: I cannot see in figure where PROSPECT N parameter is ?

Section 3 Results: Although Figure 2 was very informative, I found the Results section in general, relatively vague, perhaps some sense of % reductions in 95% credible interval.

Figure 2: Hopefully, the authors comment on some of the apparent edge-hitting parameters specifically related to canopy RTM parameters such as leaf orientation, canopy clumping and water. I worry that the information from the relatively strong and defensible leaf biochemistry prior parameters leading to relatively self contained and PFT-differentiated posteriors for the leaf biochemistry parameters is lost or made irrelevant due to biases between model simulated and observed surface reflectance that are

corrected by 'fitting' the RTM parameters.

The context of this manuscript doesn't indicate how sensitive the surface reflectance is for the suite of parameters calibrated here... perhaps included in one of cited manuscripts.

Figure 3: A map would be helpful with site codes provided. Perhaps a zoomed in map that demonstrates where these sites are spatially? Also is the stem diameter plot on the right simulated or observed? In fact, doesn't that have a large impact on the assimilation, the PFT distribution and stem diameter distribution?— has it been demonstrated that ED2 can properly simulate the competition of PFTs at the site providing the correct vegetation state, such that the parameter optimization reflects the observed vegetation state ?? Or has the vegetation state been prescribed in this case?

Figure 5: "The observed vs. predicted line had a slope of 0.37 and an intercept of 2.80, indicating that EDR calibration underpredicted LAI on average but overexagerrated across-site LAI variability." What do you attribute this clear structure in residuals between observed-simulated LAI? Line 258-270: I like the overview explanation of bringing observations closer to models or alternatively bring models closers to the observations. In the end, it's a bit of semantics, especially for using information from satellites we will always need some sort of transfer function or forward operator to convert from what a satellite observes and what a model predicts. I don't think one way or the other should take precident. The advantage in your approach, however, is the potential for the observed surface reflectance to constrain multiple model components, whether that be leaf structure or biochemistry or water status. I think that is potentially extremely valuable although I am not sure it has been demonstrated, yet, that this is the case. I feel more could be 'learned' about what information surface reflectance could provide if you could prescribe the LAI and PFT-distribution in ED2, then you could really get a grasp on what it can inform, leaf biochemistry? Within-canopy RTM parameters? Etc.

But, as you brought up in the introduction, this brings up equifinality issues. Not so much in this case for your leaf biochemistry parameters because of strong priors, but it does seem to be the case for canopy RTM parameters and predicted LAI (SLA). I think you need to caveat or address this concern.

Lines 286-310: It seems like you are pointing out sources of structural error within the radiative transfer of EDR or, to the extent, mismatch between what EDR simulates vs. what AVIRIS-Classic observes. Could this help explain why the calibration caused some posterior RTM parameters to be edge-hitting against the bounds of the priors? Also, this work was in part motivated by being better able to quantify the uncertainty in the surface reflection observations, but I am not sure I saw a clear explanation of the uncertainty that was used or provided for the AVIRIS and how was this quantified. It seems parameters have the potential to be overfit, if the observation uncertainty is not realistically quantified. May have missed this.

Line 320: Really it's the power to upscale that remote sensing products provide. However, this comes at a cost, they 'observe' reflected radiation from the land surface which indirectly characterizes things that we care about like, like leaf biochemistry, albedo etc.

Line 326: 'accurately reproduce surface reflectance and leaf area index' I think this is a bit of an overstatement, especially because the figures demonstrate systematic mismatch between the optimized surface reflectance and observed reflectance (figure 3), and strong residual error structure in LAI (Figure 5). Perhaps this approach provided 'improved' reflectance and LAI is better wording.

Line 330: I think you also need to say where this work can lead — this is of interest for those that are concerned with ecosystem functioning and that this could provide improved estimates of both biomass and land-atmosphere carbon and water exchange. Also some discussion of the differences between evergreen and deciduous forests would be helpful. Generally RTM's have more difficulty with more open canopy evergreen species and not really discussed in this manuscript.

Appendix:

Figure A2: What would be really helpful is to use the same color-coding in Figure A1 to show deciduous vs evergreen sites. Also this brings about the question – why did you choose the sites that you did to put in the manuscript itself?

Figure A3: This sort of gets at the hardwood vs conifer performance as well. I think it would be helpful to comment on this distinction in performance within the results/discussion.

Figure A13: This is also a very compelling figure that gets at the increased bias in spectra for conifer. Worth discussing in main manuscript.
* * *

---

## Referee Comment (RC2) · Tristan Quaife (Referee) · 16 Nov 2020

The manuscript "Cutting out the middleman: Calibrating and validating a dynamic vegetation model (ED2-PROSPECT5) using remotely sensed surface reflectance" by Shiklomanov et al. illustrates the assimilation of reflectance data observed from aircraft into a component of the ED2 model. In many respects this is an excellent paper. The Bayesian approach taken is state-of-the-art and the use of forward modelling of reflectance for assimilation purposes is desirable for many reasons (and yet surprisingly little progress has been made in this area for land surface studies, making this paper

especially welcome). The text is also well written on the whole.

Unfortunately I do have one rather significant concern about this paper, which may at first seem like a subtlety, but really is not. The authors are not "cutting out the middleman" so much as choosing to ignore them. I am concerned that the take home message of the paper for people less familiar with the underlying physics will be that it's OK to take this approach.

1. Assimilation of BRF and HDRFs.

[Note: I am using the definitions of reflectance quantities from Schaepman-Strub et al. (2006; hereafter S06), which the authors have also done.]

My major concern with this paper is the authors' misuse of different reflectance quantities. They are not comparing like with like and I do not agree that it is OK to assimilate quantities that are not physically consistent with those being modelled. The Sellers two-stream model predicts reflectances (BHR and DHR), whereas the AVRIS data are reflectance *factors* (BRF and HDRF). Although they are both unitless they are fundamentally different quantities and have different scales.

The authors do devote a paragraph to discussing this, but it is misleading and I am not convinced by the arguments they make. The statement that the ED radiative transfer model predicts BHR is only partly correct. It also simulates DHR and the predicted reflectance is a weighted mean of the two. The authors go on to state the AVRIS observations are most related to HDRFs. This would only be true for overcast skies. AVRIS observations are best represented as a mix of HDRFs and BRFs (although the reality is more complex of course as the down-welling diffuse flux is rarely isotropic). I argue that for most cases the AVRIS data will be most closely related to BRFs as most flight campaigns are under relatively clear skies.

I think part of the problem here arises because S06 define anything with more than 0

Using the Hogan et al (2018) paper to defend this position is disingenuous. One of the

things that paper shows quite clearly is that solar geometry effects are not well dealt with by classic two-stream formulations for complex canopies. The adjustments to the two-stream scheme made in that paper are required to make the model predicted DHR match the reference 3D model.

I am prepared to accept that in the specific experiment in this paper the limited angular sampling of AVRIS may mean that the overall effect will be small. However, when the authors make statements such as the one at Line 300 ("We therefore conclude that additional computational and conceptual challenges (as well as parameter uncertainties) associated with treatment of angular effects in similar models are unwarranted") it is very misleading: the take home message is that it is, in general, OK to take this approach. It is most definitely not.

On a related note I also don't accept the statements on lines 269 and 299 that seem to imply that it's necessary to have additional parameters to predict directional reflectances. This is not the case and hence also not a valid defence of the approach taken. The set of parameters required to define a two stream model can be used also to define a BRDF model derived from the same underlying assumptions (i.e. semi-infinite, plane parallel turbid media with point scatterers).

I propose the following modifications to address these issues:

a) Change the title to remove "cutting out the middleman."

b) Modify the discussion around this point, including stating clearly that the reflectances and reflectance factors are physically different things and that, in general, it is not appropriate to do assimilate one into a model that predicts the other.

c) Quantify the differences between the BRFs and the modelled BHRs. I noticed, poking around in the github repository, that the SAIL model is included and hence, presumably, it is trivial to take representative posterior parameters values and model both BRF (from SAIL) and BHRs. If the authors' assertion that it shouldn't make any

difference is correct then this will help to defend that. The observed reflectance factors should also be compared against SAIL predictions. I would be happy to iterate on the experimental procedure with the authors.

2. Is ED2 actually used in this paper? It seems that it is only a relatively small component of the model (i.e. the canopy radiative transfer scheme) that has been extracted. I think the title is slightly misleading in this respect. More important, I am not sure calling the code ED2-PROSPECT5 is appropriate. Perhaps EDR-PROSPECT5 would be better?

3. How are correlations between spectral channels dealt with? Do the authors use all of the AVRIS observations in the domain 400-1300nm? The errors in spectrally adjacent bands will be very highly correlated and the overall information content will be much lower the same number of independent observations.

Minor comments:

L27 The MacBean et al. (2018) paper referenced here does not assimilate a derived product in the sense the authors mean it. The assimilated data in that paper is solar induced fluorescence which, whilst it is "derived" in the sense that it is not what is being measured by the sensor, is no more derived that the surface reflectance data used in the manuscript under review. I suggest finding another reference here.

L35 "Meanwhile, the estimating" -> "Meanwhile, estimating"

L61 Surface reflectance is not assimilated in Zobbitz et al., (2014). The title is misleading (and I regret not standing my ground on that issue when we submitted that paper!); in fact fAPAR data is assimilated.

L85 Not sure this line should finish with a colon.

Eqn 3 What is meant by $r_{n+1}$ (and other variables with that subscript)? If I have understood correctly this is there are n layers, so what is the reflectance of the $n + 1$ layer? (I am sure I have just missed something here, but it was not obvious).

L102 "tau" should presumably be the Greek symbol $\tau$.

Eqn 7 I am confused by this, why is *forward* scattering defined by the sum of R+T? This is just the total scattering isn't it?

L188 This a different definition of $X$ from earlier in the paper. I suggest finding a different symbol.

L273 "DALEC-predicted foliar biomass, which required introducing an additional fixed parameter (grams of leaf carbon per leaf area) present in neither model." This statement is incorrect – that parameter already existed DALEC.

―――――――――――――

---

## Author Comment (AC1) · 5 Feb 2021

General Comments (overall quality): This manuscript uses remotely-sensed surface reflectance observations to calibrate/constrain a series of ecosystem parameters from the ED2-PROSPECT (EDR) model characterizing the land surface at 54 forested sites related to leaf biochemistry, canopy radiative transfer, and soil characteristics. An important innovation is the introduction of the radiative transfer model PROSPECT to the biosphere model ED2 to provide an improved spectrally resolved simulation

of surface reflectance. This is done, in part, to bring the observed surface reflectance closer to what the model actually predicts, helping to reduce the impact of observational uncertainty as well as more effectively constraining multiple components of the model. The authors find that through the assimilation of surface reflectance EDR can provide better simulations of the surface reflectance spectra and leaf area. The findings suggest that this approach could be used to better constrain surface energy balance as well as overall ecosystem functioning. Given many other ecosystem models include a two-stream radiative approach they contend their results should be widely applicable.

Scientific Questions/Issues: The authors bring up the issue of equifinality in the introduction, which presents a challenge for this application in that the surface reflectance can be a function of leaf biochemical properties, leaf structure, and canopy and soil radiative transfer characteristics. To some degree, equifinality was reduced in that the prior parameter distributions for biochemical leaf properties were tightly restricted, and limited to the extent the surface reflectance observations could influence them. In contrast the canopy radiative transfer parameters and LAI (through SLA) were simultaneously being adjusted by the optimization. Was hoping the authors could comment more about how equifinality of the surface reflectance influenced their results.

We agree that the topic of equifinality in the relationship between surface reflectance and canopy radiative transfer is worth additional discussion. We agree with the reviewer's assessment that equifinality was reduced by using informative priors for the leaf parameters. In addition, although not demonstrated here, another way to reduce equifinality is through the ecophysiological mechanisms embedded in the model itself — i.e., some combinations of model parameters and states that are consistent with a given surface reflectance may be excluded because they imply ecologically unrealistic

states in previous time steps or result in ecologically unrealistic outcomes in future time steps. Incorporating this dynamic aspect is an important future direction of this study.

In our revision, we added the following text to the discussion:

Nevertheless, our analysis echoed some known challenges in canopy radiative transfer modeling. One challenge is equifinality in the contributions of leaf biochemistry, leaf morphology, and different aspects of canopy structure to canopy albedo, which means that multiple variable and parameter combinations can produce very similar canopy albedo responses. We mitigated the equifinality between leaf traits and canopy structure by using informative priors on leaf traits from an independent data source (Shiklomanov 2018). However, there is additional equifinality in the effects of the EDR canopy structure parameters. For example, because the effective LAI used in EDR's actual radiative transfer calculations is defined as the product of "true" LAI and clumping factor, and because LAI is, in turn, derived from multiple parameters (leaf biomass allometry, specific leaf area), these parameters collectively cannot be independently determined from reflectance data alone. At the same time, increasing the leaf orientation factor (more horizontal, or "planophile", leaf orientation) has a similar (although not identical) effect to increasing LAI and clumping factor—namely, increasing canopy reflectance, especially in the near-infrared. Collectively, these issues may help explain some of the edge-hitting behavior (parameter distributions clustered at the ends of the distribution) observed in our posterior estimates, and some of the bias in our LAI estimates.

That being said, one major advantage of the Bayesian calibration approach is that its output is a joint posterior distribution that includes not only fully quantified uncertainties for each parameter but also the variance-covariance matrix of each parameter. Equifinality in parameters would manifest as strong pairwise correlation between parameters in the posterior distribution. Examining this correlation matrix (attached Figure 1) shows that there are some parameter pairs with strong correlations, such as the positive correlations between leaf and wood allometries for all PFTs except northern pine, and the hypothesized negative correlation between the leaf allometry (LAI) and

clumping factor, which was only observed for the early-successional hardwoods and northern pines. However, these correlations mostly do not occur in the parameters that exhibited strong edge-hitting behavior—namely, clumping and orientation factors for mid- and late-successional hardwood PFTs. Strong correlations also occurred among some of the PROSPECT parameters, and between PROSPECT and structural parameters, but contributed little to equifinality because the strong constraints on PROSPECT led to overall small covariance terms. Finally, because our calibration captured all of these covariances the presence of moderate equifinality did not preclude ecologically meaningful parameter constraints or accurate predictions because these covariances are being propagated into predictions. This is directly analogous to how a linear regression can have a tight confidence interval, despite high correlations between the slope and intercept, with that equifinality driving the characteristic hourglass shape of a regression confidence interval.

> As a follow up question to the equifinality question above – ED2 is a dynamic vegetation model with the ability to simulate competition amongst cohorts thus providing a simulation of co-existing dominant PFTs. It wasn't clear how well the simulation of cohort competition influenced the final distribution of PFTs and to what extent this matched the site level observed vegetation state. Given that the parameter optimization was PFT specific, the precise vegetation PFT distribution could have a large impact. Was the PFT distribution prescribed?

In this study, the vegetation composition at each site (including the PFT distribution and size-age structure) was prescribed in detail based on data from the NASA Forest Forest Functional Types (FFT) field campaign. In our revision, we have clarified this point in the first paragraph of "Site and data description", and briefly revisited it in the results in the context of future directions involving dynamic model simulations where the PFT distribution is predicted with some uncertainty.

[Figure]

Was there any attempt to withhold some site reflectance data and apply
the calibrated model parameters at those sites? It seems the optimized
surface reflectance simulations were performed at sites that were already
calibrated. The fact that the authors performed an across-site joint assim-
ilation may in part account for this, but was interested how the calibrated
parameters would perform at sites outside the calibration sites.

We acknowledge that cross-validation or out-of-sample validation are useful tests of
model performance, and in our revision, we recommend these activities as future di-
rections for this work. However as the reviewer points out, because our calibration
was joint across all sites, we did not feel that a separate validation at other sites not
used in the calibration was necessary. With 54 sites in our calibration, any single site
represents <2% of the data, and for a joint calibration without site random effects, we
have every reason to believe that the calibration is not overfitting to any individual site;
trying to fit any one site well would cause others to do worse (especially given the large
observed variability in forest structure) unless the EDR model structure was reason-
able and the parameters chosen were genuinely good choices. We have added this
information to our revised discussion.

It is known that radiative transfer models are challenged in simulating ev-
ergreen species in part because of the irregular and open space canopy
structure. Many of the figures in the supplement demonstrate stronger bi-
ases in simulated surface reflectance exist for evergreen sites as compared
to deciduous. Was hoping the authors could comment on this, and recom-
mendations for getting around this.

The reviewer brings up a great point about conifer canopies historically being harder
to capture. Although some conifer-dominated sites did demonstrate significant biases
in reflectance predictions, our analysis of reflectance bias by composition, structure,

and spectral region (Figures A3 and A4) shows that these biases are not systematic (though they may drive greater predictive variance). In our revision, we describe this analysis in more detail in the Methods and Results.

> Whereas leaf level biochemistry related parameters were well constrained by the priors, many of the radiative transfer posterior parameters seemed to be edge hitting parameters. To what extent do the authors believe this behavior was caused from structural error in canopy radiation transfer and/or mismatch caused in part from radiation directionality differences between the simulated and observed canopy reflectance? The authors devote a considerable amount of general discussion regarding this topic, but don't directly address how this might have effected their own results.

In our revision, we added as appendices a parameter sensitivity analysis of EDR (attached Figure 2) and an analysis comparing EDR to PRO4SAIL, a similar 1D two-stream canopy radiative transfer model popular in the remote sensing community. These analyses point to two likely explanations for the edge-hitting behavior of some parameters. The first—described above—is equifinality in the effects of several structural parameters on LAI and canopy albedo.

The second explanation is a structural issue in the EDR model that leads it to systematically underestimate albedo, particularly for low solar zenith angles (sun directly overhead). A detailed description of the issue is provided in Yuan et al. (2017). Briefly, EDR (and the Sellers, 1985, model from which EDR is derived) defines direct radiation backscatter as a function of the single-scattering albedo, which in turn is an integral involving the leaf scattering phase function and leaf projected area function. The Sellers (1985) analytical solution to this integral assumes a uniform scattering phase function, which is appropriate for point scatterers but less so for horizontal surfaces like leaves. The practical consequence of this assumption is a lower value of the direct radiation backscatter and therefore a tendency to underestimate albedo, which is consistent with

the results of our comparison of EDR and SAIL. Our EDR calibration is likely to be compensating for this behavior via a preference for higher effective LAI values (i.e., higher values of leaf biomass allometry and clumping factor) and more horizontal leaf distributions (i.e., higher leaf orientation factor), both of which increase albedo. We have added this information to our discussion.

> The manuscript begins by justifying the inclusion of the PROSPECT model in to ED2 to bring the model closer to the observations, to, in part help reduce the uncertainty of the observation that are assimilated into EDR. More explanation on how the surface reflectance observational uncertainty was quantified here, and what it represents, and to what extent overconfidence in uncertainty may have led to the posterior edge hitting parameters.

As shown in equation 26, observation error in the reflectance data was not estimated a priori based on the instrument itself, but was modeled as the residual error between the model and the data, analogous to what is done for any linear or nonlinear regression model. A key difference, however, is that the error model accounts for the known heteroskedasticity in spectral data (i.e., the size of the variance increases with the magnitude of the reflectance). In terms of random spectral errors, there is no reason to expect this variance to be overconfident for inferences made on these landscapes, especially as the study sites were not all imaged on the same day under the same atmospheric conditions, though we'd agree that one might not want to apply this variance to entirely different ecoregions. Furthermore, because the variance slope and intercept are fit parameters, whose parametric uncertainty is being quantified and propagated, this makes it even less likely that our uncertainty estimate is overconfident. That said, the current approach does not formally account for any possible systematic errors in the observations, which could have a more serious impact on inferences. However, we would note that we are unaware of any derived data products that account for these systematic errors either. Furthermore, in addition to the same uncertainties about

reflectance that we face, those products additionally contain numerous uncertainties about model structure, parameters, and covariate data whose uncertainties are rarely fully propagated, meaning that the alternative approach (calibration to derived data products) is more likely to result in overconfidence in uncertainties than the approach taken here. Along these lines we added a line to the Discussion about how accounting for systematic data errors would be a useful future direction.

That said, we feel it is highly unlikely that overconfidence in surface reflectance estimates contributed to the edge-hitting behavior of some of our posteriors. Random errors would not result in a systematic parameter bias and, given the long history of the instrument and maturity of atmospheric correction approaches, any systematic errors in AVIRIS are likely to be quite small relative to the structural uncertainty in ED2's RTM (i.e., biologically implausible parameters should not be necessary to capture observational data biases of the magnitude likely to be present).

> Detailed Comments: Abstract and manuscript in general: Need more discussion on what we hope to gain by this. We don't really care about surface reflectance (although energy balance is important), but we do care about how LAI, chlorophyll, pigments and water status influence ecosystem functioning through carbon and water exchange. I think this needs to be emphasized more, and provide evidence that this sort of setup can accomplish this.

We agree that additional emphasis on the implications of this work is warranted. In our revisions, we emphasize several important implications in the abstract, introduction, and discussion.

First, as you say, energy balance is important, and the contribution of vegetation to land surface albedo is a critical mechanism by which vegetation influences regional and global climate (Bonan 2008). Therefore, ensuring the accuracy of model simulations of

albedo, including its sensitivity to vegetation structure and composition, is essential to accurately projecting the effects of climate- and land use-driven changes to terrestrial ecosystems on future climate.

Second, canopy radiative transfer directly affects many physiological, ecological, and physical processes included in complex demographically-enabled vegetation models like ED2 (Viskari et al. 2019). Light availability and absorption is a first-order control on photosynthesis, and ability to survive under different light levels is an essential component of a tree species' position in forest succession. Meanwhile, temperature—which is strongly influenced by albedo—directly affects the rates of both enzyme-kinetic physiological processes and evaporation.

Finally, ED2 and similar models are highly sensitive to many of the leaf traits constrained by this analysis (e.g., Dietze et al. 2014; Raczka et al. 2018; Shiklomanov et al. 2020). Given the large variability of these traits through space and time, remote sensing is an essential data source for model parameterization, and our work provides a useful approach for doing so.

> Line 1: Remove 'derived'. The fact that they are 'data products' and not 'observations' gets across the point.

While we agree that "data products" should imply a difference from observations, treatment of remote sensing data products as true observations without accounting for uncertainties or biases is widespread in the Earth science community. Therefore, we think it is important to emphasize that these products are derived.

> Line 5: 'compared against airborne and satellite data' Technically, this is still data and not observations in that even reflectance data requires RTM models, I believe. But it is more direct relationship

We agree that additional nuance is required here (especially when also considering reviewer T. Quaife's comments). Therefore, we have revised this section to capture the fact that although these data are still derived (processing steps include atmospheric correction and orthorectification), they rely on fewer assumptions (especially about the land surface), have fewer processing steps, and therefore are closer to observations.

Line 23: add 'to' calibrate or constrain

We have revised this accordingly.

Line 25: I know exactly what you mean by 'constrain', but could you use 'calibrate' or 'inform' in this context?

We have replaced "constrain" with "inform".

Line 32: "More sophisticated approaches for estimating vegetation properties based on physically-based radiative transfer models face issues of equifinality, whereby many different combinations of vegetation and soil properties can ultimately produce the same modeled surface reflectance (Combal et al., 2003; Lewis and 35 Disney, 2007)." This is important I think – and raises a key point for this analysis – is there not equifinality when trying to constrain leaf structure vs. leaf status? I would think equifinality could be a problem here, and I think you need to acknowledge this and how you might address this – Strong priors? Demonstration that surface reflectance can tease apart these two things...

Please see our above response about equifinality.

[Figure]

Line 52: Awkward topic sentence. Simplify "Some land surface models already include there own . . . that allowd for a more direct comparison to remotely sensed surface reflectance."

We have revised this sentence according to the reviewer's suggestion.

Line 57:"Canopy radiative transfer plays a particularly important role in the current generation of demographically-enabled dynamic vegetation models, where differences in canopy radiative transfer representations and parametrizations have major impacts on predicted community composition and biogeochemistry (Loew et al., 2014; Fisher et al., 2018; Viskari et al., 2019)." Seems weird to word it this way. Isn't it the other way around, community composition and biogeochemistry impact the RTM?

It is true that composition and structure impacts the RTM, but all of these studies show that the opposite is true as well! These studies illustrate that because the RTM determines the overall energy balance of the ecosystem and the distribution of light within the canopy, RTM formulation and parameters profoundly impact the predicted biogeochemical fluxes and vegetation dynamics. For example, Viskari et al. (2019) show that uncertainties in canopy RT can cascade and impact a number of associated processes, including photosynthesis, energy balance, internal competition, and demography. We have revised the sentences here to more clearly and explicitly convey this idea.

Line 72: ". . . will significantly constrain model parameters related to canopy structure." So the goal all along was to constrain canopy structure with surface reflectance, not necessarily foliar biochemistry? Maybe talk a bit more about the differences in sensitivity of surface reflectance to canopy structure vs foliar biochemistry.

[Figure]

Our objective was to evaluate which parameters could be constrained. The list of candidate parameters included parameters related to both structure and biochemistry. However, we hypothesized that, because of the informative priors on foliar biochemistry, the constraint would be relatively greater for canopy structural parameters (as stated here).

Per our earlier responses, we have included an additional parameter sensitivity analysis of EDR and discussion thereof. We have also added more text on the issue of equifinality to the introduction and discussion.

> Section 2.3: Can you provide a sense of scale? For example for the 54 sites, what spatial range was the inventory data taken, and what spatial resolution did AVIRIS cover? Trying to get a feel for spatial mismatch, etc. Were sites chosen because they were rather homogeneous for certain PFTs?

In response to this and Reviewer T. Quaife's comments, we have elaborated on the methods behind the data used in this analysis. Specifically, each of the 54 sites here consisted of a 60 x 60 m transect within which forest inventory data were collected. AVIRIS-Classic data were extracted as the average of a 3x3 pixel array (each pixel is 15-20m, depending on aircraft altitude) centered on the site transect center, resulting in a single composite spectrum for the 60 x 60 m area. Additional details on the sampling methodology are described in Singh et al. (2015) and references therein.

> Figure 1: Was a little surprised to see many sites so close to Lake Superior. No issues with interference from nearby water reflectance?

Although sites do appear very close to Lake Superior in the map we provide, all sites are sufficiently inland (several kilometers) that contributions from water reflectance of

nearby pixels can safely be assumed to be negligible. In the revision, this figure has been broken up into two figures

> Line 200: Could you provide a bit more explanation of what including the EDR predicted LAI term within your probability function accomplishes? EDR response becomes saturated to LAI so is this an artificial way to account for increased reflectance?

Yes, the LAI penalty in the likelihood function accounts for the saturating effect of increasing LAI on reflectance. This effect is not unique to EDR—rather, it is a well-known consequence of the exponential extinction of light through a medium, following Beer's Law. Therefore, a canopy with an unrealistically high LAI like 15 has virtually the same reflectance as a canopy with a high but more feasible LAI like 6 (all else being equal), and therefore a likelihood calculation that does not penalize excessively high LAI values would consider both outcomes equally likely.

We have added additional text to this effect to the methods, and have added the attached figure to the supplement demonstrating the saturating effect of LAI on reflectance.

> Line 227: So, to evaluate the model you compared the EDR-spectra against the AVIRIS observations at sites that were used to calibrate the model? Was there any attempt to withhold some site data and apply the calibrated model at those sites?

As we stated in our main response, because our calibration was joint across all sites, we did not feel that a separate validation at other sites not used in the calibration was necessary.

> Line 232: Can you quantify what 'informative' means

We have clarified this sentence to say these are "leaf parameters whose prior distributions were already independently constrained by an earlier analysis". These prior distributions are shown in Figure 2.

Line 233: I cannot see in figure where PROSPECT N parameter is?

In our experience, labelling PROSPECT's "N" parameter as such is confusing to readers unfamiliar with PROSPECT because it suggests leaf nitrogen content. Therefore, we prefer the more explicit name "# mesophyll layers". We have revised the methods and results text in a few places to clarify this.

Section 3 Results: Although Figure 2 was very informative, I found the Results section in general, relatively vague, perhaps some sense of % reductions in 95% credible interval.

We have revised the results to include more precise statistics, including the suggested relative reductions in the width of the 95% credible interval. For example, relative to the prior, posterior credible intervals were 19% to 58% as wide for clumping factor, 14% to 78% as wide for leaf orientation factor, and 1% to 10% as wide for leaf biomass allometry.

Figure 2: Hopefully, the authors comment on some of the apparent edge-hitting parameters specifically related to canopy RTM parameters such as leaf orientation, canopy clumping and water. I worry that the information from the relatively strong and defensible leaf biochemistry prior parameters leading to relatively self contained and PFT differentiated posteriors for the leaf biochemistry parameters is lost or made irrelevant due to biases between model simulated and observed surface reflectance that are corrected

by 'fitting' the RTM parameters. The context of this manuscript doesn't indicate how sensitive the surface reflectance is for the suite of parameters calibrated here... perhaps included in one of cited manuscripts.

We added text to the results section highlighting the edge-hitting behavior for wood biomass allometry, canopy clumping, and leaf orientation. We also added figures showing the sensitivity of EDR to the relevant parameters in this figure to the supplementary information.

For wood biomass allometry, the edge-hitting behavior approaching zero is consistent with the typically small-to-negligible contribution of aboveground woody elements (stems and branches) to reflectance of dense canopies during the growing season (Banskota et al 2015). However, these woody elements are a large biomass sink, so constraining wood allometry parameters is important. We have added text to the discussion about the importance of constraining wood allometry, the limitations of doing so using canopy reflectance alone, and additional analytical (e.g., via known covariance with other parameters) or observational (e.g., leaf-off optical, LiDAR, radar, in situ) constraints that should be explored.

For canopy clumping, we observed the edge-hitting behavior primarily for mid- and late-hardwood PFTs in the direction of no clumping. In EDR, clumping factor appears only as a scaling factor on LAI—namely, EDR defines the effective LAI, eLAI, as the product of LAI and clumping factor, and the main radiative transfer calculations use eLAI to quantify the depth of vegetation. This makes the clumping factor parameter highly confounded with LAI (i.e., a two-fold increase in LAI is perfectly compensated by a two-fold decrease in clumping factor). The edge hitting behavior of clumping factor at its maximum—1.0—suggests the tendency of the calibration for these PFTs to prefer larger leaf area indices, which we also see in the leaf biomass allometry. As mentioned in our earlier response, this may be compensating for a tendency of EDR to underestimate albedo (increasing LAI tends to increase albedo). Moreover,

it is expected that dense, mature hardwood PFTs would trend toward a less-clumped canopy compared with a more open, clumped evergreen needle-leaf canopy.

For leaf orientation, we primarily observe edge-hitting behavior approaching the most horizontal leaf orientation for the late conifers and mid- and late-hardwoods. As mentioned in the earlier response, and as with LAI above, this may be compensation for EDR's tendency to under-predict albedo.

Finally, the leaf water parameters are not truly edge-hitting but do show a consistent directional shift with the AVIRIS data suggesting a higher leaf water content than the prior leaf-level data. This is perhaps not surprising as retrieval of canopy water content with hyperspectral data, and specifically AVIRIS, has been one of the most widely-utilized methods in the literature (e.g. Gao and Goetz, 1995; Clevers et al., 2010). In this case, our parameter distributions for EWT suggest that given the time of year when the imagery was collected the vegetation tended toward higher canopy moisture conditions, likely given this was during the peak of the growing season, and peak greenness and neither year had any indication of lower than normal precipitation or drought.

Banskota, A., S. P. Serbin, R. H. Wynne, V. A. Thomas, M. J. Falkowski, N. Kayastha, J. P. Gastellu-Etchegorry, and P. A. Townsend. 2015. An LUT-Based Inversion of DART Model to Estimate Forest LAI from Hyperspectral Data. Selected Topics in Applied Earth Observations and Remote Sensing, IEEE Journal of 8:3147-3160.

Gao, B.-C., & Goetz, A. F. H. (1995). Retrieval of equivalent water thickness and information related to biochemical components of vegetation canopies from AVIRIS data. Remote Sensing of Environment, 52(3), 155–162. https://doi.org/10.1016/0034-4257(95)00039-4

Clevers, J. G. P. W., Kooistra, L., & Schaepman, M. E. (2010). Estimating canopy water content using hyperspectral remote sensing data. International Journal of Applied Earth Observation and Geoinformation, 12(2), 119–125.

https://doi.org/10.1016/j.jag.2010.01.007

> Figure 3: A map would be helpful with site codes provided. Perhaps a zoomed in map that demonstrates where these sites are spatially? Also is the stem diameter plot on the right simulated or observed? In fact, doesn't that have a large impact on the assimilation, the PFT distribution and stem diameter distribution?— has it been demonstrated that ED2 can properly simulate the competition of PFTs at the site providing the correct vegetation state, such that the parameter optimization reflects the observed vegetation state ?? Or has the vegetation state been prescribed in this case?

We have split up the original Figure 1 into two separate figures: one showing a larger map with sites labelled (attached Figure 3) and one showing the density vs. diameter plot (updated to include a self-thinning curve based on T. Andrews, unpublished; attached Figure 4).

As stated in the main response, the vegetation composition at each site (including the PFT distribution and size-age structure) was prescribed in detail based on NASA Forest Functional Types (FFT) campaign field data. In this figure's caption, we have clarified that these are the observed stand structures.

Zeide, B. (2010). Comparison of self-thinning models: an exercise in reasoning. Trees, 24, 1117–1126. https://doi.org/10.1007/s00468-010-0484-z

> Figure 5: "The observed vs. predicted line had a slope of 0.37 and an intercept of 2.80, indicating that EDR calibration underpredicted LAI on average but overexagerrated across-site LAI variability." What do you attribute this clear structure in residuals between observed-simulated LAI?
We agree the clear mismatch between predicted and observed LAI was not given sufficient attention in our original draft. We have incorporated the following analysis into our revision:

EDR tended to underpredict LAI at sites with lower mean DBH and overpredict LAI at sites with higher mean DBH (attached figure). However, there was no relationship between LAI bias (predicted - observed) and stand density, proportion of conifer PFTs, or bias in predicted reflectance (figures attached). Breaking the LAI bias down based on PFT site dominance shows that the bias was most pronounced in sites dominated by Early Hardwood and Northern Pine trees (attached figure). Based on these results, we conclude that the LAI bias can be attributed to an underestimation of the leaf biomass allometry parameter, as both Early Hardwood and Northern Pine PFTs had relatively low estimates of the leaf biomass allometry parameter. In part, this may be a consequence of site sampling: both Early Hardwood- and Pine-dominated sites skewed towards stands with low DBH, where sensitivity to changes in the allometry parameter are relatively low. For pine sites, the challenges with modeling conifer radiative transfer (see earlier responses) likely also played a part. Overall, this analysis reiterates the importance of evaluating models against multiple distinct variables—after all, none of these biases would have been apparent from looking at the reflectance simulations alone.

> Line 258-270: I like the overview explanation of bringing observations closer to models or alternatively bring models closers to the observations. In the end, it's a bit of semantics, especially for using information from satellites we will always need some sort of transfer function or forward operator to convert from what a satellite observes and what a model predicts. I don't think one way or the other should take precident. The advantage in your approach, however, is the potential for the observed surface reflectance to constrain multiple model components, whether that be leaf structure or biochemistry or water status. I think that is potentially extremely valuable

although I am not sure it has been demonstrated, yet, that this is the case. I feel more could be 'learned' about what information surface reflectance could provide if you could prescribe the LAI and PFT-distribution in ED2, then you could really get a grasp on what it can inform, leaf biochemistry? Within-canopy RTM parameters? Etc.

We disagree that the difference between bringing models closer to observations and vice-versa is a purely semantic one – the fact that all observations need some transformation does not mean that all transformations are the same. Instead, we argue that comparing model output to a highly derived data product (e.g., MODIS GPP) is closer to a model intercomparison than a model validation, since generating those data products invariably requires their own models (whether statistical or process-based) that make specific assumptions about different processes, often very different assumptions than the model the products are compared with. Thus it's often really an "apples to oranges" comparison. In other words, in addition to the "shared" observational uncertainties (e.g. atmospheric correction, instrument calibration) that are present in both approaches, derived data products include numerous additional assumptions and uncertainties that are avoided in a spectra-to-spectra comparison. Meanwhile, as you point out, the approach of bringing models closer to observations is advantageous precisely because of the model's emergent covariance across many disparate variables rooted in specific assumptions about biophysical and ecological processes. In practice, for an observation operator that is almost entirely disconnected from the model (for example, if we just took the total LAI from ED2 and used a standalone RTM like SAIL to simulate the reflectance), we agree that there's relatively little advantage relative to a derived product, but the more tightly an observation operator is integrated into a model (as in our analysis, where canopy RTM parameters and outputs are used elsewhere in ED2), the greater its value.

To address this, we have significantly revised the discussion to (1) more explicitly highlight the kinds of assumptions typically made in remote sensing data product generation; and (2) to discuss the relative advantage of building models with remote sensing-friendly observation operators, especially when such operators are tightly integrated with other processes in the model. For the latter, we now discuss some future directions for ED2 and optical remote sensing specifically, such as using the PROSPECT leaf model to couple the model's representations of plant hydraulics (for leaf water content) and shade-driven trait plasticity (for leaf structure and pigmentation) to canopy radiative transfer. Similar approaches that leverage the model's internal RTM can also be applied to other remotely-sensed data, such as lidar (canopy structure), radar/microwave (structure and water content), thermal (canopy traits and water use), and fluorescence (photosynthetic traits). Indeed, this paper is an important first step towards the ultimate goal of leveraging internally-consistent process-based models to make joint inferences across multiple remotely-sensed data constraints simultaneously.

> But, as you brought up in the introduction, this brings up equifinality issues. Not so much in this case for your leaf biochemistry parameters because of strong priors, but it does seem to be the case for canopy RTM parameters and predicted LAI (SLA). I think you need to caveat or address this concern.

We agree that equifinality between canopy structure and leaf biochemistry in general, and the specific ways that canopy structure is represented in EDR, are concerns that warrant further discussion. As we stated in the introduction, and as you point out in your review, an effective way to address equifinality is by incorporating prior information that can constrain the parameters. In this study, we showed that external informative priors on leaf biochemistry parameters are effective. Other data—for instance, observations of canopy structure from active remote sensing (LiDAR, radar) or in situ measurements—could help alleviate some of the other issues with our results. Moreover, if our approach was applied in dynamic model simulations, the internal logic of the model's dynamics of leaf biochemistry and canopy structure would provide additional constraint on the possible parameter space, which is an explicit future goal.

We have elaborated on all of these points in the revised discussion.

> Lines 286-310: It seems like you are pointing out sources of structural error within the radiative transfer of EDR or, to the extent, mismatch between what EDR simulates vs. what AVIRIS-Classic observes. Could this help explain why the calibration caused some posterior RTM parameters to be edge-hitting against the bounds of the priors? Also, this work was in part motivated by being better able to quantify the uncertainty in the surface reflection observations, but I am not sure I saw a clear explanation of the uncertainty that was used or provided for the AVIRIS and how was this quantified. It seems parameters have the potential to be overfit, if the observation uncertainty is not realistically quantified. May have missed this.

See earlier responses.

> Line 320: Really it's the power to upscale that remote sensing products provide. However, this comes at a cost, they 'observe' reflected radiation from the land surface which indirectly characterizes things that we care about like, like leaf biochemistry, albedo etc.

We agree the reflectance data collected by passive optical remote sensing are affected by many different surface features and processes, often in confounding ways, which makes them challenging to use in isolation. Fully leveraging the power of remote sensing requires additional sources of information from both in situ data and the understanding of biophysical and ecological processes embedded in our vegetation models. We have revised this text and expanded this text accordingly.

> Line 326: 'accurately reproduce surface reflectance and leaf area index' I think this is a bit of an overstatement, especially because the figures demonstrate systematic mismatch between the optimized surface re-
flectance and observed reflectance (figure 3), and strong residual error
structure in LAI (Figure 5). Perhaps this approach provided 'improved' re-
flectance and LAI is better wording.

We agree that we overstated our LAI prediction accuracy here, and have revised this
text to temper our conclusions accordingly. However, as we pointed out in earlier com-
ments, although our reflectance predictions were not perfect everywhere, the quanti-
tative analysis of surface reflectance revealed no systematic biases by PFT or stem
density.

Line 330: I think you also need to say where this work can lead —
this is of interest for those that are concerned with ecosystem function-
ing and that this could provide improved estimates of both biomass and
land-atmosphere carbon and water exchange. Also some discussion of
the differences between evergreen and deciduous forests would be help-
ful. Generally RTM's have more difficulty with more open canopy evergreen
species and not really discussed in this manuscript.

Our significantly revised discussion (see earlier comments) includes much more text
on ways that ED2 (and other vegetation models) can be further enhanced to better take
advantage of passive optical and other remote sensing techniques. In addition, to ad-
dress this comment specifically, we have added several sentences to the conclusions
about the role of better-constrained vegetation models in improved estimates and un-
derstanding of biomass and vegetation-atmosphere interactions. Finally, earlier in the
discussion, we have added a few sentences about known the challenges with canopy
radiative transfer modeling in conifer species, though again, our results did not show
any systematic biases in reflectance predictions by plant functional type.

Appendix:

Figure A2: What would be really helpful is to use the same color-coding in Figure A1 to show deciduous vs evergreen sites. Also this brings about the question – why did you choose the sites that you did to put in the manuscript itself?

Figures A5-A13 provide the requested breakdown of plots by PFT composition. For the main figure, we selected sites that, collectively, span the geographic, stand-structure, and PFT composition space of our study. We have added this information to the Figure 3 caption, highlighted these plots in our site map, and mention this in the revised Results section.

Figure A3: This sort of gets at the hardwood vs conifer performance as well. I think it would be helpful to comment on this distinction in performance within the results/discussion.

As mentioned above, we have added additional text about our analysis of performance by PFT to the methods, results, and discussion. As this figure clearly shows, there is no systematic bias in reflectance simulations for conifer species.

Figure A13: This is also a very compelling figure that gets at the increased bias in spectra for conifer. Worth discussing in main manuscript.

See earlier comments about this. We note that even this figure shows no systematic bias—three sites have underpredicted reflectance (AK06, AK60, OFO1), two sites have overpredicted reflectance (MN02, MN04), and the remaining 3 sites have relatively accurate reflectance predictions (BH02, SF01, SF04).

Please also note the supplement to this comment:
https://gmd.copernicus.org/preprints/gmd-2020-324/gmd-2020-324-AC1-
supplement.pdf

---

## Author Comment (AC2) · 5 Feb 2021

The manuscript "Cutting out the middleman: Calibrating and validating a dynamic vegetation model (ED2-PROSPECT5) using remotely sensed surface reflectance" by Shiklomanov et al. illustrates the assimilation of reflectance data observed from aircraft into a component of the ED2 model. In many respects this is an excellent paper. The Bayesian approach taken is state-of-the-art and the use of forward modelling of reflectance for assimilation purposes is desirable for many reasons (and yet surprisingly little

progress has been made in this area for land surface studies, making this paper especially welcome). The text is also well written on the whole.

Unfortunately I do have one rather significant concern about this paper, which may at first seem like a subtlety, but really is not. The authors are not "cutting out the middleman" so much as choosing to ignore them. I am concerned that the take home message of the paper for people less familiar with the underlying physics will be that it's OK to take this approach.

1. Assimilation of BRF and HDRFs.

[Note: I am using the definitions of reflectance quantities from Schaepman-Strub et al. (2006; hereafter S06), which the authors have also done.]

My major concern with this paper is the authors' misuse of different reflectance quantities. They are not comparing like with like and I do not agree that it is OK to assimilate quantities that are not physically consistent with those being modelled. The Sellers two-stream model predicts reflectances (BHR and DHR), whereas the AVRIS data are reflectance factors (BRF and HDRF). Although they are both unitless they are fundamentally different quantities and have different scales.

The authors do devote a paragraph to discussing this, but it is misleading and I am not convinced by the arguments they make. The statement that the ED radiative transfer model predicts BHR is only partly correct. It also simulates DHR and the predicted reflectance is a weighted mean of the two. The authors go on to state the AVRIS observations are most related to HDRFs. This would only be true for overcast skies. AVRIS observations are best represented as a mix of HDRFs and BRFs (although the reality is more complex of course as the down-welling diffuse flux is rarely isotropic). I argue that for most cases the AVRIS data will be most closely related to BRFs as most flight campaigns are under relatively clear skies.

[Figure]

I think part of the problem here arises because S06 define anything with more than 0

Using the Hogan et al (2018) paper to defend this position is disingenuous. One of the things that paper shows quite clearly is that solar geometry effects are not well dealt with by classic two-stream formulations for complex canopies. The adjustments to the two-stream scheme made in that paper are required to make the model predicted DHR match the reference 3D model.

I am prepared to accept that in the specific experiment in this paper the limited angular sampling of AVRIS may mean that the overall effect will be small. However, when the authors make statements such as the one at Line 300 ("We therefore conclude that additional computational and conceptual challenges (as well as parameter uncertainties) associated with treatment of angular effects in similar models are unwarranted") it is very misleading: the take home message is that it is, in general, OK to take this approach. It is most definitely not.

On a related note I also don't accept the statements on lines 269 and 299 that seem to imply that it's necessary to have additional parameters to predict directional reflectances. This is not the case and hence also not a valid defence of the approach taken. The set of parameters required to define a two stream model can be used also to define a BRDF model derived from the same underlying assumptions (i.e. semi-infinite, plane parallel turbid media with point scatterers).

We thank the reviewer for their insightful and valuable comments on this topic. We fully agree that there is a subtle but important difference between reflectance simulated by EDR and reflectance factors measured by AVIRIS that precludes a direct comparison between the two. We also recognize that our original discussion of this topic was

flawed, and as stated, misleading.

That said, because we were using AVIRIS classic imagery that underwent additional processing for the NASA Forest Functional Types (FFT) project (Singh et al 2015), we feel that our analysis is still valid. In addition to the standard atmospheric correction and orthorectification conducted by NASA JPL, the AVIRIS data we used for model calibration were also cross-track illumination corrected, as well as BRDF-corrected, following the procedure of Lucht et al. (2000, doi:10.1109/36.841980). Briefly, the latter approach estimates "intrinsic surface albedo" — the precise quantity that is simulated by EDR — from angular reflectance data through application of a polynomial approximation to the Ross-Li semiempirical BRDF model. The full AVIRIS processing pipeline for the AVIRIS data (including the BRDF approximation) we used is described in Singh et al. (2015, doi:10.1890/14-2098.1).

We have addressed this comment in the following ways: First, we have added additional information about our AVIRIS processing pipeline (including the BRDF correction) to the "Site and data description" of the Methods, along with a reference to the Singh et al. (2015) paper from which our data originated.

Second, and more importantly, we have completely rewritten our discussion of BRDF effects, taking care to emphasize that surface reflectance factors from AVIRIS and similar sensors cannot be directly compared to intrinsic albedo / reflectance simulations from EDR and similar models without BRDF correction. The revised discussion now reads as follows:

A significant caveat to the broader application of our approach is that there is a subtle but significant difference between the physical quantity EDR predicts and the quantities typically measured by optical remote sensing. Specifically, EDR predicts the hemispherical reflectance—the ratio of total radiation leaving the surface to the total radiation incident upon the surface, integrated over all viewing angles (a.k.a, blue-sky albedo). On the other hand, optical remote sensing platforms typically measure the

directional reflectance factor—the ratio of actual radiation reflected by a surface to the radiation reflected from an ideal diffuse scattering source (e.g., a Spectralon calibration panel) subject to the same illumination, in a specific viewing direction (Schaepman-Strub et al. 2006). These two quantities are numerically equivalent only for ideal Lambertian surfaces or, for non-Lambertian surfaces, under specific sun-sensor geometries. However, vegetation canopies—the focus of this study—are known to exhibit reflectance with very strong angular dependence, so a comparison of canopy hemispherical reflectance with a remotely sensed directional reflectance factor is invalid.

Our specific analysis is valid because—as described in the Methods—we used AVIRIS data that were also BRDF-corrected, whereby the directional reflectance estimates from the atmospheric correction process were further converted to estimates of hemispherical reflectance ("intrinsic surface albedo", sensu Lucht et al. 2000) via a polynomial approximation of the Ross-Li semiempirical BRDF model (Lucht et al. 2000). Another dataset that would have been valid for our analysis (albeit, one with much lower spatial and spectral resolution) is the MODIS albedo product (MOD43), which takes advantage of the angular sampling of the MODIS instrument to quantify the surface BRDF characteristics and therefore more precisely estimate the albedo (Wang et al. 2004; Schaaf & Wang 2015). However, our approach as described here would not be valid for the surface reflectance products produced by nadir-viewing instruments like Landsat, Sentinel-2, or most airborne platforms, at least without additional processing steps on the data, or, preferably, modification of the underlying radiative transfer models to allow for the prediction of directional reflectance. Fortunately, the assumptions and parameters that comprise two-stream radiative transfer models like EDR and its parent model (Sellers 1985) are readily adaptable to prediction of directional reflectance. For example, the SAIL model (Verhoef 1984)—which predicts both hemispherical and directional reflectance, and which has a long history of successful application to remote sensing—makes the same general assumptions as EDR and even shares many of the underlying coefficients (c.f., Yuan et al. 2017). Alternatively, land surface models can take advantage of recent advances in radiative transfer theory to improve their

accuracy without significant computational penalty (e.g., Hogan et al. 2018).

Singh A, Serbin SP, McNeil BE, Kingdon CC, Townsend PA. 2015. Imaging spectroscopy algorithms for mapping canopy foliar chemical and morphological traits and their uncertainties. Ecological Applications 25(8): 2180-2197.

Wang, Z., Zeng, X., Barlage, M., Dickinson, R. E., Gao, F., & Schaaf, C. B. (2004). Using MODIS BRDF and Albedo Data to Evaluate Global Model Land Surface Albedo. Journal of Hydrometeorology, 5(1), 3–14. https://doi.org/10.1175/1525-7541(2004)005<0003:UMBAAD>2.0.CO;2

Schaaf, C., Wang. (2015). MCD43A1 MODIS/Terra+Aqua BRDF/Albedo Model Parameters Daily L3 Global - 500m V006. NASA EOSDIS Land Processes DAAC. http://doi.org/10.5067/MODIS/MCD43A1.006

I propose the following modifications to address these issues:

a) Change the title to remove "cutting out the middleman."

We respectfully disagree. The "middleman" in the title refers to the modeling activities typically used to derive level 3 and 4 products such as GPP and LAI, and in targeting lower-level reflectance data products, we have successfully "cut this middleman out". Taking into account the significant re-framing of the discussion (see above response) in this revision, we feel this title is still accurate.

b) Modify the discussion around this point, including stating clearly that the reflectances and reflectance factors are physically different things and that, in general, it is not appropriate to do assimilate one into a model that predicts the other.

See earlier response.

> c) Quantify the differences between the BRFs and the modelled BHRs. I
> noticed, poking around in the github repository, that the SAIL model is in-
> cluded and hence, presumably, it is trivial to take representative posterior
> parameters values and model both BRF (from SAIL) and BHRs. If the au-
> thors' assertion that it shouldn't make any difference is correct then this
> will help to defend that. The observed reflectance factors should also be
> compared against SAIL predictions. I would be happy to iterate on the ex-
> perimental procedure with the authors.

We have added to the supplement some comparisons of simulations between EDR and
PRO4SAIL, parameterized identically (see attached figures, which will appear in the
revised supplement). For canopies with only one homogenous layer (the only canopies
PRO4SAIL can simulate) and solar geometry typical of our region (cos(Solar zenith
angle) = 0.85), EDR agrees very closely with PRO4SAIL directional reflectance over
a wide range of LAI values (0.1 to 5), but PRO4SAIL hemispherical reflectance is
higher than EDR. However, EDR is more sensitive to solar geometry than any of the
SAIL reflectance streams: When the sun is directly overhead (solar zenith angle = 0;
cos(Solar zenith angle) = 1), EDR predicts lower reflectance than even the darkest SAIL
bi-directional reflectance, but as solar angle increases, EDR predictions asymptotically
approach (but don't quite reach) the brightest SAIL bi-hemispherical reflectance.

Lower albedo predictions from EDR than SAIL can be attributed primarily to differences
in how each model defines the direct radiation backscatter coefficient in the radiative
transfer equation. A detailed description of the discrepancy is provided in Yuan et al.
(2017). Briefly, EDR (and the Sellers, 1985, model from which EDR is derived) define
direct radiation backscatter as a function of the single-scattering albedo, which in turn
is a challenging integral involving the leaf scattering phase function and leaf projected
area function. The Sellers (1985) analytical solution to this integral assumes a uniform

scattering phase function, which is appropriate for point scatterers but less so for horizontal surfaces like leaves. The practical consequence of this assumption is a lower value of the direct radiation backscatter and therefore a lower albedo; this is consistent with the results of our sensitivity analysis above. Meanwhile, the SAIL definition—and the one Yuan et al. (2017) recommend for land surface modeling applications—is a more simple function of leaf scattering, leaf angle distribution, and canopy optical depth that also produces a more accurate result.

The underestimation of albedo described above may also help explain some of the calibration issues identified by reviewer 1. Specifically, this may at least partially explain the tendency of our EDR calibration to prefer higher effective LAI values (in particular, the tendency towards higher values of leaf biomass allometry and clumping factor for some PFTs) and more horizontal leaf distributions (i.e., higher leaf orientation factor).

Given that underestimating albedo can have significant consequences for the biological, ecological, and physical predictions of ED2 (c.f., Viskari et al. 2019), incorporating this fix into the ED2 canopy RTM is an important future direction of our work. However, this activity is outside the scope of this work because doing so would require propagating the different coefficients through the complex, multiple-canopy-layer solution of EDR—a non-trivial task.

The reason that we ultimately did not include SAIL simulations is that there is no obvious way (at least, to us) to convert the complex, multi-PFT canopies simulated by EDR into the single homogeneous inputs expected by SAIL. Even if we performed an analogous multi-site calibration of PRO4SAIL, the only calibrated parameters we could compare directly would be the total LAI (and even that wouldn't be a fully apples-to-apples comparison because a PRO4SAIL would optimize that parameter directly for each site, whereas in EDR, we are targeting multiple parameters that interact to give LAI)—for the remaining parameters, we would still have to aggregate the multi-PFT EDR parameters in a non-trivial, non-obvious way.

We have revised our discussion to incorporate all of the information above.

Yuan, H., Dai, Y., Dickinson, R. E., Pinty, B., Shangguan, W., Zhang, S., et al. (2017). Reexamination and further development of two-stream canopy radiative transfer models for global land modeling. Journal of Advances in Modeling Earth Systems, 9(1), 113–129. https://doi.org/10.1002/2016MS000773

> 2. Is ED2 actually used in this paper? It seems that it is only a relatively small component of the model (i.e. the canopy radiative transfer scheme) that has been extracted. I think the title is slightly misleading in this respect. More important, I am not sure calling the code ED2-PROSPECT5 is appropriate. Perhaps EDR-PROSPECT5 would be better?

No, ED2 is not used directly in this analysis. However, the results of this analysis can be used directly to parameterize ED2. The R version of the ED canopy radiative transfer model is mathematically identical to the actual Fortran version and was only extracted for analytical convenience and computationally efficiency (since running the original version of ED2, which is what we initially implemented, incurred significant computational overhead for initializing ED2's many variables).

We have kept the title as-is, but have added additional text to the discussion highlighting how this approach could be used directly with the ED2 model.

> 3. How are correlations between spectral channels dealt with? Do the authors use all of the AVRIS observations in the domain 400-1300nm? The errors in spectrally adjacent bands will be very highly correlated and the overall information content will be much lower the same number of independent observations.

We did not account for correlations across spectral channels, but we agree that they are significant (especially for hyperspectral data) and could therefore influence the results. To address this issue, we have rerun our analysis using a multivariate normal likelihood with an AR1 autocorrelation matrix to capture the covariance between bands. To avoid introducing new uncertainty into the inversion, we fixed the correlation coefficient (rho) of this model to the average of the lag-1 autocorrelation calculated from the residuals from our current results. In our revision, we added text describing this process to the methods and updated all of our results figures.

Minor comments:

L27 The MacBean et al. (2018) paper referenced here does not assimilate a derived product in the sense the authors mean it. The assimilated data in that paper is solar induced fluorescence which, whilst it is "derived" in the sense that it is not what is being measured by the sensor, is no more derived that the surface reflectance data used in the manuscript under review. I suggest finding another reference here.

We have revised this accordingly.

L35 "Meanwhile, the estimating" -> "Meanwhile, estimating"

We have revised this accordingly.

L61 Surface reflectance is not assimilated in Zobbitz et al., (2014). The title is misleading (and I regret not standing my ground on that issue when we submitted that paper!); in fact fAPAR data is assimilated.

We have revised this accordingly.

L85 Not sure this line should finish with a colon.

We have revised this accordingly.

Eqn 3 What is meant by $r_{n+1}$ (and other variables with that subscript)? If I have understood correctly this is there are $n$ layers, so what is the reflectance of the $n+1$ layer? (I am sure I have just missed something here, but it was not obvious).

EDR models fluxes not at canopy layers, but at the boundaries between layers. Boundary $i$ is defined as the air space immediately below the i$^{th}$ canopy layer (counting from the bottom of the canopy). For $n$ canopy layers, $i = n$ refers to the space immediately below the tallest cohort layer, while $i = n + 1$ is the (imaginary) boundary above the tallest cohort (i.e., between the canopy and the atmosphere). Therefore, $r_{n+1}$ refers to the reflectance at the top of the canopy. We have added this clarification to the methods.

L102 "tau" should presumably be the Greek symbol $\tau$.

We have revised this accordingly.

Eqn 7 I am confused by this, why is forward scattering defined by the sum of R+T? This is just the total scattering isn't it?

We agree that this is just total scattering. We have revised this sentence to say "scattering (nu) and backscattering (omega) of canopy elements...".

L188 This a different definition of X from earlier in the paper. I suggest finding a different symbol.

We have replaced the earlier definition of X (in the EDR matrix equation MX = Y) with the symbol A to eliminate this ambiguity.

> L273 "DALEC-predicted foliar biomass, which required introducing an additional fixed parameter (grams of leaf carbon per leaf area) present in neither model." This statement is incorrect – that parameter already existed DALEC.

We have removed this sentence from our discussion.

Please also note the supplement to this comment:
https://gmd.copernicus.org/preprints/gmd-2020-324/gmd-2020-324-AC2-supplement.pdf

**Supplement:**

**Supplementary figures for reviewer T. Quaife**

February 5, 2021

[Figure]

Figure 1: Comparison of EDR and PRO4SAIL simulations for single-cohort canopies over a range of leaf area indices (LAI). These simulations assume common PROSPECT parameters, and sun-sensor geometries (nadir sensor, cosine solar zenith angle of 0.85). For EDR, we use a single cohort with the prescribed LAI. "SAIL: HR" is the "hemispherical reflectance" ("blue-sky albedo"), calculated (as in EDR) as the average of directional-hemispherical (DHR) and bi-hemispherical (BHR) reflectance streams weighted by the direct sky fraction (0.9; same value as in EDR). Similarly, "SAIL: DR" is the "directional reflectance", calculated analogously from the bi-directional (BDR) and hemispherical-directional (HDR) reflectance streams. Individual SAIL fluxes are shown with dotted lines. Here, cos(solar zenith angle) is 0.85 – a typical value for summer in our study domain.

[Figure]

Figure 2: (New supplementary figure) Same as above, but over a range of solar geometries (czen = cosine of solar zenith angle). All simulations use a common LAI of 3.

---

## Author Response (AR1)

**Response to reviewers**

**March 4, 2021**

We are pleased to present a significantly revised version of our manuscript titled "Cutting out the middleman: Calibrating and validating a dynamic vegetation model (ED2-PROSPECT5) using remotely sensed surface reflectance" for consideration for publication in *Geoscientific Model Development*. We are very grateful to the two reviewers for their constructive feedback on our analysis, and we firmly believe addressing their suggestions has made our manuscript significantly better. In Section R1, we provide a summary of the most significant changes in this revision. We provide a point-by-point response to each reviewer's comments in Sections R2 and R3. All line numbers refer to the highlighted changes version of the manuscript.

**R1 Summary of revisions**

**R1.1 Revisions to the inversion algorithm**

Reviewer T. Quaife remarked that we did not mention how our inversion algorithm accounted for correlations between closely-spaced hyperspectral channels. In our original submission, we did not account for this spectral autocorrelation, and our likelihood expression assumed all bands were independent. However, we recognize that this was flawed. In our revision, we have updated our inversion algorithm to account for autocorrelation in the residuals by imposing an order-1 autogregressive (AR1) correlation structure on the residual covariance matrix in our likelihood expression (calculated exogenously, with the AR1 correlation coefficient calculated from the residuals from the original simulation). This is described in our revised methods (lines 329–337).

While implementing this, we made several other changes to our inversion algorithm as well (all documented in the revised methods). For one, rather than prescribing a common set of illumination conditions (direct vs. diffuse radiation and solar zenith angle) for all observations, we now fully account for variability in both across all observations by extracting the solar geometry from the AVIRIS metadata and prescribing radiation conditions (direct-diffuse fraction) based on the hourly MERRA-2 meteorological reanalysis product (lines 307–310). In addition, we have replaced our original lognormal likelihood penalty for LAI predictions with a uniform distribution that guarantees that LAI predictions do not exceed a reasonable threshold but also does not differentiate between LAI values below that threshold (lines 345–348). We have updated all of our results figures and text accordingly. The impact of these methodological changes on our results was relatively minor, and none of the core conclusions have changed.

**R1.2 Discussion of reflectance fluxes**

Reviewer T. Quaife remarked that our original manuscript incorrectly downplayed the important differences between directional reflectance factor (which is observed by most optical remote sensing instruments) and hemispherical reflectance (a.k.a., "albedo"; the quantity simulated by EDR and related models). We fully agree that these are two distinct (albeit related) physical quantities that, in general, cannot be compared. We also recognize that our original discussion of this topic was flawed and, as stated, misleading. To address this comment, we have completely rewritten our discussion of reflectance fluxes, taking care to emphasize that surface reflectance factors estimated by AVIRIS and similar sensors cannot be directly compared to hemispherical reflectance simulations from EDR and similar models without additional processing steps (lines 564–594).

That said, because we were using AVIRIS classic imagery that underwent additional processing for the NASA Forest Functional Types (FFT) project (Singh et al., 2015), we feel that our analysis is still valid. In addition to the standard atmospheric correction and orthorectification conducted by NASA JPL, the AVIRIS data we used for model calibration were also cross-track illumination corrected, as well as BRDF-corrected, following the procedure of Lucht et al. (2000). Briefly, the latter approach estimates "intrinsic surface albedo" — the precise quantity that is simulated by EDR — from angular reflectance data through application of a polynomial approximation to the Ross-Li semiempirical BRDF model. The full AVIRIS processing pipeline for the AVIRIS data (including the BRDF approximation) we used is described in Singh et al. (2015). This information was absent in our original submission, but now appears in our expanded "Site and data description" section (lines 293–302).

**R1.3   Structural issues in EDR**

Reviewer 1 noted that several of our parameters exhibited edge-hitting behavior and suggested that we discuss this behavior and the extent to which it might be symptomatic of structural issues with EDR. At the same time, Reviewer T. Quaife suggested that we compare EDR and PRO4SAIL — another two-stream canopy radiative transfer model popular in the remote sensing community due to its ability to predict both directional and hemispherical reflectance fluxes — to explore the differences between hemispherical and directional reflectance. In our revision, we added sensitivity analyses of EDR to some of its parameters (Figures A1, A2, and A3), and a comparison of EDR and equivalently parameterized PRO4SAIL (Figures 10 and A16).

An important takeaway from this analysis is that, compared to PRO4SAIL, EDR systematically underpredicts hemispherical reflectance. This can be attributed to a difference in how EDR and SAIL define the direct radiation backscatter coefficient (see also discussion in Yuan et al. 2017). This underestimation of hemispherical reflectance by EDR may help explain some of the calibration issues identified by Reviewer 1. Specifically, both increasing canopy clumping (which increases effective leaf area index) and increasing the leaf orientation factor (more horizontal leaves) increase the simulated canopy albedo, so pushing these parameters up against their physical upper bounds may be a way for our inversion to compensate for EDR's underestimation of albedo. We have included a discussion of this in our revised Discussion section (lines 544–557).

**R1.4   Discussion of equifinality**

Reviewer 1 noted that, while we mentioned in our Introduction the equifinality between leaf biochemistry and canopy structure in canopy radiative transfer, we did not revisit this important issue in in the rest of our manuscript. We agree that the topic of equifinality in canopy radiative transfer is worth additional discussion. In our analysis, equifinality was reduced by using informative priors for the leaf parameters. We note in our revised discussion that, in future work, this prior information could also come from other kinds of remote sensing measurements (lines 512–514) as well as through the ecophysiological mechanisms embedded in the model itself — i.e., some combinations of model parameters and states that are consistent with a given surface reflectance may be excluded because they imply ecologically unrealistic states in previous time steps or result in ecologically unrealistic outcomes in future time steps (lines 475–482).

**R1.5   Revised model description**

In the process of revising our manuscript (and particularly, of investigating the discrepancy between EDR and PRO4SAIL), we realized that while we correctly described canopy radiative transfer coefficients in EDR (e.g., inverse optical depth; leaf angle distribution functions) and correctly implemented EDR in our code, we did not correctly describe the multi-layer canopy radiative transfer solution actually used by EDR. In addition, we felt that our model description was not well-organized. Therefore, in this revision, we have completely re-written the Model Description to true to what the EDR code is actually doing, and, in our opinion, to be more clear (Section 2.1). Importantly, we now split this section into two sub-sections: The first subsection describes the radiative transfer coefficients and their derivations (which should make it easier to cross-reference against the descriptions of its parent model — Sellers 1985 — and other radiative transfer

models). The second subsection describes EDR's unique solution to these coefficients for a multi-layered heterogeneous canopy.

**R2   Reviewer 1**

General Comments (overall quality): This manuscript uses remotely-sensed surface reflectance observations to calibrate/constrain a series of ecosystem parameters from the ED2-PROSPECT (EDR) model characterizing the land surface at 54 forested sites related to leaf biochemistry, canopy radiative transfer, and soil characteristics. An important innovation is the introduction of the radiative transfer model PROSPECT to the biosphere model ED2 to provide an improved spectrally resolved simulation of surface reflectance. This is done, in part, to bring the observed surface reflectance closer to what the model actually predicts, helping to reduce the impact of observational uncertainty as well as more effectively constraining multiple components of the model. The authors find that through the assimilation of surface reflectance EDR can provide better simulations of the surface reflectance spectra and leaf area. The findings suggest that this approach could be used to better constrain surface energy balance as well as overall ecosystem functioning. Given many other ecosystem models include a two-stream radiative approach they contend their results should be widely applicable.

Scientific Questions/Issues: The authors bring up the issue of equifinality in the introduction, which presents a challenge for this application in that the surface reflectance can be a function of leaf biochemical properties, leaf structure, and canopy and soil radiative transfer characteristics. To some degree, equifinality was reduced in that the prior parameter distributions for biochemical leaf properties were tightly restricted, and limited to the extent the surface reflectance observations could influence them. In contrast the canopy radiative transfer parameters and LAI (through SLA) were simultaneously being adjusted by the optimization. Was hoping the authors could comment more about how equifinality of the surface reflectance influenced their results.

Please see Section R1.5 in our main response.

As a follow up question to the equifinality question above – ED2 is a dynamic vegetation model with the ability to simulate competition amongst cohorts thus providing a simulation of co-existing dominant PFTs. It wasn't clear how well the simulation of cohort competition influenced the final distribution of PFTs and to what extent this matched the site level observed vegetation state. Given that the parameter optimization was PFT specific, the precise vegetation PFT distribution could have a large impact. Was the PFT distribution prescribed?

In this study, the vegetation composition at each site (including the PFT distribution and size-age structure) was prescribed in detail based on data from the NASA Forest Forest Functional Types (FFT) field campaign. In our revision, we have clarified this point in the "Site and data description" (lines 287–289).

Was there any attempt to withhold some site reflectance data and apply the calibrated model parameters at those sites? It seems the optimized surface reflectance simulations were performed at sites that were already calibrated. The fact that the authors performed an across-site joint assimilation may in part account for this, but was interested how the calibrated parameters would perform at sites outside the calibration sites.

We acknowledge that cross-validation or out-of-sample validation are useful tests of model performance, and in our revised discussion, we recommend these activities as future directions for this work. However as the reviewer points out, because our calibration was joint across all sites, we did not feel that a separate validation at other sites not used in the calibration was necessary. With 54 sites in our calibration, any single site represents $< 2\%$ of the data, and for a joint calibration without site random effects, we have every reason to believe that the calibration is not overfitting to any individual site; trying to fit any one site well would cause others to do worse (especially given the large observed variability in forest structure) unless the EDR model structure was reasonable and the parameters chosen were genuinely good choices. We have added this information to our revised discussion (lines 483–489).

It is known that radiative transfer models are challenged in simulating evergreen species in part because of

the irregular and open space canopy structure. Many of the figures in the supplement demonstrate stronger biases in simulated surface reflectance exist for evergreen sites as compared to deciduous. Was hoping the authors could comment on this, and recommendations for getting around this.

The reviewer brings up a great point about conifer canopies historically being harder to capture. Although some conifer-dominated sites did demonstrate significant biases in reflectance predictions, our analysis of reflectance bias by composition, structure, and spectral region (Figures 4, 5, A6–A15) shows that these biases are not systematic (though they may drive greater predictive variance). In our revision, we describe this analysis in more detail in the Methods (lines 372–378) and Results (lines 424–434).

Whereas leaf level biochemistry related parameters were well constrained by the priors, many of the radiative transfer posterior parameters seemed to be edge hitting parameters. To what extent do the authors believe this behavior was caused from structural error in canopy radiation transfer and/or mismatch caused in part from radiation directionality differences between the simulated and observed canopy reflectance? The authors devote a considerable amount of general discussion regarding this topic, but don't directly address how this might have effected their own results.

Please see Section R1.4.

The manuscript begins by justifying the inclusion of the PROSPECT model in to ED2 to bring the model closer to the observations, to, in part help reduce the uncertainty of the observation that are assimilated into EDR. More explanation on how the surface reflectance observational uncertainty was quantified here, and what it represents, and to what extent overconfidence in uncertainty may have led to the posterior edge hitting parameters.

As shown in equation 48, observation error in the reflectance data was not estimated *a priori* based on the instrument itself, but was modeled as the residual error between the model and the data, analogous to what is done for any linear or nonlinear regression model. A key difference, however, is that the error model accounts for the known heteroskedasticity in spectral data (i.e., the size of the variance increases with the magnitude of the reflectance). In terms of random spectral errors, there is no reason to expect this variance to be overconfident for inferences made on these landscapes, especially as the study sites were not all imaged on the same day under the same atmospheric conditions, though we'd agree that one might not want to apply this variance to entirely different ecoregions. Furthermore, because the variance slope and intercept are fit parameters, whose parametric uncertainty is being quantified and propagated, this makes it even less likely that our uncertainty estimate is overconfident. That said, the current approach does not formally account for any possible systematic errors in the observations, which could have a more serious impact on inferences. However, we would note that we are unaware of any derived data products that account for these systematic errors either. Furthermore, in addition to the same uncertainties about reflectance that we face, those products additionally contain numerous uncertainties about model structure, parameters, and covariate data whose uncertainties are rarely fully propagated, meaning that the alternative approach (calibration to derived data products) is more likely to result in overconfidence in uncertainties than the approach taken here.

That said, we feel it is highly unlikely that overconfidence in surface reflectance estimates contributed to the edge-hitting behavior of some of our posteriors. Random errors would not result in a systematic parameter bias and, given the long history of the instrument and maturity of atmospheric correction approaches, any systematic errors in AVIRIS are likely to be quite small relative to the structural uncertainty in EDR (i.e., biologically implausible parameters should not be necessary to capture observational data biases of the magnitude likely to be present).

Detailed Comments: Abstract and manuscript in general: Need more discussion on what we hope to gain by this. We don't really care about surface reflectance (although energy balance is important), but we do care about how LAI, chlorophyll, pigments and water status influence ecosystem functioning through carbon and water exchange. I think this needs to be emphasized more, and provide evidence that this sort of setup can accomplish this.

We agree that additional emphasis on the implications of this work is warranted. In our revisions, we emphasize several important implications in the abstract (lines 1–7) and introduction (lines 74–93).

First, as you say, energy balance is important, and the contribution of vegetation to land surface albedo is a critical mechanism by which vegetation influences regional and global climate (Bonan, 2008). Therefore, ensuring the accuracy of model simulations of albedo, including its sensitivity to vegetation structure and composition, is essential to accurately projecting the effects of climate- and land use-driven changes to terrestrial ecosystems on future climate.

Second, canopy radiative transfer directly affects many physiological, ecological, and physical processes included in complex demographically-enabled vegetation models like ED2 (Viskari et al., 2019). Light availability and absorption is a first-order control on photosynthesis, and ability to survive under different light levels is an essential component of a tree species' position in forest succession. Meanwhile, temperature—which is strongly influenced by albedo—directly affects the rates of both enzyme-kinetic physiological processes and evaporation.

Finally, ED2 and similar models are highly sensitive to many of the leaf traits constrained by this analysis (Raczka et al., 2018; Dietze et al., 2014; Shiklomanov et al., 2020). Given the large variability of these traits through space and time, remote sensing is an essential data source for model parameterization, and our work provides a useful approach for doing so.

Line 1: Remove 'derived'. The fact that they are 'data products' and not 'observations' gets across the point.

While we agree that "data products" should imply a difference from observations, treatment of remote sensing data products as true observations without accounting for uncertainties or biases is widespread in the Earth science community. Therefore, we think it is important to emphasize that these products are derived.

Line 5: 'compared against airborne and satellite data' Technically, this is still data and not observations in that even reflectance data requires RTM models, I believe. But it is more direct relationship

We agree that additional nuance is required here (especially when also considering reviewer T. Quaife's comments). Therefore, we have revised this section to capture the fact that although these data are still derived (processing steps include atmospheric correction and orthorectification), they rely on fewer assumptions (especially about the land surface), have fewer processing steps, and therefore are closer to observations (lines 61–64).

Line 23: add 'to' calibrate or constrain

We have revised this accordingly.

Line 25: I know exactly what you mean by 'constrain', but could you use 'calibrate' or 'inform' in this context?

We have replaced "constrain" with "inform".

Line 32: "More sophisticated approaches for estimating vegetation properties based on physically-based radiative transfer models face issues of equifinality, whereby many different combinations of vegetation and soil properties can ultimately produce the same modeled surface reflectance (Combal et al., 2003; Lewis and 35 Disney, 2007)." This is important I think – and raises a key point for this analysis – is there not equifinality when trying to constrain leaf structure vs. leaf status? I would think equifinality could be a problem here, and I think you need to acknowledge this and how you might address this – Strong priors? Demonstration that surface reflectance can tease apart these two things. . .

Please see Section R1.4.

Line 52: Awkward topic sentence. Simplify "Some land surface models already include there own . . . that allowd for a more direct comparison to remotely sensed surface reflectance."

We have revised this sentence according to the reviewer's suggestion.

Line 57:"Canopy radiative transfer plays a particularly important role in the current generation of demographically-enabled dynamic vegetation models, where differences in canopy radiative transfer representations and parametrizations have major impacts on predicted community composition and biogeochemistry

(Loew et al., 2014; Fisher et al., 2018; Viskari et al., 2019)." Seems weird to word it this way. Isn't it the other way around, community composition and biogeochemistry impact the RTM?

It is true that composition and structure impacts the RTM, but all of these studies show that the opposite is true as well! These studies illustrate that because the RTM determines the overall energy balance of the ecosystem and the distribution of light within the canopy, RTM formulation and parameters profoundly impact the predicted biogeochemical fluxes and vegetation dynamics. For example, Viskari et al. (2019) show that uncertainties in canopy RT can cascade and impact a number of associated processes, including photosynthesis, energy balance, internal competition, and demography. We have revised the sentences here to more clearly and explicitly convey this idea (line 74–90).

Line 72: "...will significantly constrain model parameters related to canopy structure." So the goal all along was to constrain canopy structure with surface reflectance, not necessarily foliar biochemistry? Maybe talk a bit more about the differences in sensitivity of surface reflectance to canopy structure vs foliar biochemistry.

Our objective was to evaluate which parameters could be constrained. The list of candidate parameters included parameters related to both structure and biochemistry. However, we hypothesized that, because of the informative priors on foliar biochemistry, the constraint would be relatively greater for canopy structural parameters (as stated here).

Per our earlier responses (Section R1.3), we have included an additional parameter sensitivity analysis of EDR and additional discussion of equifinality.

Section 2.3: Can you provide a sense of scale? For example for the 54 sites, what spatial range was the inventory data taken, and what spatial resolution did AVIRIS cover? Trying to get a feel for spatial mismatch, etc. Were sites chosen because they were rather homogeneous for certain PFTs?

In response to this and Reviewer T. Quaife's comments, we have elaborated on the methods behind the data used in this analysis. Specifically, each of the 54 sites here consisted of a $60 \times 60$ m transect within which forest inventory data were collected. AVIRIS-Classic data were extracted as the average of a $3 \times 3$ pixel array (each pixel is $\tilde{1}5$–20m, depending on aircraft altitude) centered on the site transect center, resulting in a single composite spectrum for the $60 \times 60$ m area. Additional details on the sampling methodology are described in Singh et al. (2015) and references therein.

Figure 1: Was a little surprised to see many sites so close to Lake Superior. No issues with interference from nearby water reflectance?

Although sites do appear very close to Lake Superior in the map we provide, all sites are sufficiently inland (several kilometers) that contributions from water reflectance of nearby pixels can safely be assumed to be negligible. In the revision, this figure has been broken up into two figures (Figures 1 and 2).

Line 200: Could you provide a bit more explanation of what including the EDR predicted LAI term within your probability function accomplishes? EDR response becomes saturated to LAI so is this an artificial way to account for increased reflectance?

Yes, the LAI penalty in the likelihood function accounts for the saturating effect of increasing LAI on reflectance. This effect is not unique to EDR—rather, it is a well-known consequence of the exponential extinction of light through a medium, following Beer's Law. Therefore, a canopy with an unrealistically high LAI like 15 has virtually the same reflectance as a canopy with a high but more feasible LAI like 6 (all else being equal), and therefore a likelihood calculation that does not penalize excessively high LAI values would consider both outcomes equally likely.

We have added additional text to this effect to the methods (lines 340–347), and have added the attached figure to the supplement demonstrating the saturating effect of LAI on reflectance (Figure A1). However, as noted in Section R1.1, in our revision, we have changed this penalty from a lognormal distribution to a uniform distribution between 0 and 10.

Line 227: So, to evaluate the model you compared the EDR-spectra against the AVIRIS observations at sites that were used to calibrate the model? Was there any attempt to withhold some site data and apply the calibrated model at those sites?

As we stated above, because our calibration was joint across all sites, we did not feel that a separate validation at other sites not used in the calibration was necessary.

Line 232: Can you quantify what 'informative' means

We have clarified this sentence to say these are "leaf parameters whose prior distributions were already independently constrained by an earlier analysis". These prior distributions are shown in Figure 3.

Line 233: I cannot see in figure where PROSPECT N parameter is?

In our experience, labelling PROSPECT's "N" parameter as such is confusing to readers unfamiliar with PROSPECT because it suggests leaf nitrogen content. Therefore, we prefer the more explicit name "# mesophyll layers". We have revised the methods and results text in a few places to clarify this.

Section 3 Results: Although Figure 2 was very informative, I found the Results section in general, relatively vague, perhaps some sense of % reductions in 95% credible interval.

We have revised the results (lines 380–388) to include more precise statistics, including the suggested relative reductions in the width of the 95% credible interval.

Figure 2: Hopefully, the authors comment on some of the apparent edge-hitting parameters specifically related to canopy RTM parameters such as leaf orientation, canopy clumping and water. I worry that the information from the relatively strong and defensible leaf biochemistry prior parameters leading to relatively self contained and PFT differentiated posteriors for the leaf biochemistry parameters is lost or made irrelevant due to biases between model simulated and observed surface reflectance that are corrected by 'fitting' the RTM parameters. The context of this manuscript doesn't indicate how sensitive the surface reflectance is for the suite of parameters calibrated here... perhaps included in one of cited manuscripts.

Please see our response in Section R1.3.

Figure 3: A map would be helpful with site codes provided. Perhaps a zoomed in map that demonstrates where these sites are spatially? Also is the stem diameter plot on the right simulated or observed? In fact, doesn't that have a large impact on the assimilation, the PFT distribution and stem diameter distribution?— has it been demonstrated that ED2 can properly simulate the competition of PFTs at the site providing the correct vegetation state, such that the parameter optimization reflects the observed vegetation state ?? Or has the vegetation state been prescribed in this case?

We have split up the original figure into two separate figures: one showing a larger map with sites labelled (Figure 1) and one showing the density vs. diameter plot (Figure 2).

As stated in the main response, the vegetation composition at each site (including the PFT distribution and size-age structure) was prescribed in detail based on NASA Forest Functional Types (FFT) campaign field data. We have clarified that these are the observed stand structures used as prescribed EDR inputs in the methods (lines 287–289).

Figure 5: "The observed vs. predicted line had a slope of 0.37 and an intercept of 2.80, indicating that EDR calibration underpredicted LAI on average but overexagerrated across-site LAI variability." What do you attribute this clear structure in residuals between observed-simulated LAI?

We agree the clear mismatch between predicted and observed LAI was not given sufficient attention in our original draft. In our revision, we now devote more space to this in the Results (lines 437–441) and Discussion (lines 533–542).

Line 258-270: I like the overview explanation of bringing observations closer to models or alternatively bring models closers to the observations. In the end, it's a bit of semantics, especially for using information from satellites we will always need some sort of transfer function or forward operator to convert from what a satellite observes and what a model predicts. I don't think one way or the other should take precident. The advantage in your approach, however, is the potential for the observed surface reflectance to constrain multiple model components, whether that be leaf structure or biochemistry or water status. I think that is potentially extremely valuable although I am not sure it has been demonstrated, yet, that this is the case. I feel more could be 'learned' about what information surface reflectance could provide if you could

prescribe the LAI and PFT-distribution in ED2, then you could really get a grasp on what it can inform, leaf biochemistry? Within-canopy RTM parameters? Etc.

We disagree that the difference between bringing models closer to observations and vice-versa is a purely semantic one—the fact that all observations need some transformation does not mean that all transformations are the same. Instead, we argue that comparing model output to a highly derived data product (e.g., MODIS GPP) is closer to a model intercomparison than a model validation, since generating those data products invariably requires their own models (whether statistical or process-based) that make specific assumptions about different processes, often very different assumptions than the model the products are compared with. Thus, it is often really an "apples to oranges" comparison. In other words, in addition to the *shared* observational uncertainties (e.g., atmospheric correction, instrument calibration) that are present in both approaches, derived data products include numerous additional assumptions and uncertainties that are avoided in a spectra-to-spectra comparison. Meanwhile, as you point out, the approach of bringing models closer to observations is advantageous precisely because of the model's emergent covariance across many disparate variables rooted in specific assumptions about biophysical and ecological processes. In practice, for an observation operator that is almost entirely disconnected from the model (for example, if we just took the total LAI from ED2 and used a standalone RTM like SAIL to simulate the reflectance), we agree that there's relatively little advantage relative to a derived product, but the more tightly an observation operator is integrated into a model (as in our analysis, where canopy RTM parameters and outputs are used elsewhere in ED2), the greater its value.

To address this, we have revised the discussion to discuss the relative advantage of building models with remote sensing-friendly observation operators, especially when such operators are tightly integrated with other processes in the model (lines 475–482). For the latter, we now discuss some future directions for ED2 and optical remote sensing specifically, such as using the PROSPECT leaf model to couple the model's representations of plant hydraulics (for leaf water content) to canopy radiative transfer. Similar approaches that leverage the model's internal RTM can also be applied to other remotely-sensed data, such as lidar (canopy structure), radar/microwave (structure and water content), thermal (canopy traits and water use), and fluorescence (photosynthetic traits). Indeed, this paper is an important first step towards the ultimate goal of leveraging internally-consistent process-based models to make joint inferences across multiple remotely-sensed data constraints simultaneously.

But, as you brought up in the introduction, this brings up equifinality issues. Not so much in this case for your leaf biochemistry parameters because of strong priors, but it does seem to be the case for canopy RTM parameters and predicted LAI (SLA). I think you need to caveat or address this concern.

Please see Section R1.4.

Lines 286-310: It seems like you are pointing out sources of structural error within the radiative transfer of EDR or, to the extent, mismatch between what EDR simulates vs. what AVIRIS-Classic observes. Could this help explain why the calibration caused some posterior RTM parameters to be edge-hitting against the bounds of the priors? Also, this work was in part motivated by being better able to quantify the uncertainty in the surface reflection observations, but I am not sure I saw a clear explanation of the uncertainty that was used or provided for the AVIRIS and how was this quantified. It seems parameters have the potential to be overfit, if the observation uncertainty is not realistically quantified. May have missed this.

Please see Section R1.3

Line 320: Really it's the power to upscale that remote sensing products provide. However, this comes at a cost, they 'observe' reflected radiation from the land surface which indirectly characterizes things that we care about like, like leaf biochemistry, albedo etc.

We agree the reflectance data collected by passive optical remote sensing are affected by many different surface features and processes, often in confounding ways, which makes them challenging to use in isolation. Fully leveraging the power of remote sensing requires additional sources of information from both in situ data and the understanding of biophysical and ecological processes embedded in our vegetation models. We have revised this text and expanded this text accordingly.

Line 326: 'accurately reproduce surface reflectance and leaf area index' I think this is a bit of an overstatement, especially because the figures demonstrate systematic mismatch between the optimized surface reflectance and observed reflectance (figure 3), and strong residual error structure in LAI (Figure 5). Perhaps this approach provided 'improved' reflectance and LAI is better wording.

We agree that we overstated our LAI prediction accuracy here, and have revised this text to temper our conclusions accordingly. However, as we pointed out in earlier comments, although our reflectance predictions were not perfect everywhere, the quantitative analysis of surface reflectance revealed no systematic biases by PFT or stem density.

Line 330: I think you also need to say where this work can lead — this is of interest for those that are concerned with ecosystem functioning and that this could provide improved estimates of both biomass and land-atmosphere carbon and water exchange. Also some discussion of the differences between evergreen and deciduous forests would be helpful. Generally RTM's have more difficulty with more open canopy evergreen species and not really discussed in this manuscript.

Our revised discussion (see earlier comments) and conclusions (lines 632–634) include more text on ways that ED2 (and other vegetation models) can be further enhanced to better take advantage of passive optical and other remote sensing techniques.

Appendix:
Figure A2: What would be really helpful is to use the same color-coding in Figure A1 to show deciduous vs evergreen sites. Also this brings about the question – why did you choose the sites that you did to put in the manuscript itself?

Figures A5-A13 provide the requested breakdown of plots by PFT composition. For the main figure, we selected sites that, collectively, span the geographic, stand-structure, and PFT composition space of our study. We have added this information to the figure caption, highlighted these plots in our site map, and mention this in the revised Results section.

Figure A3: This sort of gets at the hardwood vs conifer performance as well. I think it would be helpful to comment on this distinction in performance within the results/discussion.

As mentioned above, we have added additional text about our analysis of performance by PFT to the methods, results, and discussion. As this figure clearly shows, there is no systematic bias in reflectance simulations for conifer species.

Figure A13: This is also a very compelling figure that gets at the increased bias in spectra for conifer. Worth discussing in main manuscript.

See earlier comments about this. We note that even this figure shows no systematic bias—three sites have underpredicted reflectance (AK06, AK60, OFO1), two sites have overpredicted reflectance (MN02, MN04, BH02), and the remaining 2 sites have relatively accurate reflectance predictions (SF01, SF04).

**R3   Reviewer 2 (T. Quaife)**

The manuscript "Cutting out the middleman: Calibrating and validating a dynamic vegetation model (ED2-PROSPECT5) using remotely sensed surface reflectance" by Shiklomanov et al. illustrates the assimilation of reflectance data observed from aircraft into a component of the ED2 model. In many respects this is an excellent paper. The Bayesian approach taken is state-of-the-art and the use of forward modelling of reflectance for assimilation purposes is desirable for many reasons (and yet surprisingly little progress has been made in this area for land surface studies, making this paper especially welcome). The text is also well written on the whole.

Unfortunately I do have one rather significant concern about this paper, which may at first seem like a subtlety, but really is not. The authors are not "cutting out the middleman" so much as choosing to ignore them. I am concerned that the take home message of the paper for people less familiar with the underlying physics will be that it's OK to take this approach.

1. Assimilation of BRF and HDRFs.

[Note: I am using the definitions of reflectance quantities from Schaepman-Strub et al. (2006; hereafter S06), which the authors have also done.]

My major concern with this paper is the authors' misuse of different reflectance quantities. They are not comparing like with like and I do not agree that it is OK to assimilate quantities that are not physically consistent with those being modelled. The Sellers two-stream model predicts reflectances (BHR and DHR), whereas the AVRIS data are reflectance factors (BRF and HDRF). Although they are both unitless they are fundamentally different quantities and have different scales.

The authors do devote a paragraph to discussing this, but it is misleading and I am not convinced by the arguments they make. The statement that the ED radiative transfer model predicts BHR is only partly correct. It also simulates DHR and the predicted reflectance is a weighted mean of the two. The authors go on to state the AVRIS observations are most related to HDRFs. This would only be true for overcast skies. AVRIS observations are best represented as a mix of HDRFs and BRFs (although the reality is more complex of course as the down-welling diffuse flux is rarely isotropic). I argue that for most cases the AVRIS data will be most closely related to BRFs as most flight campaigns are under relatively clear skies.

I think part of the problem here arises because S06 define anything with more than 0

Using the Hogan et al (2018) paper to defend this position is disingenuous. One of the things that paper shows quite clearly is that solar geometry effects are not well dealt with by classic two-stream formulations for complex canopies. The adjustments to the two-stream scheme made in that paper are required to make the model predicted DHR match the reference 3D model.

I am prepared to accept that in the specific experiment in this paper the limited angular sampling of AVRIS may mean that the overall effect will be small. However, when the authors make statements such as the one at Line 300 ("We therefore conclude that additional computational and conceptual challenges (as well as parameter uncertainties) associated with treatment of angular effects in similar models are unwarranted") it is very misleading: the take home message is that it is, in general, OK to take this approach. It is most definitely not.

On a related note I also don't accept the statements on lines 269 and 299 that seem to imply that it's necessary to have additional parameters to predict directional reflectances. This is not the case and hence also not a valid defence of the approach taken. The set of parameters required to define a two stream model can be used also to define a BRDF model derived from the same underlying assumptions (i.e. semi-infinite, plane parallel turbid media with point scatterers).

Please see Section R1.2.

I propose the following modifications to address these issues:
a) Change the title to remove "cutting out the middleman".

We respectfully disagree. The "middleman" in the title refers to the modeling activities typically used to derive level 3 and 4 products such as GPP and LAI, and in targeting lower-level reflectance data products, we have successfully "cut this middleman out". Taking into account the significant re-framing of the discussion (Section R1.2) in this revision, we feel this title is still accurate.

b) Modify the discussion around this point, including stating clearly that the reflectances and reflectance factors are physically different things and that, in general, it is not appropriate to do assimilate one into a model that predicts the other.

See Section R1.2.

c) Quantify the differences between the BRFs and the modelled BHRs. I noticed, poking around in the github repository, that the SAIL model is included and hence, presumably, it is trivial to take representative posterior parameters values and model both BRF (from SAIL) and BHRs. If the authors' assertion that it shouldn't make any difference is correct then this will help to defend that. The observed reflectance factors should also be compared against SAIL predictions. I would be happy to iterate on the experimental procedure with the authors.

We have added to the results a comparison of simulations between EDR and PRO4SAIL, parameterized identically (Figures 10; lines 442–445). Lower albedo predictions from EDR than SAIL can be attributed primarily to differences in how each model defines the direct radiation backscatter coefficient in the radiative

transfer equation — see our main response (Section R1.3).

The reason that we ultimately did not include SAIL simulations is that there is no obvious way to convert the complex, multi-PFT canopies simulated by EDR into the single homogeneous inputs expected by SAIL. Even if we performed an analogous multi-site calibration of PRO4SAIL, the only calibrated parameters we could compare directly would be the total LAI (and even that would not be a fully apples-to-apples comparison because a PRO4SAIL would optimize that parameter directly for each site, whereas in EDR, we are targeting multiple parameters that interact to give LAI)—for the remaining parameters, we would still have to aggregate the multi-PFT EDR parameters in a non-trivial, non-obvious way.

2. Is ED2 actually used in this paper? It seems that it is only a relatively small component of the model (i.e. the canopy radiative transfer scheme) that has been extracted. I think the title is slightly misleading in this respect. More important, I am not sure calling the code ED2-PROSPECT5 is appropriate. Perhaps EDR-PROSPECT5 would be better?

No, ED2 is not used directly in this analysis. However, the results of this analysis can be used directly to parameterize ED2. The R version of the ED2 canopy radiative transfer model is mathematically identical to the actual Fortran version and was only extracted for analytical convenience and computationally efficiency (since running the original version of ED2, which is what we initially implemented, incurred significant computational overhead for initializing ED2's many variables).

We have kept the title as-is, but have added additional text to the discussion highlighting how this approach could be used directly with the ED2 model (lines 475–482).

3. How are correlations between spectral channels dealt with? Do the authors use all of the AVRIS observations in the domain 400-1300nm? The errors in spectrally adjacent bands will be very highly correlated and the overall information content will be much lower the same number of independent observations.

Please see Section R1.1.

Minor comments:
L27 The MacBean et al. (2018) paper referenced here does not assimilate a derived product in the sense the authors mean it. The assimilated data in that paper is solar induced fluorescence which, whilst it is "derived" in the sense that it is not what is being measured by the sensor, is no more derived that the surface reflectance data used in the manuscript under review. I suggest finding another reference here.

We have revised this accordingly.

L35 "Meanwhile, the estimating" — "Meanwhile, estimating"

We have revised this accordingly.

L61 Surface reflectance is not assimilated in Zobbitz et al., (2014). The title is misleading (and I regret not standing my ground on that issue when we submitted that paper!); in fact fAPAR data is assimilated.

We have revised this accordingly.

L85 Not sure this line should finish with a colon.

We have revised this accordingly.

Eqn 3 What is meant by $r_{n+1}$ (and other variables with that subscript)? If I have understood correctly this is there are $n$ layers, so what is the reflectance of the $n+1$ layer? (I am sure I have just missed something here, but it was not obvious).

EDR models fluxes not at canopy layers, but at the boundaries between layers. Boundary $i$ is defined as the air space immediately below the $i^{th}$ canopy layer (counting from the bottom of the canopy). For $n$ canopy layers, $i = n$ refers to the space immediately below the tallest cohort layer, while $i = n + 1$ is the (imaginary) boundary above the tallest cohort (i.e., between the canopy and the atmosphere). Therefore, $r_{n+1}$ refers to the reflectance at the top of the canopy. We have added this clarification to the methods (lines 221–228).

L102 "tau" should presumably be the Greek symbol $\tau$.

In the revised methods, this comment no longer applies.

Eqn 7 I am confused by this, why is forward scattering defined by the sum of R+T? This is just the total scattering isn't it?

In the revised methods, this comment no longer applies.

L188 This a different definition of X from earlier in the paper. I suggest finding a different symbol.

We have replaced the earlier definition of $\mathbf{X}$ (in the EDR matrix equation $\mathbf{M}X = Y$) with the symbol A to eliminate this ambiguity.

L273 "DALEC-predicted foliar biomass, which required introducing an additional fixed parameter (grams of leaf carbon per leaf area) present in neither model." This statement is incorrect – that parameter already existed DALEC.

We have removed this sentence from our discussion.

**References**

Bonan, G. B. (2008). Forests and Climate Change: Forcings, Feedbacks, and the Climate Benefits of Forests. *Science*, 320(5882):1444–1449.

Dietze, M. C., Serbin, S. P., Davidson, C., Desai, A. R., Feng, X., Kelly, R., Kooper, R., LeBauer, D., Mantooth, J., McHenry, K., and Wang, D. (2014). A quantitative assessment of a terrestrial biosphere model's data needs across North American biomes. *Journal of Geophysical Research: Biogeosciences*, 119(3):2013JG002392.

Lucht, W., Schaaf, C. B., and Strahler, A. H. (2000). An algorithm for the retrieval of albedo from space using semiempirical BRDF models. *IEEE Transactions on Geoscience and Remote Sensing*, 38(2):977–998.

Raczka, B., Dietze, M. C., Serbin, S. P., and Davis, K. J. (2018). What limits predictive certainty of long-term carbon uptake? *Journal of Geophysical Research: Biogeosciences*, 123(12):3570–3588.

Shiklomanov, A. N., Bond-Lamberty, B., Atkins, J. W., and Gough, C. M. (2020). Structure and parameter uncertainty in centennial projections of forest community structure and carbon cycling. *Global Change Biology*, 26(11):6080–6096.

Singh, A., Serbin, S. P., McNeil, B. E., Kingdon, C. C., and Townsend, P. A. (2015). Imaging spectroscopy algorithms for mapping canopy foliar chemical and morphological traits and their uncertainties. *Ecological Applications*, 25(8):2180–2197.

Viskari, T., Shiklomanov, A., Dietze, M. C., and Serbin, S. P. (2019). The influence of canopy radiation parameter uncertainty on model projections of terrestrial carbon and energy cycling. *PLOS ONE*, 14(7):e0216512.

Yuan, H., Dai, Y., Dickinson, R. E., Pinty, B., Shangguan, W., Zhang, S., Wang, L., and Zhu, S. (2017). Reexamination and further development of two-stream canopy radiative transfer models for global land modeling. *Journal of Advances in Modeling Earth Systems*, 9(1):113–129.

---

## Author Response (AR2)

**Response to reviewers**

April 9, 2021

We are pleased to present a minor revision of our manuscript titled "Cutting out the middleman: Calibrating and validating a dynamic vegetation model (ED2-PROSPECT5) using remotely sensed surface reflectance" for consideration for publication in *Geoscientific Model Development*. We are very grateful to the two reviewers for their constructive feedback on our revised manuscript.

**1 Reviewer 1**

This reviewer was generally very pleased with the detailed nature of the response to this reviewers comments related to edge hitting parameters and equifinality. Below are a few follow up questions based on the author response.

Author Response: "As shown in equation 48, observation error in the reflectance data was not estimated a priori based on the instrument itself, but was modeled as the residual error between the model and the data, analogous to what is done for any linear or nonlinear regression model." "Furthermore, because the variance slope and intercept are fit parameters, whose parametric uncertainty is being quantified and propagated, this makes it even less likely that our uncertainty estimate is overconfident. That said, the current approach does not formally account for any possible systematic errors in the observations, which could have a more serious impact on inferences. However, we would note that we are unaware of any derived data products that account for these systematic errors either."

Response: This reviewer is well aware of this challenge, and recognizes in the absence of uncertainty estimates provided by the data product, end users are forced to make assumptions, or guess at how this may influence their assimilation. It is a bit concerning that this approach conflates potential instrument error and (known) model structural error. Perhaps use this as recommendation or call for data providers to give more quantitative estimate of uncertainty.

While we agree that uncertainty estimates in reflectance products are important, there is no obvious place to add this recommendation without adding another paragraph to what is already a long and multifaceted discussion. Therefore, we have decided not to add any additional recommendations to our discussion.

Specific comments on edited manuscript, using line-numbers as shown in tracked changes manuscript:
Line 4-7: "In addition parameters to which vegetation models are known to be highly sensitive to...",
Maybe simplify these 2-3 lengthy sentences to something more concise: "In addition certain parameters (e.g. SLA) that provide an outsized influence on vegetation model behavior, can be constrained by observations of shortwave radiation, thus reducing model forecast uncertainty." For example.

We have revised this accordingly. This section now reads as follows:

> In addition, certain parameters (e.g., SLA) that have an outsized influence on vegetation model behavior can be constrained by observations of shortwave reflectance, thus reducing model predictive uncertainty.

Line 16-17: 'Successfully constrained' This is a bit vague, maybe say something like: 'significantly reduced the parameter uncertainty'

We have revised this accordingly. This sentence now reads as follows (in both the Abstract and Conclusions):

The calibration significantly reduced uncertainty in model parameters related to leaf biochemistry and morphology and canopy structure for five plant functional types.

Line 25: "In addition, we also highlight that our specific implementation is only valid for hemispherical reflectance data (a.k.a., albedo), whereas most surface reflectance products actually estimate the directional reflectance factor. Fortunately, the assumptions and parameters that define our hemispherical reflectance model and many others in the vegetation modeling community are readily adaptable to the prediction of directional reflectance, and we recommend that these adaptations be incorporated into the next generation of vegetation models."

This paragraph seems a bit strange without clarifying. Need something like: "In this work the reflectance product was converted to hemispherical reflectance in order to directly compare with the model, however, in future work, we recommend that vegetation models add the capability to predict directional reflectance." Understandably bringing the observations closer to the model output goes against the grain of the manuscript, and tempers some of the 'novelty' in this manuscript, but is necessary.

We have revised this accordingly. This section now reads as follows:

In addition, we also highlight that, to directly compare with a two-stream radiative transfer model like EDR, we had to perform an additional processing step to convert the directional reflectance estimates of AVIRIS to hemispherical reflectance (a.k.a., "albedo"). In future work, we recommend that vegetation models add the capability to predict directional reflectance, to allow them to more directly assimilate a wide range of airborne and satellite reflectance products.

Line 465: "our work is novel because it uses a canopy radiative transfer formulation that already exists inside the model itself."

It's unclear in this context if you are considering PROSPECT-5 internal or external to the model. Clearly it is internal to EDR, but external to ED2. In this context you are referring to the two-stream approach within ED2 as being internal to the model. PROSPECT-5 is tacked on to simulate leaf reflectance and transmittance. There is nothing 'magical' about being internal or external to a certain model, but it must be internal to the data assimilation system – which in this case includes ED2 and PROSPECT-5. Suggest reframing internal/external terminology to mean internal to the data assimilation system – internal to the model terminology is a bit confusing.

Considering this comment together with Reveiwer T. Quaife's comment about the same paragraph, we have decided to cut this paragraph and expand the subsequent paragraph, which now reads as follows:

In this study, the vegetation composition at each site (including the PFT distribution and size-age structure) was prescribed in detail based on inventory data. This allowed us to focus the calibration on model parameters related canopy radiative transfer model parameters. However, ED2 is a dynamic vegetation model whose core purpose is to predict how vegetation composition and structure evolve through time. An important future direction of this work is to evaluate such dynamic ED2 simulations where vegetation composition and structure are predicted with some uncertainty. In ED2, shortwave canopy radiative transfer is already linked (through shared parameters and state variables) to other important model processes, including thermal radiative transfer, micrometeorology, photosynthesis, respiration, and competition (Longo et al. 2019), and therefore, changes in canopy radiative transfer parameters have profound consequences for ED2 predictions of ecosystem fluxes and composition (Viskari et al. 2019). Future work could further strengthen this link by embedding the PROSPECT coupling demonstrated in this study into ED2 itself, replacing ED2's currently prescribed leaf optical properties with simulated optical properties that change with leaf morphology and biochemistry. For example, the PROSPECT leaf water content parameter ($Cw$) provides a physical link between leaf optical properties and hydraulics, so such a configuration could allow surface reflectance information to constrain ED2's recently developed dynamic hydraulics module (Xu et al. 2021).

Line 515-530: This is an appropriate discussion in response to edge hitting parameters.

We are pleased that our revision addressed this comment.

Line 565-585: I am generally satisfied with this explanation for how AVIRIS data is valid for comparison with reflectance simulated by EDR. I do think – a schematic that compares the extra steps required to bring AVIRIS data to something resembling reflectance would be helpful. Also bringing the observations closer to the model goes against the main advice posed by the authors – that is include as much of the model as possible to bring it closer to the observations. Include the model within the data assimilation system.

**2    Reviewer 2**

The authors have done a commendable job in responding to comments and improving what was already an excellent paper. In particular I appreciate the rewriting of the equations for the RT scheme which is now much clearer and, I think, has dealt with a couple of inconsistencies that I alluded to in my original review. There is also some important discussion added in response to my comments and I am thankful to the authors for that.

I still have some concerns although hopefully this is now just down to nomenclature. As with my original review these centre around what the AVRIS data actually represents.

1. Around line 255, the manuscript describes a BRDF "correction" applied to the AVRIS data, which apparently produces albedo values. Assuming this is correct then it satisfies a lot of my original issues with the manuscript, but it is not entirely clear from the text. Typically a BRDF correction produces directional reflectance values that have been normalised to a common geometry — not albedo values — and indeed this is a common application of the Ross-Li kernel models. Of course it is also possible to integrate the kernel BRDF models to estimate albedo and quite possibly that is what has been done in the AVRIS processing chain. Doing this would also represent a geometric correction of sorts. The sentence that says "a polynomial approximation to the Ross-Li semi-empirical BRDF model" provides some hope that it is the latter, as the operational MODIS algorithm used to predict albedo from the kernel BRDF model approximates the integral of the kernels using a polynomial. Note that the MODIS BRDF correction (the Nadir BRDF-Adjusted Reflectance in MCD43) is not produced that way however.

I have read the relevant parts of the Singh et al. paper and it does not really shed any light on this unless the detail is hidden elsewhere (and I note that in that paper only the volumetric kernel is mentioned, which could be the case but would be a poor choice to represent the types of forest that are being used in the current paper — I actually suspect that what the Singh paper describes is not an accurate reflection of the BRDF processing).

Please can the authors clarify exactly what has been done here? Hopefully the BRDF correction has produced albedo values that can legitimately be compared to a two-stream model. If not then I think much of the authors rebuttal of my original review is invalid.

We have clarified that that we used the hemispherical integral of the fitted BRDF kernel to estimate the albedo. The revised sentence reads as follows:

> Briefly, this BRDF correction estimates "intrinsic surface albedo"—the quantity that is simulated by EDR—from directional reflectance data by fitting a polynomial approximation to the Ross-Li semiempirical BRDF model and then integrating this model over all angles.

We agree that exact choice of BRDF kernel is important in such a processing procedure. However, rather than devoting significant additional discussion to choice of BRDF kernels, we feel a much better recommendation that avoids all of these pitfalls is for vegetation models to use canopy RTMs capable of simulating directional reflectance. We already made this recommendation in our discussion, and we state this more clearly in our revised abstract, which how has the following sentences:

> In addition, we also highlight that, to directly compare with a two-stream radiative transfer model like EDR, we had to perform an additional processing step to convert the directional reflectance estimates of AVIRIS to hemispherical reflectance (a.k.a., "albedo"). In future work, we recommend that vegetation models add the capability to predict directional reflectance, to allow them to more

2. A related but minor point at line 250: "Atmospheric correction routines use this level 1 radiance product to estimate the surface reflectance (technically, hemispherical-directional reflectance factor, HDRF, sensu Schaepman-Strub et al. 2006)—a quantity that is (in theory) independent of illumination conditions and therefore can be more directly related to intrinsic physical properties of the surface."

I think this rather depends on how you define a HDRF. In the Schaepman-Strub paper it is defined as having a component of direct radiation and hence it is not true to say that it is independent of illumination conditions (their equation 14 is a clear example of this). I can, however, believe that this is what's produced in the AVRIS processing chain. Assuming I am correct about this the authors just need to delete the second half of the sentence and I think nothing else is affected.

On the other hand, if I am wrong and this really is under perfectly diffuse illumination conditions (and hence independent of sun angle) then it is unclear why the processing also includes a BRDF correction. That would need very careful explaining in the manuscript.

Please clarify.

We have taken the reviewer's first suggestion and eliminated part of this sentence. The revised sentence now reads as follows:

> Atmospheric correction routines use this level 1 radiance product to estimate the surface reflectance (technically, hemispherical-directional reflectance factor, HDRF, sensu Schaepman-Strub et al. 2006)—a quantity that can be more directly related to intrinsic physical properties of the surface.

3. Around line 400: "This reduces the number of new assumptions and variables we have to introduce and increases the extent to which constraint on canopy radiative transfer parameters propagates to other related processes in the model (Viskari et al., 2019). More importantly, in an external RTM, the only way that observed reflectance constrains the model is through the foliar biomass, and additional information from the reflectance on canopy structure is confined to the external RTM parameters."

I mentioned these points in my original review and I am not sure I saw a response (which is not a complaint, I realise there was a lot to deal with). However, I think these sentences are best deleted. It isn't true that using an RT model that is inside an ecosystem model necessarily reduces the number of assumptions and variables; that depends entirely on one's choice of RT model (both the internal and external ones). It is possible to define a full BRDF model using exactly the same variables and assumptions as those in a two-stream model.

It is also not true that the only way an external RT model can influence the process based model is via foliar biomass. It is entirely possible to build additional connections between them, to the extent where one has encompassed all the processes that they have in common.

Overall I find this part of the discussion unconvincing and I suggest it is removed.

Taking this comment together with a similar comment from Reviewer 1, we have removed this paragraph and revised the subsequent paragraph to read as follows:

> In this study, the vegetation composition at each site (including the PFT distribution and size-age structure) was prescribed in detail based on inventory data. This allowed us to focus the calibration on model parameters related canopy radiative transfer model parameters. However, ED2 is a dynamic vegetation model whose core purpose is to predict how vegetation composition and structure evolve through time. An important future direction of this work is to evaluate such dynamic ED2 simulations where vegetation composition and structure are predicted with some uncertainty. In ED2, shortwave canopy radiative transfer is already linked (through shared parameters and state variables) to other important model processes, including thermal radiative transfer, micrometeorology, photosynthesis, respiration, and competition (Longo et al. 2019), and therefore, changes in canopy radiative transfer parameters have profound consequences for ED2 predictions of ecosystem fluxes and composition (Viskari et al. 2019). Future work could further strengthen this link by embedding the PROSPECT coupling demonstrated in this study into ED2 itself, replacing ED2's currently prescribed leaf optical properties with simulated optical properties that change with leaf morphology and biochemistry. For example, the PROSPECT

leaf water content parameter ($Cw$) provides a physical link between leaf optical properties and hydraulics, so such a configuration could allow surface reflectance information to constrain ED2's recently developed dynamic hydraulics module (Xu et al. 2021).